# FTO promotes weight gain via altering *Kif1a* splicing and axonal vesicle trafficking in AgRP neurons

Daisuke Kohno [ID][1✉], Reika Kawabata-Iwakawa[2], Sotaro Ichinose [ID][3], Shigetomo Suyama[4,5], Kazuto Ohashi [ID][6], Winda Ariyani [ID][1], Tetsushi Sadakata[7], Hiromi Yokota-Hashimoto[1], Ryosuke Kobayashi[8], Takuro Horii[8], Vina Yanti Susanti [ID][1,9], Ayumu Konno [ID][10,11], Haruka Tsuneoka[1], Chiharu Yoshikawa[1], Sho Matsui[12], Akihiro Harada [ID][13], Toshihiko Yada[5,14,15,16], Izuho Hatada[8,11], Hirokazu Hirai[10,11], Masahiko Nishiyama [ID][17], Tsutomu Sasaki [ID][12] & Tadahiro Kitamura[1]

## Abstract

N⁶-methyladenosine (m⁶A) is an abundant chemical RNA modification involved in the regulation of many biological processes. The m⁶A demethylase FTO (fat mass and obesity-associated protein) is known to affect body weight, but its systemic context and underlying mechanisms remain unclear. Here, we found that mice lacking or overexpressing *Fto* in agouti-related peptide-expressing (AgRP) neurons in the hypothalamus exhibited decreased and increased body weight, respectively. FTO demethylated m⁶A on mRNAs for proteins associated with membrane trafficking and alternative splicing in AgRP neurons. Downstream, FTO-modulated alternative splicing of the axonal motor protein *Kif1a* affected its hinge region, which is relevant to the structure and function of KIF1A. Notably, *Kif1a* knockdown in AgRP neurons suppressed the weight gain of mice overexpressing *Fto*. In addition, FTO increased the trafficking and secretion of dense-core vesicles containing neuropeptides NPY and AgRP from AgRP neurons. Collectively, these results reveal a novel regulatory FTO-KIF1A axis in the brain affecting appetite-stimulating AgRP neurons and systemic energy homeostasis, via FTO regulation of the epitranscriptome of AgRP neurons.

**Keywords** AgRP; Alternative Splicing; FTO; KIF1A; Obesity
**Subject Categories** Membranes & Trafficking; Metabolism; Neuroscience

## Introduction

Obesity rates have increased continuously in most countries over the past several decades (NCD Risk Factor Collaboration, 2016). However, the molecular mechanisms underlying this increase are not fully understood. The obesity epidemic is often attributed to environmental factors, including diet and exercise (Hill, 2006; Pate et al, 2013; Ross et al, 2016), and the mechanisms by which the body responds to these factors may be the basis for the obesity epidemic. Recent studies have shown that chemical epigenetic and epitranscriptomic modifications play an important role in many biological processes, as they respond sensitively to changes in both the internal and external environments (Li et al, 2020; Widagdo et al, 2016). The formation of N⁶-methyl-adenosine (m⁶A) is the most abundant methyl modification of RNA (Zaccara et al, 2019) and directly affects RNA stability, translational efficiency, and alternative splicing (Shi et al, 2019). Furthermore, m⁶A modifications regulate several physiological processes, including embryonic development, cancer, DNA repair, and brain function (Liu et al, 2023; Widagdo et al, 2016).

m⁶A modifications are regulated by the methyltransferase-like 3 and 14 (METTL3-METTL14) complex and two demethylases—fat mass and obesity-associated protein (FTO) and AlkB homolog 5 (ALKBH5) (Shi et al, 2019). *Fto* was originally identified through genome-wide association study (GWAS) analyses, which revealed that single-nucleotide polymorphisms (SNPs) in *Fto* intron 1 were strongly associated with obesity (Dina et al, 2007; Frayling et al, 2007). The genotypes associated with these *Fto* SNPs affect the expression level of *Fto* (Berulava and Horsthemke, 2010; Pahl et al, 2023; Stratigopoulos et al, 2016; Zhang et al, 2023) and neighboring genes such as *Rpgrip1l*, *Irx3*, and *Irx5*, which are related to obesity

[1]Metabolic Signal Research Center, Institute for Molecular and Cellular Regulation, Gunma University, Maebashi, Japan. [2]Division of Integrated Oncology Research, Gunma University Initiative for Advanced Research (GIAR), Maebashi, Japan. [3]Department of Anatomy, Gunma University Graduate School of Medicine, Maebashi, Japan. [4]Department of Physiology, Keio University School of Medicine, Tokyo, Japan. [5]Division of Integrative Physiology, Department of Physiology, Jichi Medical University School of Medicine, Shimotsuke, Japan. [6]Institute for Molecular and Cellular Regulation, Gunma University, Maebashi, Japan. [7]Education and Research Support Center, Gunma University Graduate School of Medicine, Maebashi, Japan. [8]Laboratory of Genome Science, Biosignal Genome Resource Center, Institute for Molecular and Cellular Regulation, Gunma University, Maebashi, Japan. [9]Internal Medicine Department, Faculty of Medicine, Gadjah Mada University, Yogyakarta, Indonesia. [10]Department of Neurophysiology & Neural Repair, Gunma University Graduate School of Medicine, Maebashi, Japan. [11]Viral Vector Core, Gunma University, Initiative for Advanced Research (GIAR), Maebashi, Japan. [12]Division of Food Science and Biotechnology, Graduate School of Agriculture, Kyoto University, Kyoto, Japan. [13]Department of Cell Biology, Graduate School of Medicine, Osaka University, Suita, Osaka, Japan. [14]Division of Integrative Physiology, Center for Integrative Physiology, Kansai Electric Power Medical Research Institute, Kyoto, Japan. [15]Department of Diabetes, Endocrinology and Metabolism/Rheumatology and Clinical Immunology, Gifu University Graduate School of Medicine, Gifu, Japan. [16]Center for One Medicine Innovative Translational Research, Gifu University Institute for Advanced Study, Gifu, Japan. [17]Gunma University, Maebashi, Japan. ✉E-mail: daisuke.kohno@gunma-u.ac.jp

(Smemo et al, 2014). Moreover, *Fto*-knockout (Fischer et al, 2009) and *Fto*-overexpressing (Church et al, 2010) mice with unaltered *Fto* intron 1 SNPs exhibit strong body weight phenotypes and are lean and obese, respectively. These findings suggest that FTO is closely associated with body weight control.

However, the mechanisms by which FTO affects body weight are not fully understood. FTO expression in adipose tissue has been reported to be involved into body weight control (Grunnet et al, 2009; Wang et al, 2015; Wu et al, 2021; Zhang et al, 2023), but may not sufficiently explain the overall effects of FTO. Notably, brain-specific *Fto*-knockout mice also become lean, similar to whole-body *Fto*-knockout mice (Gao et al, 2010), suggesting that FTO in the brain plays a key role in controlling body weight, although the specific regions of the brain involved in the effects of FTO on body weight remain unknown. FTO is widely expressed in the brain (McTaggart et al, 2011; Olszewski et al, 2009), but its expression in the hypothalamus is higher than in most other brain areas and peripheral tissues (Gerken et al, 2007; Stratigopoulos et al, 2008).

The hypothalamic feeding center, a key brain region for body weight control, comprises several nuclei and neuron groups and integrates systemic energy signals from food intake and metabolism (Williams and Elmquist, 2012). Agouti-related peptide-expressing (AgRP) neurons, a major neuronal group, are highly colocalized with neuropeptide Y (NPY) expression (Hahn et al, 1998) and play a significant role in hunger by enhancing food-seeking behavior (Alcantara et al, 2022) and increasing the efficiency of nutrient utilization (Cavalcanti-de-Albuquerque et al, 2019; Joly-Amado et al, 2012). Although the effect of FTO on the hypothalamus has been suggested (Liu et al, 2024; Stratigopoulos et al, 2008; Tung et al, 2010), little is known about the influence of FTO in specific hypothalamic neurons on body weight. Here, we sought to identify the feeding center neurons involved in FTO-mediated control of body weight and to determine how FTO influences these neurons through epitranscription in order to obtain novel insights into the mechanisms by which feeding centers control body weight.

## Results

### FTO in AgRP neurons is necessary for weight gain

To determine which neuronal populations mediate the effects of FTO on body weight, we first analyzed the distribution of *Fto* mRNA-expressing cells in the brain (Fig. EV1A,B). *Fto* mRNA was widely expressed in the brain and particularly abundant in the hypothalamic nuclei of the feeding center, including the arcuate nucleus (ARC), ventromedial hypothalamus (VMH), and paraventricular nucleus (PVH), which is consistent with previous reports (McTaggart et al, 2011; Olszewski et al, 2009). *Fto-LacZ* mice, in which *LacZ* was inserted into *Fto*, were used to examine *Fto* expression in the main feeding neurons (Fig. EV1C–P). *Fto* was expressed in ~80% of NPY/AgRP neurons, 60% of pro-opiomelanocortin (POMC) neurons, and 40% of neurons expressing tyrosine hydroxylase, NUCB2, thyrotropin-releasing hormone, and oxytocin. These results indicated that many feeding center neurons express *Fto*, particularly NPY/AgRP neurons, as the majority of them expressed *Fto*.

Using several conditional *Fto*-knockout mice lines, we screened for the major hypothalamic feeding center neurons in which FTO plays an indispensable role in body weight control. Importantly, these mice lacked *Fto* exon 3, but the SNPs in *Fto* intron 1

remained unaltered. The body weights of AgRP neuron-specific *Fto* deletion mice (*Fto*^lox/lox^/*Agrp*-Cre) were significantly lower than those of control mice from 5 weeks of age (Fig. 1A,B). In contrast, the body weights of mice with *Fto* deletions specific for POMC (*Pomc*-Cre) and PVH (*Sim1*-Cre) and VMH (*Sf1*-Cre) neurons were comparable to those of control mice (Fig. EV2). These findings indicate that FTO in AgRP neurons, but not in POMC neurons and PVH and VMH neurons, is indispensable for body weight control.

Mice with an AgRP neuron-specific *Fto* deletion had a lower body fat percentage than control mice (Fig. 1C), although their body lengths were comparable (Fig. 1D), indicating that *Fto*^lox/lox^/*Agrp*-Cre mice were lean. Daily food intake was also significantly lower in *Fto*^lox/lox^/*Agrp*-Cre mice (Fig. 1E). Feeding behavior, assessed based on accessing the food chamber, decreased during the first hour of the ad libitum dark phase, food deprivation, and refeeding (Fig. 1F), suggesting that food-seeking behavior, primarily induced by AgRP neurons (Alcantara et al, 2022; Gouveia et al, 2021; Sternson and Eiselt, 2017), was reduced in *Fto*^lox/lox^/*Agrp*-Cre mice. However, oxygen consumption and locomotor activity were comparable (Fig. 1G,I), suggesting that a reduction in food intake, rather than increased energy expenditure, was the main cause of leanness in *Fto*^lox/lox^/*Agrp*-Cre mice. In addition to a lower respiratory exchange ratio (RER) (Fig. 1H), blood levels of free fatty acids and ketone bodies, but not triglycerides, as well as low- and high-density lipoproteins, were significantly (~twofold) lower (Fig. 1J–M), suggesting that systemic lipid utilization, rather than lipolysis of adipose tissue, increases in *Fto*^lox/lox^/*Agrp*-Cre mice. Administration of SR59230A, a β3 adrenergic receptor antagonist, restored the decreased levels of RER, free fatty acids, and ketone bodies in *Fto*^lox/lox^/*Agrp*-Cre mice (Fig. EV3), suggesting that the sympathetic nervous system, which mediates AgRP neuron-induced nutrient partitioning (Joly-Amado et al, 2012), is responsible for these phenotypes observed in *Fto*^lox/lox^/*Agrp*-Cre mice. Moreover, *Fto*^lox/lox^/*Agrp*-Cre mice were resistant to high-fat diet (HFD)-induced obesity (Fig. 1N), suggesting that FTO expression is necessary for that phenotype. These data indicated that FTO in AgRP neurons is necessary for the normal control of feeding behavior, energy expenditure, and body weight, which are the primary regulatory functions of these neurons.

### FTO in AgRP neurons promotes weight gain

To determine whether an increase in FTO level in AgRP neurons would enhance weight gain, we overexpressed *Fto* in these neurons by injecting AAV-hSyn-Flex-*Fto*-mCherry into *Agrp*-Ires-Cre mice to obtain mice with *Fto*^AgRP^ overexpression (Fig. EV4). *Fto*^AgRP^-overexpressing mice exhibited a significant increase in body weight from 3 weeks after injection (Fig. 2A,B), and their body fat percentage was also significantly higher than that of control mice (Fig. 2C), suggesting that FTO in AgRP neurons is not only indispensable for normal energy homeostasis but also promotes weight gain. In addition, body length at 25 weeks of age was slightly but significantly increased (Fig. 2D), suggesting that increased FTO levels in AgRP neurons also promoted some linear growth concomitant with weight gain. Cumulative food intake was significantly higher in *Fto*^AgRP^-overexpressing mice (Fig. 2E), whereas short-term feeding behavior did not differ significantly (Fig. 2F). Furthermore, oxygen consumption was significantly

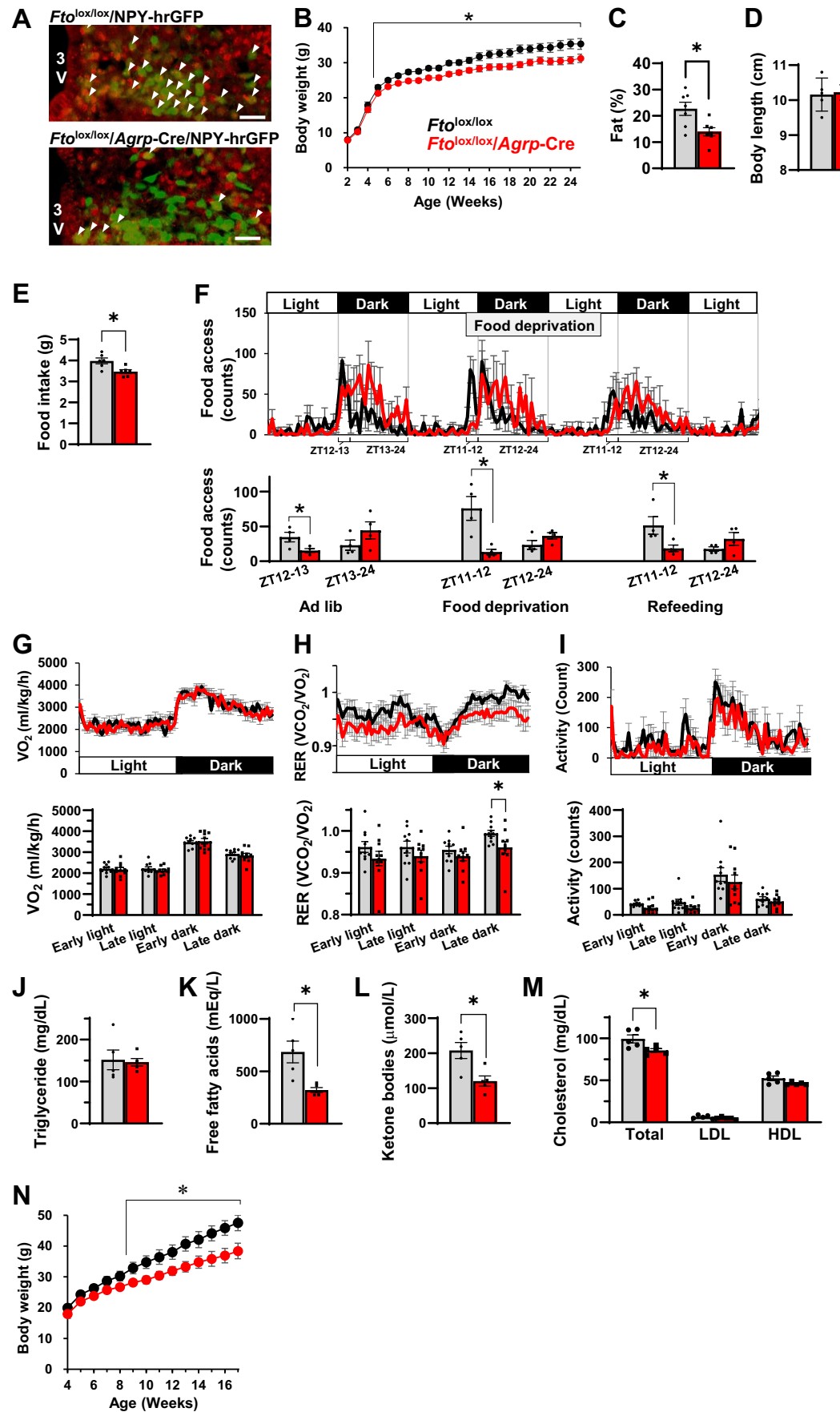

**Figure 1. FTO in AgRP neurons is necessary for weight gain.**

(A) FTO immunofluorescence (red) was colocalized with most NPY-hrGFP fluorescence (green) in $Fto^{lox/lox}$/NPY-hrGFP mice (arrow), but FTO immunofluorescence (red) was not colocalized with most NPY-hrGFP fluorescence (green) in $Fto^{lox/lox}$/Agrp-Cre/NPY-hrGFP mice. Scale bar: 30 μm. (B–D) Body weight ($n = 12$–18) (B), percentage of body fat at 25 weeks of age ($n = 7$) (C), and body length at 25 weeks of age ($n = 5$) (D) of male $Fto^{lox/lox}$ and $Fto^{lox/lox}$/Agrp-Cre mice. Error bars represent SEM. Data were analyzed using unpaired Student's $t$ test. *$P < 0.05$ (exact $P$ values: 0.049 [5-week-old], 0.012 [6-week-old], 0.002 [7-week-old], 0.001 [8-week-old], 0.001 [9-week-old], 0.001 [10-week-old], 0.001 [11-week-old], 0.001 [12-week-old], 0.004 [13-week-old], 0.007 [14-week-old], 0.017 [15-week-old], 0.017 [16-week-old], 0.005 [17-week-old], 0.008 [18-week-old], 0.004 [19-week-old], 0.009 [20-week-old], 0.018 [21-week-old], 0.022 [22-week-old], 0.012 [23-week-old], 0.006 [24-week-old], 0.032 [25-week-old] (B), 0.011 (C)). (E–I) Daily food intake ($n = 6$) (E), food access during ad libitum, food deprivation and refeeding ($n = 4$) (F), oxygen consumption ($n = 10$) (G), RER ($n = 10$) (H) and locomotor activity ($n = 10$) (I) of 10-week-old male $Fto^{lox/lox}$ and $Fto^{lox/lox}$/Agrp-Cre mice. Error bars represent SEM. Data were analyzed using unpaired Student's $t$ test (E, F, ad lib, refeeding), H) and Welch's unpaired $t$ test (F, food deprivation). *$P < 0.05$ (exact $P$ values: 0.011 (E), 0.041 [ad lib, ZT12-13], 0.031 [food deprivation, ZT11-12], 0.049 [refeeding, ZT11-12] (F), 0.049 (H)). (J–M) Triglyceride ($n = 5$) (J), free fatty acid ($n = 5$) (K), ketone bodies ($n = 5$) (L), and cholesterol ($n = 5$) (M) levels in the serum of 6-week-old male $Fto^{lox/lox}$ and $Fto^{lox/lox}$/Agrp-Cre mice. Error bars represent SEM. Data were analyzed using Student's unpaired $t$ test (L, M) and Welch's unpaired $t$ test (K). *$P < 0.05$ (exact $P$ values: 0.022 (K), 0.012 (L), 0.028 (M)). (N) Body weights of male $Fto^{lox/lox}$ ($n = 6$–8) and $Fto^{lox/lox}$/Agrp-Cre mice ($n = 6$–8) on an HFD. The black lines and gray bars indicate $Fto^{lox/lox}$ mice and the red lines and bars indicate $Fto^{lox/lox}$/Agrp-Cre mice. Error bars represent SEM. Data were analyzed using unpaired Student's $t$ test. *$P < 0.05$ (exact $P$ values: 0.042 [9-week-old], 0.029 [10-week-old], 0.043 [11-week-old], 0.042 [12-week-old], 0.022 [13-week-old], 0.042 [14-week-old], 0.029 [15-week-old], 0.018 [16-week-old], 0.022 [17-week-old]). Source data are available online for this figure.

lower in $Fto^{AgRP}$-overexpressing mice (Fig. 2G), whereas RER and locomotor activity were not altered (Fig. 2H,I), suggesting that increased food intake and decreased energy expenditure caused the weight gain.

Both m⁶A writers and erasers regulate the frequency of m⁶A modifications (He and He, 2021; Zaccara et al, 2019). Therefore, we then investigated whether the loss of function of METTL3, a key component of the m⁶A writer complex (Liu et al, 2023), would result in a phenotype similar to that of $Fto^{AgRP}$-overexpressing mice. AgRP neuron-specific $Mettl3$ knockout mice had body weights similar to those of control mice (Figs. 2J and EV5), suggesting that m⁶A modifications in AgRP neurons controlled specifically by FTO contribute to weight gain. The phenotypic discrepancy between $Fto^{AgRP}$-overexpressing and AgRP neuron-specific $Mettl3$ knockout mice highlights the unique role of FTO in body weight control.

## FTO demethylates m⁶A modifications in the exons and introns of genes encoding proteins involved in alternative splicing and trafficking

Next, we explored the molecular mechanisms underlying the FTO-induced systemic phenotypes using m⁶A immunoprecipitation of $Fto^{lox/lox}$/Agrp-Cre mouse ARC samples, followed by RNA-seq (m⁶A-seq) to investigate the substrate RNAs of FTO in AgRP neurons (Fig. 3A–D). m⁶A peaks were identified by comparison with the input control, and FTO-specific m⁶A demethylation sites were determined by comparing the m⁶A peak profiles of $Fto^{lox/lox}$ mice with those of $Fto^{lox/lox}$/Agrp-Cre mice. The IgG control immunoprecipitation was omitted from the analysis due to insufficient RNA yield, which represents a limitation of the study. An m⁶A antibody was used to detect both m⁶A and N⁶,2′-O-dimethyladenosine (m⁶Am) modifications (Linder et al, 2015). m⁶Am is a terminal modification adjacent to the 5′-end cap and has a different functional role than that of m⁶A (Sendinc and Shi, 2023). Although both m⁶A and m⁶Am are potential targets of FTO (Mauer et al, 2017), it demethylated mostly the exon and intron regions rather than the 5′-UTRs of pre-mRNA in AgRP neurons (Fig. 3A), implying that m⁶A but not m⁶Am is the major substrate of FTO in AgRP neurons. It has been reported that modifications of m⁶A in the exon and intron regions primarily affect alternative splicing (Bartosovic et al, 2017). Consistently, gene ontology analysis

revealed that genes classified based on the terms "alternative splicing" and "splice variant" were highly enriched among FTO-demethylated genes (Fig. 3B). Peak score analysis revealed that genes associated with membrane trafficking, including $Rims1$ and $Csnk1a1$ (Fig. 3C), had the highest m⁶A demethylation. Furthermore, other membrane trafficking protein-encoding genes such as the Rab protein family genes $Rab6b$ and $Rab7$, kinesin family genes $Kif1a$, $Kif1c$, and $Kif19a$, and splicing factor genes $Rbfox1$ and $Rbfox2$ were highly enriched among FTO-demethylated genes (Fig. 3C,D).

Normal RNA-seq analysis of ARC tissue from $Fto^{lox/lox}$/Agrp-Cre mice was also conducted by analyzing the input RNA m⁶A-seq sample data. A small subset of the FTO-demethylated genes, including as $Rims1$ and $Kif19a$, showed altered RNA expression levels, but most FTO-demethylated genes, including $Csnk1a1$, $Kif1a$, $Rab6b$, $Rabfox1$ and $Rbfox2$, did not (Fig. 3E). Gene ontology analysis of the differentially expressed genes revealed that 10 of the top 15 gene categories (Fig. 3F) were highly shared with those for the FTO-demethylated genes (Fig. 3B). These findings indicate that FTO-induced m⁶A demethylation could influence the expression levels of genes within the same categories.

## FTO affects the alternative splicing of $Kif1a$ and $Cadps2$

FTO expression affecting alternative splicing in AgRP neurons was supported by the following observations: (1) FTO mainly demethylated m⁶A in exons and introns; (2) FTO-demethylated genes included those classified based on the terms "alternative splicing" and "splice variant," and (3) splicing factor genes, $Rbfox1$ and $Rbfox2$ were among the target genes of FTO. Therefore, we speculate that alternative splicing is a downstream function of FTO in AgRP neurons and conducted a comprehensive analysis of alternatively spliced variants to investigate this aspect. To obtain sufficient amounts of RNA from tissue lacking the effect of FTO-induced demethylation, hypothalamic samples from $Tau$-Cre-driven neuron-specific $Fto$ deletion mice ($Fto^{lox/lox}$/$Tau$-Cre) were used (Fig. 4). $Fto^{lox/lox}$/$Tau$-Cre mice exhibited a lower body weight phenotype (Fig. EV6), similar to that of $Nestin$-Cre-driven brain-specific $Fto$ deletion mice (Gao et al, 2010). Interestingly, the ENSMUST00000086819 ($Kif1a$-201) $Kif1a$ isoform had the highest fold change in expression and the lowest $P$ value (Fig. 4A,B). KIF1A is a motor protein that belongs to the kinesin-3 family and plays a

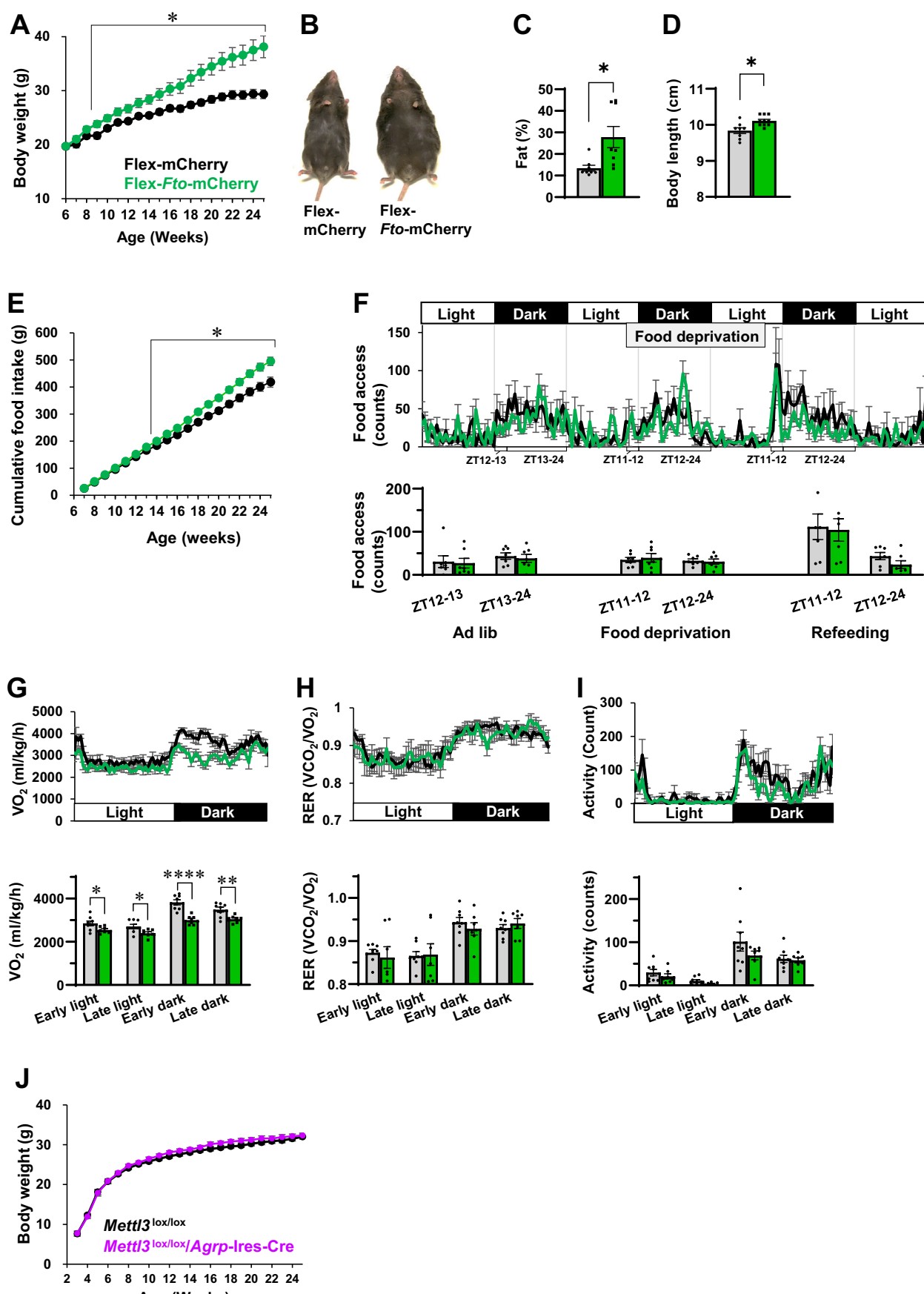

**Figure 2. FTO in AgRP neurons promotes weight gain.**

(A) AgRP neuron-specific *Fto*-overexpressing and control mice were generated by injecting AAV-hSyn-Flex-mCherry (control) or AAV-hSyn-Flex-*Fto*-mCherry into *Agrp*-Ires-Cre mice. Body weight of control mice ($n = 13$–14) and AgRP neuron-specific *Fto*-overexpressing mice ($n = 16$). Error bars represent SEM. Data were analyzed using unpaired Welch's *t* test. *$P < 0.05$ (exact *P* values: 0.021 [9-week-old], 0.026 [10-week-old], 0.027 [11-week-old], 0.011 [12-week-old], 0.019 [13-week-old], 0.005 [14-week-old], 0.014 [15-week-old], 0.011 [16-week-old], 0.010 [17-week-old], 0.004 [18-week-old], 0.002 [19-week-old], 0.001 [20-week-old], 0.001 [21-week-old], 0.002 [22-week-old], 0.0001 [23-week-old], 0.0008 [24-week-old], 0.0006 [25-week-old]). (B) Images of the mice taken at 23 weeks of age. (C, D) Body fat percentage (Flex-mCherry: $n = 8$, Flex-*Fto*-mCherry $n = 8$) (C) and body length (Flex-mCherry: $n = 8$, Flex-*Fto*-mCherry $n = 10$) (D) at 25 weeks of age. Error bars represent SEM. Data were analyzed using unpaired Welch's *t* test (C) and Student's *t* test. *$P < 0.05$ (exact *P* values: 0.021 (C), 0.0068 (D)). (E, F) Cumulative food intake (Flex-mCherry: $n = 18$, Flex-*Fto*-mCherry $n = 19$) (E), and food access during ad libitum, food deprivation, and refeeding ($n = 7$) (F). Error bars represent SEM. Data were analyzed using unpaired Student's *t* test, *$P < 0.05$ (exact *P* values: 0.022 [14-week-old], 0.017 [15-week-old], 0.017 [16-week-old], 0.010 [17-week-old], 0.006 [18-week-old], 0.005 [19-week-old], 0.003 [20-week-old], 0.004 [21-week-old], 0.003 [22-week-old], 0.002 [23-week-old], 0.001 [24-week-old], 0.002 [25-week-old] (E)). (G–I) Oxygen consumption (G), RER (H) and locomotor activity (I) of 14-week-old mice (Flex-mCherry: $n = 8$, Flex-*Fto*-mCherry: $n = 7$). The black lines and gray bars indicate AAV-hSyn-Flex-mCherry, and red lines and red bars indicate AAV-hSyn-Flex-*Fto*-mCherry, respectively. Error bars represent SEM. Data were analyzed using unpaired Student's *t* test, *$P < 0.05$, **$P < 0.01$, ****$P < 0.001$ (exact *P* values: 0.036 [early light], 0.027 [late light], $8.42 \times 10^{-5}$ [early dark], 0.005 [late dark] (G)). (J) Body weight of male *Mettl3*[lox/lox] (black) and *Mettl3*[lox/lox]/*Agrp*-Ires-Cre (magenta) mice ($n = 9$–12). Error bars represent SEM. Source data are available online for this figure.

role in the transport of dense-core vesicles (DCVs) and synaptic vesicles along axonal microtubules (Chiba et al, 2023; Gabrych et al, 2019; Okada et al, 1995). The second most altered isoform was ENSMUST00000018122 (*Cadps2-201*) (Fig. 4A,B). *Cadps2* encodes the CAPS2 protein, which binds to DCVs on the cytoplasmic side, plays an indispensable role in their secretion (Sadakata et al, 2007b) and co-localizes with AgRP (Fujima et al, 2020).

The *Kif1a-201* isoform includes exon 13 (Fig. EV7A). The Sashimi plot, which shows read coverage across a splice junction, showed that *Kif1a* exon 13 inclusion was lower in *Fto*[lox/lox]/*Tau*-Cre mice than in control (*Fto*[lox/lox]) mice (Fig. 4C). The percent spliced-in (PSI) value for *Kif1a* exon 13 was lower in *Fto*[lox/lox]/*Tau*-Cre mice than in control mice (Fig. 4D). *Kif1a* exon 13 encodes a hinge domain (Huo et al, 2012) (Fig. EV7A), and an identical domain in KIF1B, another member of the kinesin-3 family, enhances ATPase activity (Matsushita et al, 2009). Analysis of *Kif1a* isoforms revealed that isoforms containing exon 13 (*Kif1a-201*/ENSMUST00000086819 and *Kif1a-210*/ENSMUST00000190723) had a lower expression level in *Fto*[lox/lox]/*Tau*-Cre mice, while an isoform without exon 13 (*Kif1a-203*/ENSMUST00000171556) had a higher expression level (Fig. EV7B). These data suggest that FTO plays an indispensable role in the inclusion of *Kif1a* exon 13. KIF1A levels remained unchanged in *Fto*[lox/lox]/*Tau*-Cre mice (Fig. EV7C,D), suggesting that FTO regulates alternative splicing but not the translation of *Kif1a*.

*Cadps2-201* (CAPS2b) is a *Cadps2* transcript containing exon 22 (Sadakata et al, 2007a). The Sashimi plot showed that the inclusion of *Cadps2* exon 22 was lower in *Fto*[lox/lox]/*Tau*-Cre mice than in control mice (Fig. 4E). The PSI value for *Cadps2* exon 22 was lower in *Fto*[lox/lox]/*Tau*-Cre mice than in control mice (Fig. 4F). CAPS2b is the most potent CAPS2 isoform in enhancing brain-derived neurotrophic factor (BDNF) secretion (Sadakata et al, 2007a). The FTO-induced inclusion of *Cadps2* exon 22 may contribute to its participation in the upregulation of NPY/AgRP secretion. Genes related to membrane trafficking, including *Kif1a*, *Cadps2*, *Entpd5*, *Dbn1* (Fang et al, 2010; Shirao et al, 2017), were highly enriched among the 15 most altered isoforms (Fig. 4B). These data indicate that FTO influences the expression of membrane trafficking proteins. We also observed that the isoform profiles of some genes not targeted by FTO were also altered in *Fto*[lox/lox]/*Tau*-Cre mice (Fig. 4A), likely due to the modulation of splicing factors such as RBFOX1 and RBFOX2, which are targeted by FTO.

## FTO enhances the axonal transport of DCVs containing NPY and AgRP

Global analysis of m⁶A RNA, other RNA, and protein isoforms revealed that the most significant changes were associated with genes related to membrane trafficking, including *Kif1a*, *Rims1*, and *Cadps2*, in *Fto*[lox/lox]/*Tau*-Cre mice and/or in *Fto*[lox/lox]/*Agrp*-Cre mice, leading us to hypothesize that FTO promotes axonal transport and/or neurotransmitter release. The number of cell bodies and neurite length, assessed based on NPY-hrGFP fluorescence throughout AgRP neurons, were not altered in *Fto*[lox/lox]/*Agrp*-Cre mice (Fig. 5A–C). The density of NPY vesicles in the axons of AgRP neurons, visualized using NPY immunohistochemistry (Ramamoorthy et al, 2011) was significantly lower in the PVH and ARC of *Fto*[lox/lox]/*Agrp*-Cre mice than those of control mice (Figs. 5D,E and EV8A,B). DCVs, which contain neuropeptides, including NPY and AgRP (Ramamoorthy et al, 2011; van den Pol, 2012), stained with an anti-secretogranin II antibody almost overlapped exclusively with NPY-hrGFP in the PVH and ARC of *Fto*[lox/lox]/*Npy*-hrGFP mice (Figs. 5F and EV8C), implying that they are predominantly distributed in NPY/AgRP neurons in these areas. The density of DCVs was decreased in AgRP neurons in the PVH and ARC of *Fto*[lox/lox]/*Agrp*-Cre mice (Figs. 5F,G and EV8C,D), and increased in those of *Fto*[AgRP]-overexpressing mice (Fig. EV8E–H) compared to controls, suggesting that FTO plays a critical role in promoting DCV transport in AgRP neurons. Electron microscopy revealed a lower number of DCVs in *Fto*[lox/lox]/*Agrp*-Cre mice and, correspondingly, a higher number of DCVs in *Fto*[AgRP]-overexpressing mice than in control mice (Fig. 5H–K), indicating that FTO enhanced DCV transport in AgRP neurons. Consistent with these results, AgRP release from brain slices after high-potassium stimulation was decreased in *Fto*[lox/lox]/*Agrp*-Cre mice and increased in *Fto*[AgRP] overexpressing mice (Fig. 5L,M). Similarly, AgRP release induced by low glucose and glutamate, which are physiological stimulants of AgRP neuron activity (Liu et al, 2012; Yoon and Diano, 2021), was decreased in *Fto*[lox/lox]/*Agrp*-Cre mice and increased in *Fto*[AgRP] overexpressing mice (Fig. EV8I,J).

Gamma-aminobutyric acid (GABA) is another important neurotransmitter released by AgRP neurons (Tong et al, 2008) and is stored in synaptic vesicles, which are potentially regulated by KIF1A (Chiba et al, 2023). Vesicular GABA transporter (VGAT), a marker of synaptic vesicles, is localized on these vesicles. VGAT density in NPY fibers in the PVH was unchanged in *Fto*[lox/lox]/*Agrp*-Cre and *Fto*[AgRP] overexpressing

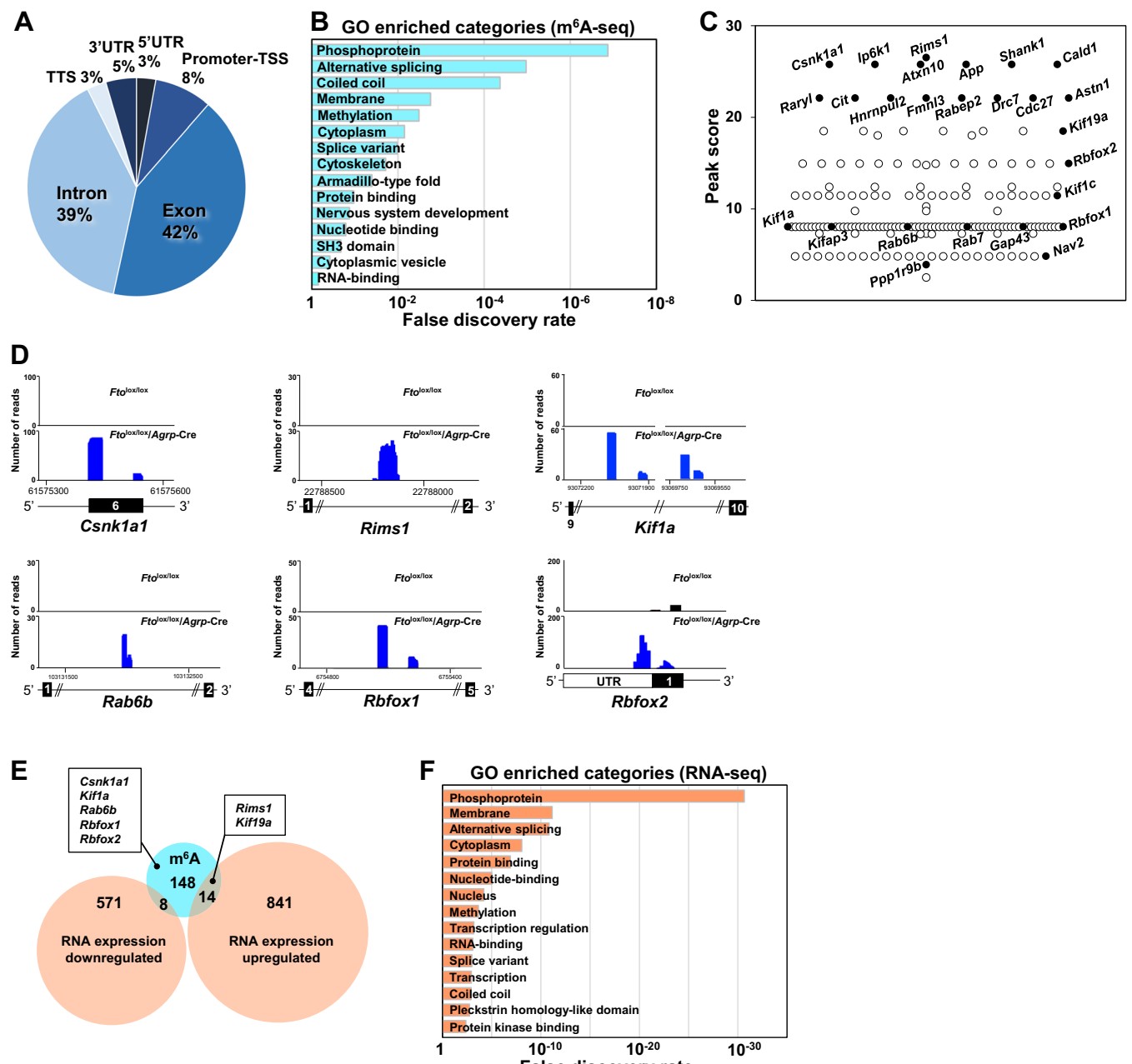

**Figure 3. FTO demethylates m⁶A modifications in exons and introns of genes encoding for proteins involved in alternative splicing and trafficking.**

FTO-dependent demethylation of m⁶A in AgRP neurons was analyzed by m⁶A immunoprecipitation followed by m⁶A-seq analysis of ARC tissues from $Fto^{lox/lox}$ and $Fto^{lox/lox}$/$Agrp$-Cre mice. (A) The distribution of FTO-dependent m⁶A demethylation in six non-overlapping transcript segments is presented as a percentage of the peak number. (B) The top 15 gene ontology categories associated with FTO-dependent demethylation of m⁶A are shown. (C) Distribution of the maximum score of FTO-dependent m⁶A demethylation enriched to show trafficking- and alternative splicing-related genes. (D) Representative m⁶A-seq traces for $Csnk1a1$, $Rims1$, $Kif1a$, $Rab6b$, $Rbfox1$, and $Rbfox2$. (E) Overlap of FTO-demethylated genes and genes with upregulated or downregulated expression in the ARC of $Fto^{lox/lox}$/$Agrp$-Cre mice compared to $Fto^{lox/lox}$ mice. The numbers indicate the number of genes. (F) The top 15 gene ontology categories in which RNA expression levels were altered in the ARC of $Fto^{lox/lox}$/$Agrp$-Cre mice compared to $Fto^{lox/lox}$ mice. Source data are available online for this figure.

mice compared to that in control mice (Fig. EV9), implying that FTO and its downstream pathways do not play a critical role in synaptic vesicle transport.

These data suggested that FTO strongly enhanced the release of DCV-stored neurotransmitters from AgRP neurons. As neurotransmitter release is often dependent on neuronal activity, we

investigated this aspect and found that the action potential and membrane potential levels in AgRP neurons in $Fto^{lox/lox}$/$Agrp$-Cre mice were comparable to those in control mice (Fig. EV10A–D). In addition, the intracellular $Ca^{2+}$ increase in AgRP/NPY neurons in response to ghrelin, a hormone that activates AgRP neuronal activity (Kohno et al, 2003), was comparable between $Fto^{lox/lox}$ and

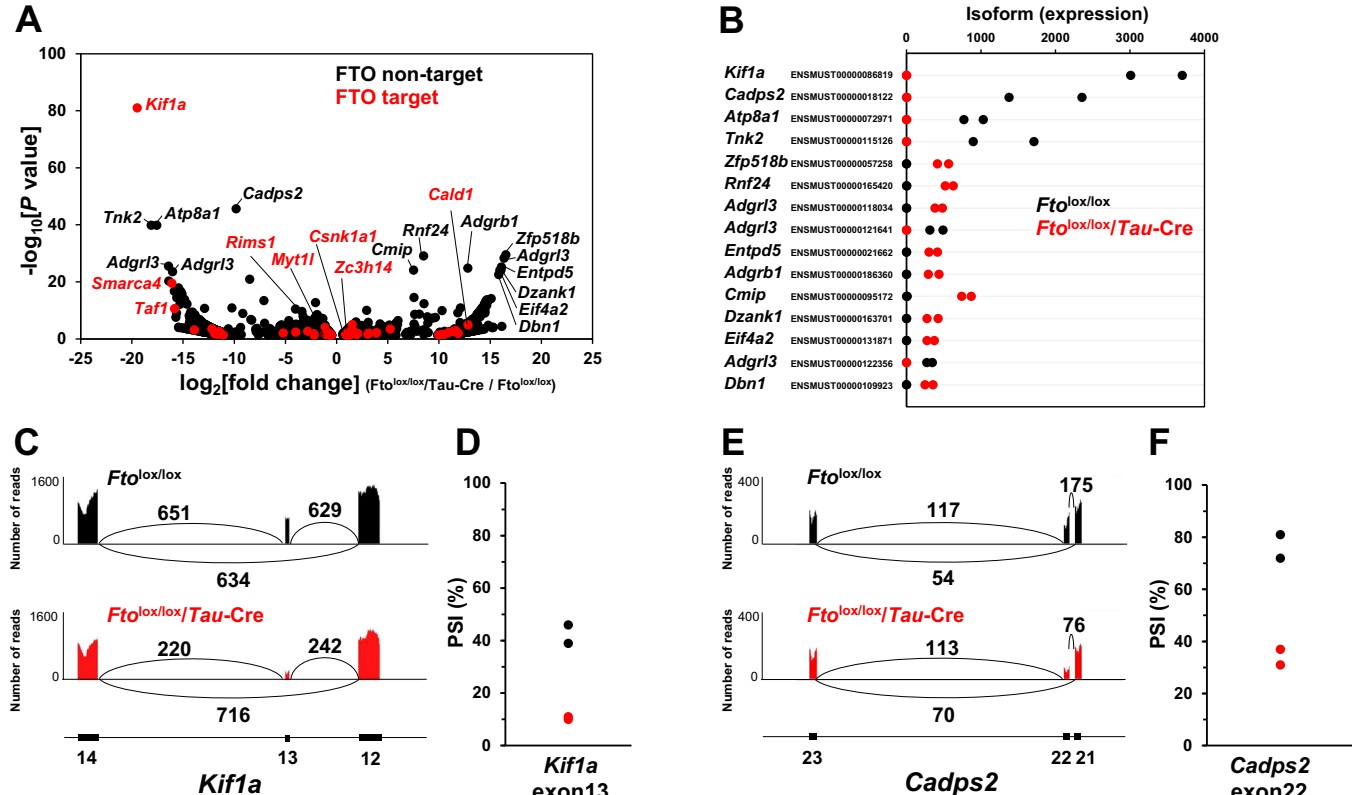

**Figure 4. FTO affects the alternative splicing of *Kif1a* and *Cadps2*.**

Transcriptome-wide isoform analysis was performed using hypothalamus tissue samples from $Fto^{lox/lox}$ and $Fto^{lox/lox}/Tau$-Cre mice. (A) Volcano plot showing fold changes in isoform expression and $P$ values for $Fto^{lox/lox}/Tau$-Cre compared to $Fto^{lox/lox}$ mice ($n = 2$ pooled samples per group; each sample pooled from three mice). Genes for which m⁶A was demethylated by FTO, as shown in Fig. 3C (FTO target), are indicated in red, whereas non-target genes (FTO non-target) are indicated in black. (B) The top 15 isoforms with the highest changes in expression levels are listed. Gene symbols and Ensembl transcript IDs are used to identify individual isoforms. (C–F) Rates of alternative splicing events represented using Sashimi plots (C, E) and PSI values calculated using SplAdder (D, F) for *Kif1a* exon 13 (C, D) and *Cadps2* exon 22 (E, F). Source data are available online for this figure.

$Fto^{lox/lox}/Agrp$-Cre mice (Fig. EV10E–G), and c-Fos expression in AgRP/NPY neurons after overnight fasting was comparable between these mice (Fig. EV10H,I). These data suggest that FTO is not necessary for the normal control of neuronal activity and immediate transcriptional response, and that FTO enhances neurotransmitter release from AgRP neurons through mechanisms other than changes in neuronal excitability.

## Alternative splicing of *Kif1a* exon 13 is predicted to alter KIF1A conformation

We speculated that with regard to alternative splicing, *Kif1a* is particularly relevant to the primary effect of FTO in AgRP neurons, that is, enhanced transport and secretion of DCVs. To investigate the functional relevance of the alternative splicing of exon 13 of *Kif1a*, we first predicted the protein structure of KIF1A after both skipping and including exon 13 using AlphaFold2 (Jumper et al, 2021). A loop structure, called the hinge region, was predicted to consist of five amino acids in the exon 13 skipped form (Fig. 6A); in contrast, the exon 13 inclusion form was predicted to contain 22 amino acids, which is much longer than that in the skipped form, owing to the replacement of an alpha-helix structure with a loop structure and the

inclusion of 10 amino acids translated from exon 13 (Fig. 6A). Furthermore, the C-terminal side of the neck coil (NC), starting from Leu388, was predicted to be shortened from 13 amino acids in the skipped form to five amino acids in the inclusion form (Fig. 6A). These changes also affected the distance between the C-terminal side of NC and the coiled-coil 1a (CC1a). The distances between the two alpha structures of the NC and CC1a in the skipped form (14.7 Å, and 12.3 Å) increased to 33.5 Å and 26.8 Å in the inclusion form. (Fig. 6B,E). Furthermore, Arg423 and Gly387 in the skipped form were predicted to be bound, whereas Arg432 and Gly387 in the homologous regions of the inclusion form were predicted to be unbound (Fig. 6C,D,F,G). These predictions suggest that alternative splicing of *Kif1a* exon 13 affects the conformation of the KIF1A hinge region and the distance between the NC and CC1a.

## The inclusion of *Kif1a* exon 13 enhances KIF1A dimerization and activity

The hinge region of kinesin-3 has been suggested to play a role in disrupting the binding between NC and CC1a, thereby preventing the formation of monomers instead of dimers (Al-Bassam et al, 2003; Ren et al, 2018; Wang et al, 2022). To analyze the effect of

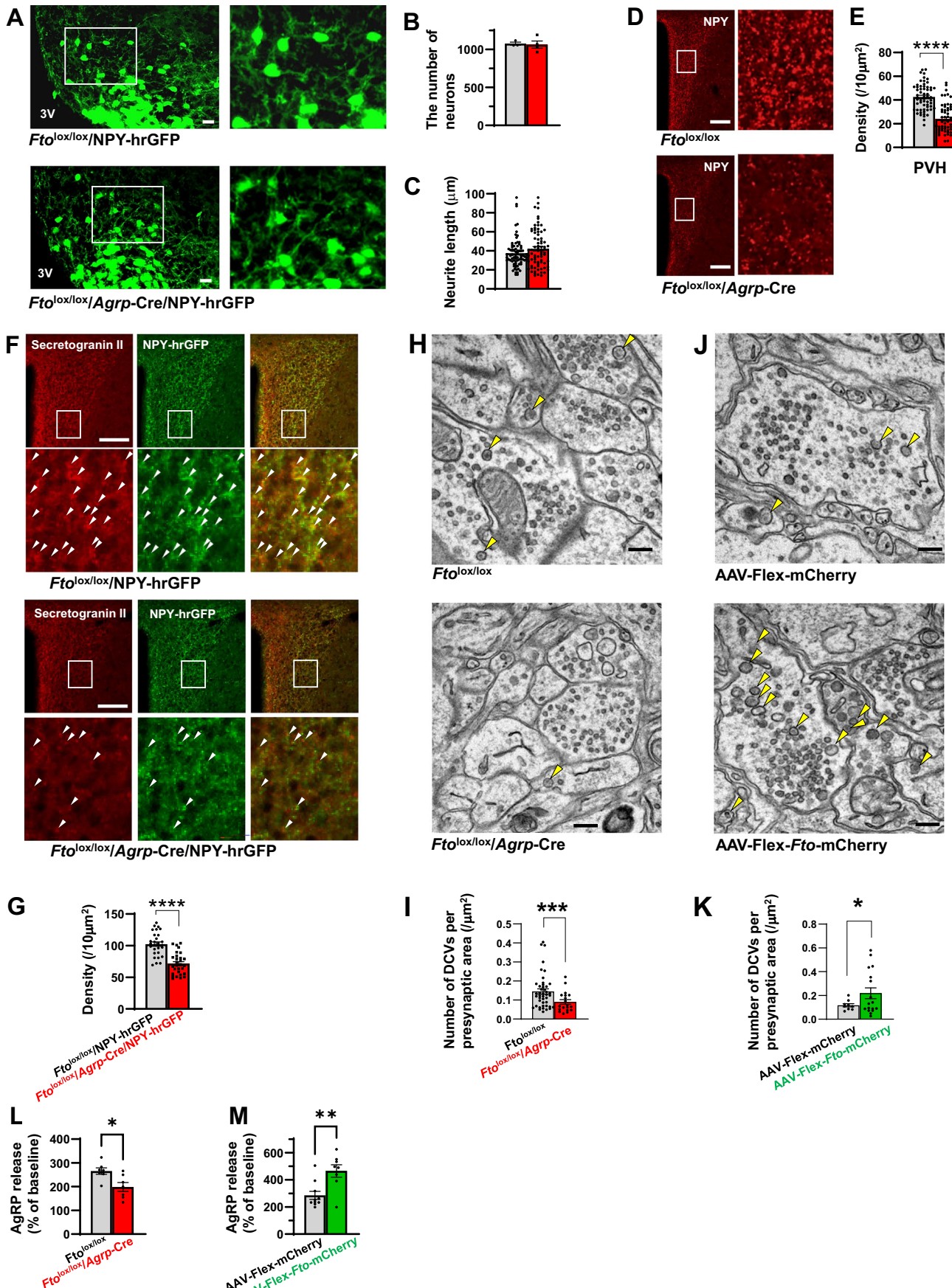

**Figure 5. FTO enhances the axonal transport of DCVs containing NPY and AgRP.**

(A) NPY-hrGFP fluorescence was used to visualize the cell bodies and neurites of NPY/AgRP neurons. Scale bars: 20 μm. (B, C) The number of NPY-hrGFP neurons in the ARC (B) and the length of NPY-hrGFP neurites (C) in $Fto^{lox/lox}$/NPY-hrGFP (gray) ($n = 3$ (B), 74 (C)) and $Fto^{lox/lox}$/$Agrp$-Cre/NPY-hrGFP ($n = 4$ (B), 75 (C)) (red) mice. Error bars represent SEM. Data were analyzed using unpaired Welch's $t$ test. (D, E) NPY immunofluorescence in the PVH of $Fto^{lox/lox}$ and $Fto^{lox/lox}$/$Agrp$-Cre mice (D) and density of NPY immunofluorescence in the PVH of $Fto^{lox/lox}$ (gray, $n = 60$) and $Fto^{lox/lox}$/$Agrp$-Cre (red, $n = 60$) mice (E). Error bars represent SEM. Data were analyzed using unpaired Student's $t$ test, ****$P < 0.001$ (exact $P$ value: $1.26 \times 10^{-14}$). Scale bars: 100 μm. (F, G) Immunofluorescence analysis of secretogranin II, a DCV marker (red), and NPY-hrGFP fluorescence (green) in the PVH (F). Arrowheads indicate secretogranin II and NPY-hrGFP immunofluorescence overlap. Density of secretogranin II immunofluorescence in the PVH of $Fto^{lox/lox}$/NPY-hrGFP (gray, $n = 29$) and $Fto^{lox/lox}$/$Agrp$-Cre/NPY-hrGFP (red, $n = 29$) mice (G). Error bars represent SEM. Data were analyzed using unpaired Student's $t$ test, ****$P < 0.001$ (exact $P$ value: $1.85 \times 10^{-8}$). Scale bars: 100 μm. (H–K) Electron microscopy analysis of DCVs (arrowheads) in $Fto^{lox/lox}$ and $Fto^{lox/lox}$/$Agrp$-Cre mice (H) and $Agrp$-Ires-Cre mice injected with AAV-hSyn-Flex-mCherry and AAV-hSyn-Flex-$Fto$-mCherry (J). Number of DCVs per presynaptic area in $Fto^{lox/lox}$ (gray, $n = 44$) and $Fto^{lox/lox}$/$Agrp$-Cre (red, $n = 19$) mice (I). Number of DCVs per presynaptic area of $Agrp$-Ires-Cre mice injected with AAV-hSyn-Flex-mCherry (gray, $n = 9$) and AAV-hSyn-Flex-$Fto$-mCherry (red, $n = 17$) (K). Error bars represent SEM. Data were analyzed using unpaired Welch's $t$ test, *$P < 0.05$, ***$P < 0.005$ (exact $P$ values: 0.003 (I), 0.044 (K)). Scale bars: 1 μm. (L, M) AgRP release in brain slices of $Fto^{lox/lox}$ (gray, $n = 7$) and $Fto^{lox/lox}$/$Agrp$-Cre (red, $n = 7$) mice (L) and $Agrp$-Ires-Cre mice injected with AAV-hSyn-Flex-mCherry (gray, $n = 10$) and AAV-hSyn-Flex-$Fto$-mCherry (red, $n = 8$) (M). Error bars represent SEM. Data were analyzed using unpaired Student's $t$ test, *$P < 0.05$, **$P < 0.01$ (exact $P$ values: 0.013 (I), 0.003 (K)). Source data are available online for this figure.

exon 13 inclusion on KIF1A dimerization, we performed size-exclusion chromatography using the recombinant KIF1A protein with exon 13 skipped or included (Figs. 7A,B and EV11). The KIF1A skipping form retention volume peaked at 13.98 mL (90 kDa), which is intermediate in size between dimeric (111 kDa) and monomeric (55.5 kDa) KIF1A. In contrast, the inclusion form peaked at 13.58 mL (109 kDa), which is closer to the size of the dimeric KIF1A, suggesting that the inclusion form is predominantly dimerized, whereas the skipping form exists as both monomeric and dimeric forms. As dimeric kinesin-3 undergoes ATP-dependent processive motility (Hammond et al, 2009; Soppina et al, 2014; Tomishige et al, 2002), we hypothesized that the inclusion of exon 13 would increase the microtubule-dependent ATPase activity of KIF1A. As expected, microtubule-dependent ATPase activity was higher in the inclusion form of exon 13 than in the skipped form (Fig. 7C,D).

We then performed a microtubule gliding assay to examine whether increased ATPase activity affected motor velocity. In this assay, the skipped form of KIF1A-203 (1-473)-His or the inclusion form of KIF1A-201(1-482)-His (Fig. 7A), were immobilized on the coverslip surface using an anti-His tag antibody. Taxol-stabilized microtubules, visualized using Alexa Fluor 647, were then glided by the exposed motor domains. The gliding velocity of microtubules driven by the inclusion form was significantly higher than that driven by the skipped form (Fig. 7E–G).

To confirm the results of the in vitro velocity assay at the cellular level, we examined the dynamics of axonal KIF1A-EGFP overexpression in primary neurons. The velocity of full-length KIF1A-EGFP was significantly higher in the inclusion vs. skipping form (Fig. 7H–J). These results suggested that the inclusion of exon 13 increases the activity of ATPase, thereby enhancing KIF1A velocity. To assess the effects of KIF1A isoforms on cell function, we used differentiated PC12 cells that stably overexpressed $Kif1a$ with either exon 13 skipping or inclusion. After high-potassium treatment, the amount of endogenous NPY was higher in cells overexpressing the inclusion form than in those overexpressing the skipping form (Fig. 7K,L). Furthermore, the number of NPY vesicles in the fibers was significantly higher in cells overexpressing the inclusion form than in cells overexpressing the skipped form (Fig. 7M,N), suggesting that exon 13 inclusion enhances the neurosecretory function, at least in differentiated PC12 cells.

## The FTO-KIF1A pathway in AgRP neurons is physiologically relevant

The results of prior experiments suggested that FTO expression in AgRP neurons affects body weight. We also examined whether the expression of $Fto$ in AgRP neurons varied with feeding conditions. Using short hairpin hybridization chain reaction (shHCR) in situ hybridization for $Fto$ and $Agrp$, we found that $Fto$ mRNA expression levels in AgRP neurons increased significantly after overnight fasting compared to ad libitum feeding mice and persisted even after 3 h of refeeding (Fig. 8A,B). This suggests that FTO in AgRP neurons is upregulated under fasting and refeeding conditions.

We also examined whether alternative splicing of $Kif1a$ exon 13, is influenced by feeding conditions. Isoform-specific PCR using ARC cDNA revealed that exon 13 skipping was predominant under ad libitum feeding conditions; exon 13 inclusion percentage increased significantly in mice fasted overnight, and this increase continued after 3 h of refeeding (Fig. 8C,D). Consistently, digital PCR revealed that exon 13-containing $Kif1a$ mRNA copy number relative to total $Kif1a$ mRNA copy number increased in the ARC of mice fasted overnight (Fig. 8E). Similarly, secretogranin II distribution in the PVH was significantly increased in mice fasted overnight, and this increase remained significant even after 3 h of refeeding (Fig. 8F,G). These data suggest that the FTO-KIF1A pathway and subsequent DCV transport are upregulated in AgRP neurons during fasting and refeeding.

To determine whether KIF1A is indispensable for FTO-induced weight gain, we knocked down $Kif1a$ in AgRP neurons while overexpressing $Fto$ by injecting both AAV-U6-$Kif1a$sgRNA-hSyn-Flex-mCherry and AAV-hSyn-Flex-$Fto$-mCherry into Rosa26-LSL-Cas9 knock-in/$Agrp$-Ires-Cre mice (Fig. EV12). Knockdown of AgRP neuron-specific $Kif1a$ with $Fto$ overexpression did not result in a significant increase in body weight compared to that in control mice that did not overexpress $Fto$, whereas AgRP neuron-specific $Fto$ overexpression mice exhibited significantly higher body weights compared to control mice (Fig. 8H), suggesting that KIF1A partially mediates FTO-induced weight gain. These data indicated that the FTO-KIF1A pathway in AgRP neurons is upregulated during fasting and contributes to weight gain.

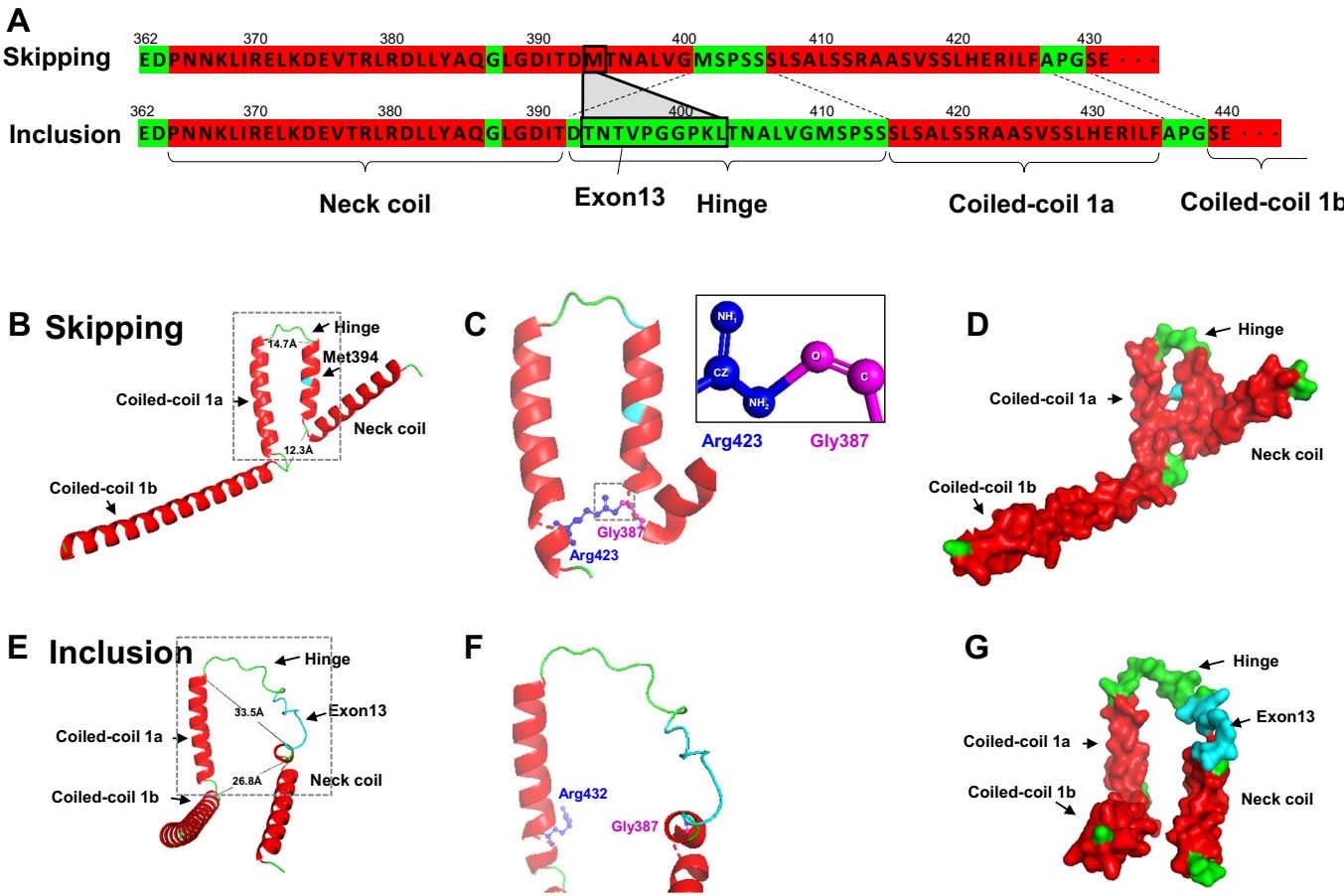

**Figure 6. Alternative splicing of *Kif1a* exon 13 is predicted to alter the conformation of KIF1A.**

(A) AlphaFold2 prediction of protein structures of the neck coil, hinge, and coiled-coil regions of the KIF1A isoform with exon 13 skipped or included. Alpha-helix (red), loop (green). The molecular graphic structures of the exon 13 skipped (B–D) and included (E–G) variants are depicted as ribbon diagrams (B, C, E, F), combined ribbon and stick diagrams (C, F), and surface representations (D, G). Alpha-helices, loops, and exon 13 are indicated in red, green, and light blue, respectively. Distance between Gly400 and Ser406: 14.7 Å, between Gly387 and Pro428: 12.3 Å (B), between Thr392 and Ser415: 33.5 Å, between Gly387 and Pro437: 26.8 Å.

## Discussion

In this study, we demonstrated that FTO in AgRP neurons promotes weight gain by enhancing DCV axonal transport. At the molecular level, FTO demethylates m⁶A in the exons and introns of pre-mRNAs encoding proteins involved in membrane trafficking and alternative splicing. Among other effects, FTO-induced m⁶A demethylation stimulated the inclusion of *Kif1a* exon 13, which enhances KIF1A dimerization and function. As the FTO-KIF1A pathway is upregulated during fasting conditions and contributes to weight gain, we conducted further investigations and identified epitranscriptional regulation and alternative splicing in AgRP neurons as key regulatory processes for KIF1A function and ultimately controlling feeding and metabolism (Fig. EV13). Thus, our findings highlight the importance of axonal DCV transport in AgRP neurons for weight gain.

Phenotypic analyses of mouse models lacking or overexpressing FTO as well as of plant models in which FTO is introduced exogenously have shown that this protein universally affects body weight and size (Church et al, 2010; Fischer et al, 2009; Yu et al,

2021), but the specific targets and mechanisms involved remain unclear. In this study, we found that FTO is expressed in AgRP neurons and plays a role in regulating body weight. The reduction in the body weight of mice with AgRP neuron-specific *Fto* deletion was lower than that in mice with global or neuron-specific *Fto* deletion (Fischer et al, 2009; Gao et al, 2010). Notably, the more pronounced body weight phenotype in the latter two models was against the background of the general growth delay phenotype, which was not observed in our AgRP neuron-specific *Fto* deletion mice. Specific deletion or overexpression of *Fto* in AgRP neurons had little effect on linear growth, suggesting that FTO influences growth by acting on neurons other than AgRP neurons. Regarding control of body weight, our results suggest that AgRP neurons play an exclusive role among all main hypothalamic feeding neurons. FTO was abundantly expressed in AgRP neurons, which likely explains its strong effect in these cells compared with those in other hypothalamic feeding neurons (Fig. EV1).

The METTL3/14 complex is the major RNA m⁶A methylase, in this study, deletion of *Mettl3* in AgRP neurons did not alter body weight. On the other hand, in addition to FTO, ALKBH5 is another

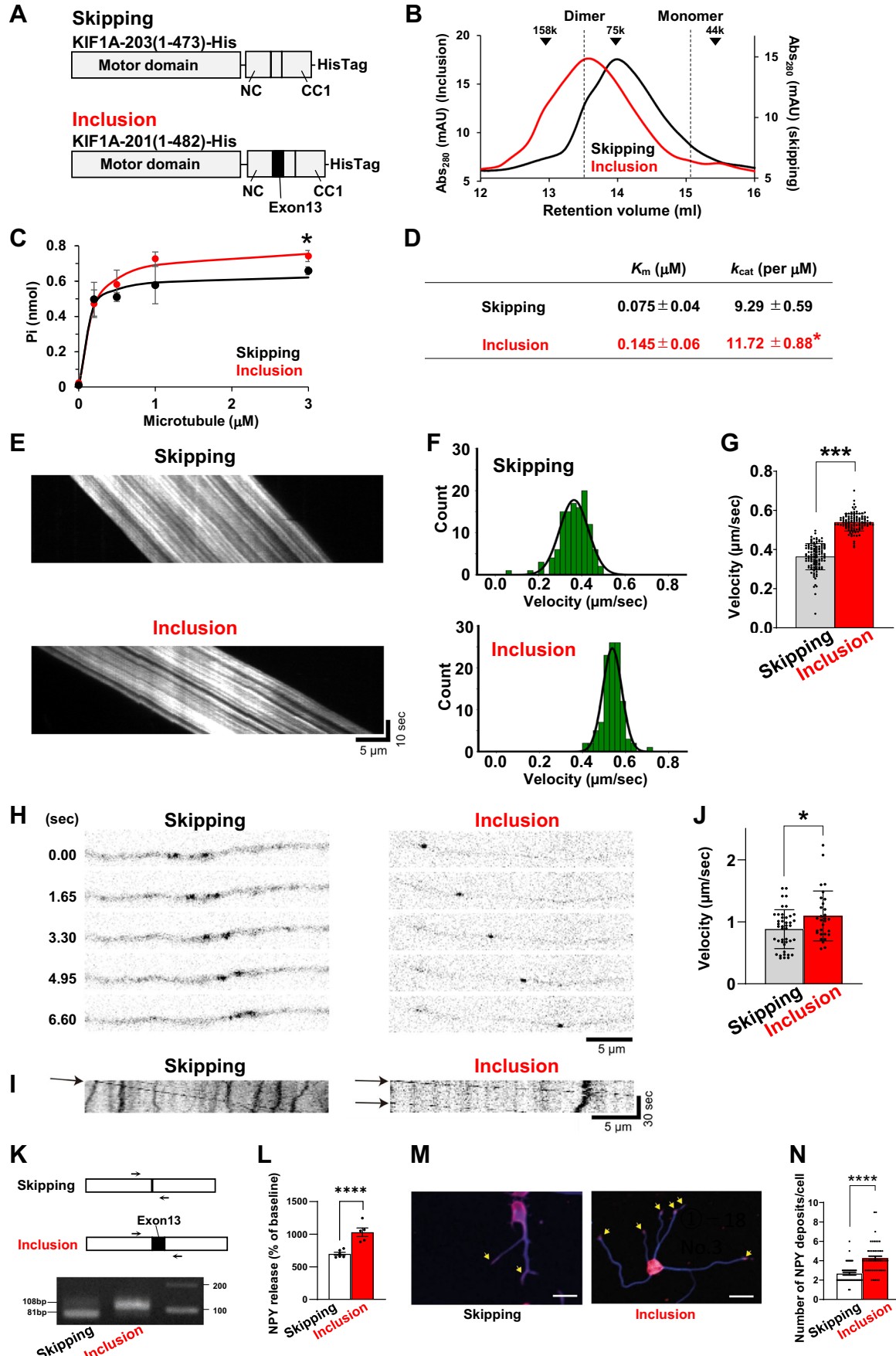

**Figure 7. Inclusion of *Kif1a* exon 13 enhances KIF1A dimerization and activity.**

(A) Schematics of recombinant KIF1A exon 13 skipped or included proteins. (B) Size-exclusion chromatography results for recombinant KIF1A proteins. Arrowheads indicate standard protein peaks. Dotted lines indicate expected retention volumes of dimeric (111 kDa) and monomeric (55.5 kDa) KIF1A. (C) Results of microtubule-dependent ATPase activity assay for recombinant proteins by measuring the Pi concentration generated by ATP hydrolysis. Data represent mean ± SD values ($n = 3$). Data were analyzed using unpaired Student's $t$ test; *$P < 0.05$ (exact $P$ value: 0.019). (D) Microtubule-dependent ATPase activity assay $K_m$ and $k_{cat}$ values. Data represent mean ± SD values ($n = 3$). *$P < 0.05$. Data were analyzed using unpaired Student's $t$ test; *$P < 0.05$ (exact $P$ value: 0.016). (E) Representative taxol-stabilized microtubule kymographs with 20% AF647 labeling showing the movement driven by each KIF1A isoform. Scale bar: 5 µm horizontally, 10 s vertically. (F) Histograms of the velocities of each KIF1A isoform obtained using the fitted gliding assay, with normal distribution curves. $n = 119$ for skipped; $n = 110$ for included. (G) Bar plots of KIF1A velocity (mean ± SD) obtained using the gliding assay. Data were analyzed using unpaired Welch's $t$ test; ***$P < 0.001$ (exact $P$ value: $2.07 \times 10^{-60}$). $n = 119$ for skipped; $n = 110$ for included. (H) Full-length EGFP-tagged exon 13 skipped or included KIF1A proteins were expressed in primary hippocampal neurons. Representative time-lapse imaging of KIF1A traveling anterogradely along the axon. Scale bar: 5 µm. (I) Representative kymograph for each KIF1A isoform. Arrows indicate typical KIF1A traveling anterogradely. Scale bar: 5 µm horizontally, 30 s vertically. (J) Velocity of exon 13 skipped ($n = 26$) and included ($n = 30$) KIF1A traveling anterogradely in the axon. Data represent mean ± SD values. Data were analyzed using unpaired Welch's $t$ test; *$P < 0.05$ (exact $P$ value: 0.018). (K) PC12 cells that stably expressed full-length KIF1A exon 13 skipped or included were obtained, and the expression of the *Kif1a* isoforms was confirmed by reverse transcription (RT)-PCR using the indicated primers. (L) High-potassium-induced NPY release from PC12 cells. $n = 6$ for skipped; $n = 5$ for included. Error bars represent SEM. Data were analyzed using unpaired Student's $t$ test; ****$P < 0.001$ (exact $P$ value: 0.0006). (M, N) Differentiated PC12 cells stably overexpressing *Kif1a* isoforms were immunocytochemically stained using an anti-NPY antibody (red) and the neuronal marker β-tubulin III (blue) (M). Arrows indicate NPY vesicles. Scale bar: 50 µm. N number of NPY deposits in neurites/cells ($n = 50$ for each group). Error bars represent SEM. Data were analyzed using unpaired Student's $t$ test; ****$P < 0.001$ (exact $P$ value: $4.52 \times 10^{-8}$). Source data are available online for this figure.

demethylase. The m⁶A sites, which are FTO substrates, may be different from those demethylated by ALKBH5, and it is possible that demethylation of FTO substrates only induces a gain in body weight. Another reason for the different effects of *Mettl3* and *Fto* expression manipulations is that the METTL3/14 complex also methylates DNA (Woodcock et al, 2019), whereas FTO demethylates m⁶A only in nuclear RNA (Jia et al, 2011). Differences in the substrates may be another reason for this discrepancy.

Based on our finding that FTO demethylation occurred in exons and introns but not in the 5′ UTR regions of pre-mRNA, it was suggested that the main substrates of FTO in AgRP neurons are m⁶A, but not m⁶Am epitranscriptomic marks. Although it has been reported that FTO preferentially demethylates m⁶Am over m⁶A in vitro and in some cell types (Mauer et al, 2017), this was not observed in AgRP neurons. Consistent with previous studies (Bartosovic et al, 2017), m⁶A in the exonic and intronic regions was associated with alternative splicing in our study. FTO-demethylated mRNAs were enriched for genes classified as "alternative splicing" genes by DAVID ontology analysis. This category includes genes that control alternative splicing, such as *Rbfox1* and *Rbfox2*, and genes with splice variants. Our data suggest that FTO affects alternative splicing by controlling splicing factors in addition to altering m⁶A patterns in mRNAs that undergo alternative splicing. Small nuclear RNAs (snRNAs) can also be demethylated by FTO to mediate alternative splicing (Mauer et al, 2019). Our m⁶A-seq method was not optimized for snRNAs, and we were unable to analyze snRNA demethylation. Further studies are required to determine whether snRNAs mediate the effects of FTO in AgRP neurons. Since *Kif1a* is a direct target of FTO demethylation, the alternative splicing of *Kif1a* exon 13 could be directly regulated by m⁶A modifications. Our m⁶A-seq analysis detected FTO-induced m⁶A demethylation sites only in the distal region relative to *Kif1a* exon 13 (Fig. 3D). Alternative splicing of *Kif1a* exon 13 could be regulated by distal regulatory mechanisms. Alternatively, because several m⁶A modification sites were predicted in the introns adjacent to *Kif1a* exon 13 according to the sequence-based m⁶A modification site predictor (Fan et al, 2024; Zhou et al, 2016), demethylation of proximal m⁶A modification sites near exon 13, which were not detected by our m⁶A-seq,

possibly because of its limited sensitivity, may contribute to the upregulation of *Kif1a* exon 13 inclusion.

In this study, we also showed that FTO enhances DCV trafficking in AgRP neurons. Numerous studies have been conducted on NPY-containing DCVs in adrenal medullary PC12 cells, but DCVs in hypothalamic NPY/AgRP neurons have not been intensively studied. According to the results of our immunohistochemical analysis using an anti-secretogranin II antibody (Fig. 5F), DCVs were highly and preferentially colocalized with NPY/AgRP fibers in the ARC. AgRP neuron-specific *Fto*-knockout mice had significantly lower secretogranin II levels, suggesting that DCVs in the ARC are predominantly distributed in NPY/AgRP neurons. Although secretogranin II is also present in POMC neurons (Hotta et al, 2009), DCV may be less widely distributed in POMC neurons than in NPY/AgRP neurons. This may be because nerve fibers immunopositive for NPY are denser than those immunopositive for POMC in the ARC and PVH (Marraudino et al, 2021). These results suggest that DCVs are predominantly found in NPY/AgRP neurons. Furthermore, mice lacking *Cadps2* exhibited lower body weights than control animals (Mishima et al, 2015; Sadakata et al, 2007b). These data support the notion that DCVs are the most abundant in NPY/AgRP neurons. Next-generation sequencing analysis of isoforms in mice with neuron-specific *Fto* deletion revealed that the alternative splicing of *Kif1a* and *Cadps2*, which are highly associated with DCV trafficking, were altered to the greatest extent (Fig. 4), suggesting that FTO controls DCV trafficking. This may explain why the deletion of *Fto* in AgRP neurons, but not in POMC neurons, resulted in phenotypic differences, although we were unable to perform AgRP neuron-specific alternative splicing analysis due to technical limitations. According to the VGAT immunohistochemistry results, the synaptic vesicles are not affected by FTO. The transport of synaptic vesicles in AgRP neurons may be primarily mediated by other motor proteins, such as KIF1B and KIF5, rather than by KIF1A.

Deletion or overexpression of *Fto* in AgRP neurons altered both axonal transport and AgRP release. As enhanced axonal transport is presumed to facilitate the timely supply of vesicles required for secretion, increased AgRP release may be attributed to enhanced axonal transport. Additionally, FTO-demethylated genes related to

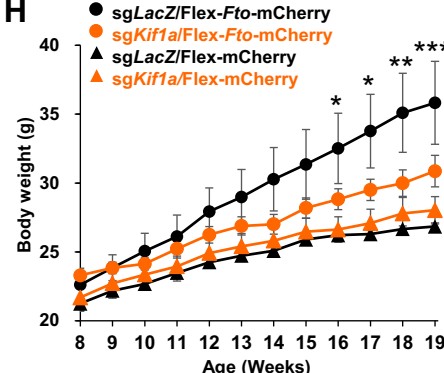

**Figure 8. The FTO-KIF1A pathway in AgRP neurons is physiologically relevant.**

(A) shHCR in situ hybridization for *Fto* (green) and *Agrp* (red) in the ARC of ad libitum fed, overnight-fasted, and 3 h refed C57BL/6J mice. Arrowheads indicate *Fto* in situ signals in AgRP neurons. Scale bar: 20 μm. (B) In situ hybridization in *Agrp*-expressing neurons was measured in 24–32 areas from three mice per group. Error bars represent SEM. Data were analyzed using one-way ANOVA followed by Tukey's multiple comparisons test; ****$P < 0.001$ (exact $P$ values: $1.55 \times 10^{-10}$ (ad lib vs. fasting), $1.62 \times 10^{-10}$ (ad lib vs. refeeding)). (C) *Kif1a* exon 13 skipping and inclusion were analyzed using RT-PCR analysis of ARC samples from ad libitum fed ($n = 4$), overnight-fasted ($n = 4$), and 3 h refed ($n = 3$) C57BL/6J mice. Each lane represents one mouse. (D) Band densities of the RT-PCR products were analyzed and expressed as PSI values. ad libitum fed ($n = 4$), overnight fasted ($n = 4$), and 3 h refed ($n = 3$). Error bars represent SEM. Data were analyzed using one-way ANOVA followed by Tukey's multiple comparisons test; **$P < 0.01$ (exact $p$ values: 0.001 (ad lib vs. fasting), 0.008 (ad lib vs. refeeding)). (E) mRNA copy numbers of total *Kif1a* and exon 13-including *Kif1a* were quantified using digital PCR from ARC samples of ad libitum fed ($n = 5$), overnight-fasted ($n = 5$), and 3 h refed ($n = 5$) C57BL/6J mice, and the percentage of exon 13-including *Kif1a* mRNA in total *Kif1a* mRNA was calculated. Error bars represent SEM. Data were analyzed using one-way ANOVA followed by Tukey's multiple comparisons test; *$P < 0.05$ (exact $P$ values: 0.036 (ad lib vs. fasting)). (F) Immunofluorescence analysis for secretogranin II (green), and NPY (red) in the PVH of ad libitum fed, overnight-fasted, and 3 h refed C57BL/6J mice. Arrowheads indicate secretogranin II immunofluorescence. Scale bar: 50 μm. (G) Density of secretogranin II immunofluorescence in the PVH area adjacent to NPY immunofluorescence (60–72 areas) from $n = 3$ mice per group. Error bars represent SEM. Data were analyzed using one-way ANOVA followed by Tukey's multiple comparisons test; *$P < 0.05$, ***$P < 0.0005$, ****$P < 0.0001$ (exact $P$ values: $1.13 \times 10^{-9}$ (ad lib vs. fasting), 0.0001 (ad lib vs. refeeding), 0.022 (fasting vs. refeeding)). (H) Body weight of Rosa26-LSL-Cas9 knock-in/*Agrp*-Ires-Cre mice injected with AAV-U6-*Kif1a*sgRNA-hSyn-Flex-mCherry or AAV-U6-LacZsgRNA-hSyn-Flex-mCherry and AAV-hSyn-Flex-*Fto*-mCherry or AAV-hSyn-Flex-mCherry to induce AgRP neuron-specific *Kif1a* knockdown and/or *Fto* overexpression. The body weight of mice injected with AAV-U6-sg*LacZ*/hSyn-Flex-*Fto*-mCherry ($n = 8$) was significantly higher than that of mice injected with AAV-U6-sg*LacZ*/hSyn-Flex-mCherry ($n = 7$) from 16 weeks of age, while the body weight of mice injected with U6-sg*Kif1a*/hSyn-Flex-*Fto*-mCherry ($n = 8$) was not significantly different from that of mice in the AAV-U6-sg*Kif1a*/hSyn-Flex-mCherry ($n = 7$) and other groups. Error bars represent SEM. Data were analyzed using one-way ANOVA followed by Tukey's multiple comparisons test; *$P < 0.05$, **$P < 0.01$ (exact $P$ values: 0.030 [16-week-old], 0.012 [17-week-old], 0.009 [18-week-old], 0.007 [19-week-old] (AAV-U6-sg*LacZ*/hSyn-Flex-*Fto*-mCherry vs. AAV-U6-sg*LacZ*/hSyn-Flex-mCherry)). Source data are available online for this figure.

neurotransmitter release, such as *Rims1*, could mediate the enhancement of AgRP release. Further studies are required to elucidate the detailed mechanisms underlying the regulation of AgRP release.

The inclusion rate of *Kif1a* exon 13 was associated with feeding conditions—it decreased under ad libitum conditions and increased during fasting. As the release of AgRP and NPY is thought to increase during fasting and is low under ad libitum conditions, these observations are consistent with the physiological requirements for DCV transport. The superprocessive movement of dimeric KIF1A (Soppina et al, 2014) and the slow motility of monomeric KIF1A (Okada et al, 1995) may serve as important regulators of NPY/AgRP neurons. AgRP and NPY vesicles are known to be preferentially distributed in neuronal fibers rather than in cell bodies, unless colchicine is injected intracerebroventricularly (Leger et al, 1987). In the hypothalamus, NPY and AgRP vesicles are localized in axons but not dendrites (Ramamoorthy et al, 2011). The significance of the abundant distribution of AgRP and NPY DCVs in axons is not yet understood, but this characteristic distribution pattern may be affected by the speed and pattern of axonal transport mediated by KIF1A. The actual rates of AgRP and NPY transport under physiological conditions remain to be clarified.

Interestingly, the high inclusion rate of exon 13 persisted even 3 h after refeeding, at which point feeding behavior was almost complete. One explanation for this phenomenon is that the effects of overnight fasting persist even after refeeding. A similar upregulation trend was also observed in the expression of *Fto* mRNA and secretogranin II in AgRP neurons. Strong hunger may cause prolonged activation of the FTO-KIF1A pathway, and the mechanisms underlying this phenomenon may be related to those involved in the regulation of *Fto* transcription. Metabolite levels during refeeding could also play a role, as the effects of FTO are influenced by its enzymatic activity, which in turn is affected by several metabolites, including nicotinamide adenine dinucleotide phosphate (NADP) (Wang et al, 2020).

In conclusion, the results of this study demonstrate that FTO-mediated epitranscriptional regulation in AgRP neurons alters the

alternative splicing of *Kif1a*, thus enhancing KIF1A activity, DCV transport, food intake, positive energy balance, and weight gain. These findings may enhance our understanding of the molecular mechanisms that control motor proteins and energy homeostasis.

## Methods

### Reagents and tools table

| Reagent/resource | Reference or source | Identifier or catalog number |
| --- | --- | --- |
| **Experimental models** | | |
| Fto<sup>tm1a(EUCOMM)Wtsi</sup> mouse | International Mouse Resource Center IMSR, EMMA | RRID: IMSR_EM:05094 |
| Agrp-Cre mouse | Xu et al, 2005 | MGI:3688399 |
| Agrp-Ires-Cre mouse | Tong et al, 2008 | RRID:IMSR_JAX:012899 |
| Sf1-Cre mouse | Dhillon et al, 2006 | RRID:IMSR_JAX:012462 |
| Sim1-Cre mouse | Balthasar et al, 2005 | RRID:IMSR_JAX:006395 |
| Pomc-Cre mouse | Balthasar et al, 2004 | RRID:IMSR_JAX:005965 |
| Tau-Cre mouse | Muramatsu et al, 2008 | NA |
| NPY-hrGFP mouse | van den Pol et al, 2009 | RRID:IMSR_JAX:006417 |
| tdTomato (Ai14) mouse | Madisen et al, 2010 | RRID:IMSR_JAX:007908 |
| Rosa26-LSL-Cas9 knock-in mouse | Platt et al, 2014 | RRID:IMSR_JAX:026175 |
| FLP knock-in mice | Farley et al, 2000 | RRID:IMSR_JAX:003946 |
| **Recombinant DNA** | | |
| Mouse Fto cDNA | Sino Biological | MG53159-G |
| pAAV-hSyn-DIO-hM4D (Gi)-mCherry | Addgene | Cat #44362 |
| pAAV-hSyn-DIO-mCherry | Addgene | Cat #50459 |

| Reagent/resource | Reference or source | Identifier or catalog number |
|---|---|---|
| **Antibodies** | | |
| Rabbit anti-FTO antibody | Proteintech | 27226-1-AP |
| Rabbit anti-GAPDH antibody | Santa Cruz | sc-25778 |
| Rabbit anti-KIF1A antibody | Sigma-Aldrich | SAB2104191 |
| Goat anti-rabbit IgG-HRP | Cayman Chemical | 10004301 |
| Rabbit anti-m$^6$A antibody | Abcam | ab151230 |
| Normal rabbit IgG | Cell signaling technology, | #2729 |
| Rabbit secretogranin II antibody | BIODESIGN | K55101R |
| Mouse secretogranin II antibody | Abcam | ab20245 |
| Rabbit VGAT antibody | GeneTex | GTX101908 |
| Rabbit anti-NPY antibody | Immunostar | 22940 |
| Mouse NPY antibody | Santa Cruz | sc-133080 |
| Rabbit anti-FTO antibody | LifeSpan Biosciences | LS-B7788 |
| Rabbit anti-POMC antibody | Phoenix Pharmaceuticals | H-029-30 |
| Rabbit anti-NUCB2 antibody | Sigma-Aldrich | N6789 |
| Rabbit anti-TH antibody | Merck Millipore | AB152 |
| Mouse anti-oxytocin antibody | Merck Millipore | MAB5296 |
| Guinea pig anti-vasopressin antibody | Peninsula Laboratories | T-5048 |
| Rabbit anti-CRH antibody | Peninsula Laboratories | T-4037 |
| Rabbit anti-TRH antibody | Santa Cruz | sc-366754 |
| Rabbit-c-Fos antibody | Cell signaling technology | #2250 |
| Rabbit anti-DsRed antibody | Takara Bio | 632496 |
| Mouse anti-β-tubulin III antibody | Sigma-Aldrich | T8578 |
| Alexa Fluor 488 goat anti-rabbit IgG | Thermo Fisher Scientific | A-11008 |
| Alexa Fluor 594 donkey anti-rabbit IgG | Thermo Fisher Scientific | A-21207 |
| Alexa Fluor 488 goat anti-mouse IgG | Thermo Fisher Scientific | A-11029 |
| Alexa Fluor 594 goat anti-mouse IgG | Thermo Fisher Scientific | A-11005 |
| Alexa Fluor 488 goat anti-guinea pig IgG | Thermo Fisher Scientific | A-11073 |
| Donkey anti-mouse Alexa Fluor 405 | Thermo Fisher Scientific | A31553 |

| Reagent/resource | Reference or source | Identifier or catalog number |
|---|---|---|
| Donkey anti-rabbit Alexa Fluor 594 | Thermo Fisher Scientific | A-21207 |
| Sheep anti-DIG-alkaline phosphatase antibody | Roche | 1093274 |
| **Oligonucleotides and other sequence-based reagents** | | |
| **Chemicals, enzymes, and other reagents** | | |
| **Software** | | |
| IBM SPSS Statistics 23 | IBM | |
| GraphPad Prism v.10.4.1 | GraphPad Software | |
| **Other** | | |
| Illumina NextSeq 500 | Illumina | |
| Fluorescent AgRP EIA kit | Phoenix Pharmaceuticals | FEK-003-57 |
| Mouse neuropeptide Y EIA | RayBiotech Life | EIAM-NPY-1 |

## Animals

All animal experiments were approved by the Institutional Animal Care and Use Committee of Gunma University. The mice were housed at room temperature (22–24 °C) with a 12-h light/dark cycle. All the mice used in this study were backcrossed with C57BL/6J mice for more than six generations. Regular feed (CLEA Rodent Diet CE-2; CLEA Japan, Tokyo, Japan) and water were provided ad libitum unless otherwise specified. For the special diet study, an HF diet (HFD32, CLEA Japan, Tokyo, Japan) with 32% fat and 6.75% sucrose was fed to the mice from 4 weeks of age.

## Generation of conditional *Fto*-KO mice

*Fto*$^{tm1a(EUCOMM)Wtsi}$ mice (RRID: IMSR_EM:05094) were obtained from the International Mouse Phenotyping Consortium. Heterozygous mice with a *lacZ* sequence inserted into the intron between *Fto* exons 2 and 3 (*Fto-lacZ* mice) were used to analyze *lacZ* expression. To generate a conditional *Fto*-knockout mouse, *Fto*$^{tm1a(EUCOMM)Wtsi}$ mice were crossed with FLP knock-in mice (Farley et al, 2000) to remove the *lacZ* cassette flanked by FLP recombinase target (FRT) sequences. Then, the mice (*Fto*$^{lox/+}$) were crossed with *Tau*-Cre (Muramatsu et al, 2008), *Agrp*-Cre (Xu et al, 2005), *Sf1*-Cre (Dhillon et al, 2006), *Sim1*-Cre (Balthasar et al, 2005), or *Pomc*-Cre (Balthasar et al, 2004) mice to obtain conditional *Fto*-knockout, and NPY-hrGFP mice (van den Pol et al, 2009), and tdTomato mice (Ai14) (Madisen et al, 2010) were used to visualize specific cells.

## Generation of AgRP neuron-specific *Fto*-overexpressing mice

Mouse *Fto* cDNA (Sino Biological, Beijing, China, MG53159-G) and an Ires DNA fragment were ligated using the Mighty Mix DNA Ligation Kit (Takara Bio, Otsu, Shiga, Japan, 6023). The *Fto*-Ires

fragment was then inserted into pAAV-hSyn-DIO-hM4D (Gi)-mCherry (Addgene, plasmid #44362) (Krashes et al, 2011) between the AgeI and NheI sites, replacing the sequences for hM4D(Gi). AAV vectors were generated using the ultracentrifugation method, as previously described (Konno and Hirai, 2020). Briefly, the plasmids pAAV-hSyn-*Fto*-mCherry, pAAV-hSyn-DIO-mCherry (Addgene, #50459), pHelper.gck, and pAAV2/9 (AAV9) were co-transfected into HEK293T cells using polyethylenimine. Viral particles were harvested from the conditioned medium 6 days after transfection and purified using iodixanol (Optiprep; AXS-1114542-250ML; Alere Technologies AS, Oslo, Norway) by density gradient ultracentrifugation. The viral solution was further concentrated and formulated in D-PBS using a Vivaspin 20 (Sartorius, Göttingen, Germany). The genomic titers of the AAV vectors were determined by real-time quantitative PCR (qPCR) using the Power SYBR Green PCR Master Mix (Thermo Fisher Scientific) and the primers 5′-CTGTTGGGCACTGACAATTC-3′ and 5′-GAAGGGACGTAG-CAGAAGGA-3′ for the WPRE sequence. The expression plasmid was used as the standard. AAV-hSyn-Flex-*Fto*-mCherry or AAV-hSyn-Flex-mCherry ($2.69 \times 10^{13}$ vg/mL) was injected bilaterally (1 μl/side) into the ARC (coordinates of the bregma: anterior–posterior −1.7, lateral [from midline]: ±0.3 mm, dorsal–ventral: −5.8 mm) of 6-week old *Agrp*-Ires-Cre mice (Tong et al, 2008) using a Hamilton 10-μL syringe with a 33-gauge blunt-ended needle (#701) and a microinjector (IMS20, Narishige, Tokyo, Japan) mounted on a stereotaxic instrument at an injection rate of 100 nL/min.

## Generation of *Mettl3*-floxed mice

*Mettl3*-floxed mice were obtained through electroporation using a previously reported method with some modification (Horii et al, 2017; Kohro et al, 2020). According to the target DNA sequences, donor single-stranded oligodeoxynucleotides (ssODNs) with 5′ and 3′-homology arms flanking *loxP* and a restriction site were used (Fig. EV5A,B). Opti-MEM I (Thermo Fisher Scientific) containing pre-annealed CRISPR RNA (crRNA) (Alt-R CRISPR–Cas9 crRNA, IDT)/trans-activating CRISPR RNA (Alt-R CRISPR–Cas9 tracrRNA, IDT) (3 μM), recombinant Cas9 protein (100 ng/μl; GeneArt Platinum Cas9 Nuclease, Thermo Fisher Scientific) and ssODN (400 ng/μl; Ultramer, IDT) was used as the electroporation medium. First, a left *loxP* site was introduced into intron 1 of *Mettl3* by electroporation using C57BL/6J-derived zygotes. The edited embryos were then transferred to the oviducts of pseudopregnant female ICR mice to obtain left *loxP* male *Mettl3* mice. Next, *Mettl3*-floxed mice were obtained by introducing a right *loxP* site into intron 10 of *Mettl3* using male-derived zygotes derived from the left *loxP*. The loxed alleles were confirmed by PCR sequencing using the following primer sets: left *loxP*: 5′-AGCAGTG AGGGCAGAGAATC-3′ (Mettl3L-P1), 5′-GGAAAGGGTCAGT CCAGTCA-3′ (Mettl3L-P2); right *loxP*: 5′-CTCTTGCCTCCCT ACCTCCT-3′ (Mettl3R-P1), 5′-AGCCAGGCCTACTTCATTCA-3′ (Mettl3R-P3) (Fig. EV5B).

## Generation of AgRP neuron-specific *Kif1a* knockdown mice

The *Kif1a* CRISPR/Cas9 target sites were selected using Invitrogen TrueDesign Genome Editor. The target sgRNA sequences are as follows: sg*Kif1a*1: 5′-CAACTCCACAGAAATGGCCG-3′, sg*Kif1a*2: 5′-GAGGCATAGTTGATGTCCTC-3′, and sg*LacZ*: 5′-TGCGAAT ACGCCCACGCGAT-3′(Platt et al, 2014). The pMax plasmid vectors containing the U6 promoter, sgRNA, and gRNA scaffold, pMax-U6-sg*Kif1a*1-gRNA scaffold-U6-sg*Kif1a*2-gRNA scaffold (tandem) pMax-U6-sg*LacZ*-gRNA scaffold were purchased from GeneArt (Thermo Fisher Scientific). The U6-sgKif1a1-gRNA scaffold, U6-sg*Kif1a*2-gRNA scaffold, and U6-sg*LacZ*-gRNA were amplified by PCR. After restriction enzyme treatment, the PCR products were inserted into the MluI and ApaI sites of pAAV-hSyn-DIO-mCherry (Addgene #50459) to generate pAAV-U6-*Kif1a*sgRNA-hSyn-Flex-mCherry or pAAV-U6-LacZsgRNA-hSyn-Flex-mCherry. AAV vectors were generated as described above. Rosa26-LSL-Cas9 knock-in mice (Jackson strain #026175) (Platt et al, 2014) were crossed with *Agrp*-Ires-Cre mice to obtain heterozygous mice for both transgenes. Equal volumes of AAV-hSyn-Flex-sg*Kif1a* or AAV-hSyn-Flex-sg*LacZ*-mCherry ($4.82 \times 10^{12}$ vg/mL) and AAV-hSyn-*Fto*-mCherry or AAV-hSyn-Flex-mCherry ($2.69 \times 10^{13}$ vg/mL) were mixed and injected bilaterally (1 μl/side) into male mice at 7 weeks of age.

## In vivo analysis of mouse phenotypes

Body weights of group-housed mice were measured weekly. The equal numbers of knockout and control littermates were used for body weight measurements. Fat and lean masses were evaluated using a CT scanner (LaTheta; Hitachi Aloka Medical, Tokyo, Japan). The distance from the nose to the anus was recorded for body length measurements; food intake was also measured for individual mice. After 3 days of acclimation, locomotor activity and respiratory metabolism were measured simultaneously using an infrared light beam detection system (ACTIMO-100; Shinfactory, Fukuoka, Japan) and the Oxymax apparatus (Columbus Instruments, Columbus, OH, USA). Locomotor activity was measured as the total number of beam breaks along the *x* and *y* axes. Oxygen consumption was calculated by dividing the lean body weight with that measured by a CT scanner.

## Analysis of blood samples

Food was removed from the home cage for 3 h, and blood was then collected from the tail vein or after decapitation. Serum lipid levels were determined using a Hitachi 7180 autoanalyzer (Hitachi, Tokyo, Japan). Several reagents were used: L-Type TG M (FUJIFILM Wako Pure Chemical Industries, Osaka, Japan) for triglycerides, NEFA-SS (Eiken Chemical, Tokyo, Japan) for free fatty acids, L-Type CHO M (FUJIFILM Wako Pure Chemical Industries) for total cholesterol, Cholestest LDL (SEKISUI MEDICAL, Tokyo, Japan) for LDL cholesterol, Cholestest N HDL (SEKISUI MEDICAL) for HDL cholesterol, and Autokit Total Ketone Bodies (FUJIFILM Wako Pure Chemical Industries) for total ketone bodies.

## Western blotting

Tissues were lysed in RIPA buffer containing protease inhibitors, and supernatants were collected after centrifugation at $20,000 \times g$ for 10 min. Next, 6× sample buffer (NACALAI TESQUE, Kyoto, Japan, 09499-14) was added to the samples, which were incubated at 95 °C for

3 min, loaded onto SDS-PAGE gels (SuperSep Ace, FUJIFILM Wako Pure Chemical Corporation) and separated. Proteins were transferred to nitrocellulose membranes (Cell signaling technology, Danvers, MA, #12369) which were then blocked using StartingBlock™ (TBS) blocking buffer (Thermo Fisher Scientific, 37542) for 30 min and incubated overnight at 4 °C with the following primary antibodies as required: rabbit anti-FTO antibody (Proteintech; Rosemont, IL, 27226-1-AP, 1:1000), rabbit anti-GAPDH antibody (Santa Cruz; Dallas, TX, sc-25778, 1:1000), and rabbit anti-KIF1A antibody (Merck; SAB2104191, 1:1000) diluted in Can Get Signal Immunoreaction Enhancer Solution 1 (Toyobo, Osaka, Japan). After washing, the membranes were incubated with goat anti-rabbit IgG-HRP (Cayman Chemical; Ann Arbor, MI, 10004301, 1:10,000) for 1 h at room temperature. Chemiluminescence was detected using the Amersham ECL Prime Western Blot Detection Reagent (GE Healthcare; Chicago, IL, USA). Band intensity was measured using the ImageQuant LAS 4000 imager (GE Healthcare).

## m⁶A-seq

$m^6A$-seq was performed as previously described (Meyer et al, 2012; Widagdo et al, 2016) with minor modifications. Briefly, ARC tissues were obtained from 6-week-old male $Fto^{lox/lox}$ and $Fto^{lox/lox}$/Agrp-Cre mice, and total RNA was extracted using the QIAzol lysis reagent (Qiagen, Hilden, Germany). RNA was pooled from 15 mice (37 μg) for each group, with two groups prepared for each genotype. RNA was fragmented using a solution containing 100 mM $ZnCl_2$, 100 mM Tris-HCl (pH 7.0), and 0.1% Igepal CA-630 (Sigma, I8896) at 95 °C for 3 min. After ethanol purification, fragmented RNA was immunoprecipitated using rabbit anti-$m^6A$ antibody (Abcam, ab151230) and preincubated with anti-rabbit Dynabeads (Thermo Fisher Scientific, 11203D) for 2 h at 4 °C. Immunoprecipitated RNA was treated with elution buffer containing 5 mM Tris-HCl (pH 7.5), 1 mM EDTA, 0.05% SDS, 84 μg proteinase K (FUJIFILM Wako Pure Chemicals), and 15 μL RNaseOUT (Thermo Fisher Scientific), after which RNA was extracted using the QIAzol lysis reagent. Input samples that did not undergo immunoprecipitation and immunoprecipitated samples were used for library preparation ($n = 2$ per genotype). Immunoprecipitation with normal rabbit IgG (Cell Signaling Technology, #2729) did not yield a sufficient amount of RNA for library preparation, and the normal rabbit IgG control was omitted. A stranded library was prepared using the Ovation Mouse RNA-Seq system (NuGEN, San Carlos, CA, 0348-32). Ribosomal RNA was removed using an InDA-C-mediated adapter cleavage system (NuGEN, 0348-32). The final libraries were run on a NextSeq 500 using the NextSeq 500 High Output v2 Kit (Illumina, San Diego, CA, USA) for paired-end 43-bp sequencing. Raw reads were initially checked using FastQC software. For analysis of the $m^6A$ peak, the data were mapped to the reference mouse genome (mm10) using BWA. MACS2 was used to identify regions enriched by immunoprecipitation with $m^6A$. For differential analysis of gene expression in $Fto^{lox/lox}$ and $Fto^{lox/lox}$/Agrp-Cre mice, RNA sequencing data from the input samples were used. Between 42.7 and 49.3 M reads were obtained from each library, with an average of 45.9 M reads. Data were assigned to the reference mouse genome (mm10) using STAR and the TCC-edgeR (Robinson et al, 2010; Sun et al, 2013) package in R version 3.6.2 (R Foundation for Statistical Computing, Vienna, Austria, https://www.R-project.org/) was used to normalize and identify differentially expressed genes. Genes that differed from the control ($p < 0.5$) were used for the analysis. Sequencing data are accessible via the NCBI Sequence Read Archive (accession ID PRJNA430767). Gene ontology analysis was performed using DAVID 6.8 (https://davidbioinformatics.nih.gov).

## Isoform expression analysis

Hypothalamic tissues were dissected from 6-week-old male $Fto^{lox/lox}$ and $Fto^{lox/lox}$/Tau-Cre mice. Tissues from three mice were pooled for each group, and two groups were prepared for each genotype. Total RNA was isolated using the RNeasy Micro kit (QIAGEN, 74004). A strand library was prepared from 1 mg of total RNA using the KAPA mRNA HyperPrep Kit (Nippon Genetics, Tokyo, Japan) according to the manufacturer's instructions. The final libraries were run on a NextSeq 500 system with a NextSeq 500 High Output v2 Kit (Illumina, San Diego, CA) for 75 bp paired-end sequencing. Isoform analysis was performed using bowtie2 (version 2.2.9 (Langmead and Salzberg, 2012))-RSEM (version 1.3.3 (Li and Dewey, 2011)) pipeline. Briefly, reads were aligned against mm10 using bowtie2-RSEM with the default setting. Reads between 32.8 and 36.5 M were obtained from each library, with an average of 35.1 M reads. Normalization and detection of differentially expressed isoforms (DEI) were performed using the TCC-edgeR (Robinson et al, 2010; Sun et al, 2013) package in R version 3.6.2. Isoforms with a false discovery rate (FDR)-adjusted P value < 0.05 were deemed significant as they were DEIs. For the calculation of alternative splicing events, the reads were aligned using STAR (version 2.5.3a) to the mm10 genome with the following settings: --outFilterMultimapScoreRange 1 --outFilterMultimapNmax 20 --outFilterMismatchNmax 10 --alignIntronMax 500000 --alignMatesGapMax 1000000 --sjdbScore 2 --alignSJDBoverhangMin 1 --genomeLoad NoSharedMemory --limitBAMsortRAM 31000000000 --outFilterMatchNminOverLread 0.33 --outFilterScoreMinOverLread 0.33 --sjdbOverhang 100 --outSAMstrandField intronMotif --outSAMattributes NH HI NM MD AS XS --limitSjdbInsertNsj 2000000 --outSAMunmapped None --outSAMtype BAM SortedByCoordinate --outSAMheaderHD @HD VN: 1.4 --outSAMattrRGline ID: {ID} --outSAMmultNmax 1. The PSI value was then derived using SplAdder version 2.4.2 (Kahles et al, 2016) using the output bam files by STAR alignment. To visualize read coverage across splice junctions, RNA-seq data were processed using the Sashimi plot (Katz et al, 2015) function in Integrative Genomics Viewer (Broad Institute, http://software.broadinstitute.org/software/igv/) to visualize read coverage across splice junctions.

## Immunohistochemical analyses

Animals were deeply anesthetized with an intraperitoneal injection of 10% pentobarbital (10 mL/kg body weight) and then transcardially perfused with saline, followed by 10% neutral-buffered formalin (062-01661, FUJIFILM Wako Pure Chemical Industries, Osaka, Japan) at zeitgeber time (ZT) 6–9. The animals used for TRH and CRH immunohistochemistry were injected intracerebroventricularly with 0.5 μL of phosphate-buffered saline (PBS) containing 20 μg of colchicine 24 h before transcardial perfusion. The brains were removed, stored in the same fixative overnight at 4 °C, and then immersed in 20% sucrose in PBS overnight at 4 °C.

Coronal sections were cut to 25 μm using a cryostat (Leica CM3050S) (1:5 series). After washing with PBS, the sections were blocked with 1% bovine serum albumin (BSA) (Sigma, A2153) and 1% normal goat serum (NGS) (Rockland Immunochemicals, Gilbertsville, PA, USA, D204-00-0050) diluted in PBS containing 0.25% Triton X-100 (PBT) for 30 min. Sections were then incubated with rabbit secretogranin II antibody (BIODESIGN, K55101R, 1:500), mouse secretogranin II antibody (Abcam, ab20245, 1:500), rabbit VGAT antibody (GeneTex, GTX101908, 1:500), rabbit anti-NPY antibody (Immunostar, 22940, 1:2000), mouse NPY antibody (Santa Cruz, sc-133080, 1:100), rabbit anti-FTO antibody (LifeSpan Biosciences, Seattle, WA, LS-B7788, 1:200), rabbit anti-POMC antibody (Phoenix Pharmaceuticals, H-029-30, 1:500), rabbit anti-NUCB2 antibody (Sigma, N6789, 1:100), rabbit anti-TH antibody (Merck Millipore, AB152, 1:500), mouse anti-oxytocin antibody (Merck Millipore, MAB5296, 1:600), guinea pig anti-vasopressin antibody (Peninsula Laboratories, San Carlos, CA, T-5048, 1:500), rabbit anti-CRH antibody (Peninsula Laboratories, T-4037, 1:1000), rabbit anti-TRH antibody (Santa Cruz, sc-366754, 1:500), rabbit-c-Fos antibody (Cell signaling technology, #2250, 1:2000), rabbit anti-KIF1A antibody (abcam, ab180153, 1:100), or rabbit anti-DsRed antibody (for mCherry) (Takara Bio, 632496, 1:2500) diluted in 1% BSA/1% NGS/PBT overnight. After washing in PBS, sections were incubated with Alexa Fluor 488 goat anti-rabbit IgG (Thermo Fisher Scientific, Carlsbad, CA, A-11008, 1:400), Alexa Fluor 594 donkey anti-rabbit IgG (Thermo Fisher Scientific, A-21207, 1:400), Alexa Fluor 350 goat anti-rabbit IgG (Thermo Fisher Scientific, A-11046, 1:400), Alexa Fluor 488 goat anti-mouse IgG (Thermo Fisher Scientific, A-11029, 1:400), Alexa Fluor 594 goat anti-mouse IgG (Thermo Fisher Scientific, A-11005, 1:400), or Alexa Fluor 488 goat anti-guinea pig IgG (Thermo Fisher Scientific, A-11073, 1:400) diluted in 1% BSA/PBS for 40 min. For double staining of FTO and POMC, after FTO-immunofluorescence staining, the sections were incubated overnight with an anti-POMC antibody that was biotinylated using a Biotin Labeling Kit (Dojindo Molecular Technologies, Rockville, MD, LK03) (1:400) diluted in 1.5% PBT. After washing with PBS, sections were incubated with streptavidin Alexa Fluor 594 conjugate (Thermo Fisher Scientific, S-11227, 1:400) for 40 min. The sections were then washed, mounted on slides, and cover-slipped with mounting medium (Vector Laboratories, H-1200).

The neurite density was determined as follows: After immuno-fluorescence staining for NPY or secretogranin II, fluorescence images were acquired using a fluorescence microscope (BZ-9000; Keyence). Subsequently, using ImageJ software v1.48 (National Institutes of Health, Bethesda, MD, USA), each image plane was binarized to isolate the labeled fibers from the background, and the dimensions of the immunopositive areas were calculated. For neurite length measurements, 25 z-stack confocal images (0.3-μm pitch, 7.2 μm thick) of NPY-hrGFP were acquired using a fluorescence microscope (BZ-X800, Keyence) and a full focus depth composition image was generated using BZ-X 3D analysis software (BZ-H4R, Keyence). The length of each neurite was measured using a BZ-X measurement application (BZ-H4M, Keyence).

## Electron microscopy

Electron microscopic analysis of the DCV was performed as previously described (Tsuchiya et al, 2010). Briefly, mice were perfused with a fixative solution consisting of 2.5% glutaraldehyde and 2% paraformaldehyde in 0.1 M cacodylate buffer (pH 7.4). The hypothalamic tissues were dissected. The tissues were fixed in the same buffer overnight at 4 °C, post-fixed in 1% $O_SO_4$ in 0.1 M cacodylate buffer for 1.5 h on ice, and embedded in Epon. Next, 80-nm-thick sections were prepared, contrasted with saturated aqueous solutions of uranyl acetate and lead citrate, and photographed using a JEM-1010 electron microscope (JEOL, Tokyo, Japan). The number of DCVs, defined by the presence of a dense core, was counted.

## Measurement of AgRP release from brain slices

AgRP release from the brain slices was measured as previously described (Enriori et al, 2007; Nakajima et al, 2016) with slight modifications. Briefly, a 2-mm-thick coronal brain slice between −0.30 mm and −2.30 mm from the bregma was cut using a MicroSlicer DTK-1000N (DOSAKA EM, Kyoto, Japan). The slices were then incubated in a 24-well plate containing aCSF with 10 mM glucose, equilibrated with 95% $O_2$ and 5% $CO_2$ at 37 °C for 1 h. Subsequently, the slices were incubated in 500 μL of equilibrated aCSF containing 10 mM glucose and 1.7 μL/mL protease inhibitor cocktail (Sigma) at 37 °C for 1 h to determine the basal release of AgRP. The slices were then incubated for an additional hour in either aCSF containing 10 mM glucose and 50 mM KCl, or aCSF containing 2.5 mM glucose and 100 μM glutamate. The entire supernatant was collected, and the AgRP concentration was determined using a fluorescent AgRP EIA kit (FEK-003-57, Phoenix Pharmaceuticals). AgRP release compared to the baseline was calculated.

## Electrophysiology

Whole-cell recordings were performed in $Fto^{lox/lox}$/NPY-hrGFP and $Fto^{lox/lox}$/$Agrp$-Cre/NPY-hrGFP mice. Hypothalamic slices containing NPY neurons were prepared as described previously (Suyama et al, 2016; Suyama et al, 2017). Briefly, after isoflurane and decapitated, the brains were rapidly removed and immersed in a cold (4 °C) carboxygenated high-mannitol solution containing (mM): mannitol 220, KCl 2.5, $NaH_2PO_4$ 1.23, $NaHCO_3$ 26, $CaCl_2$ 1, $MgCl_2$ 6, and glucose 10, with pH adjusted to 7.3 with NaOH. After trimming to a small block of tissue containing the hypothalamus, coronal slices (300 μm thick) were cut on a vibratome and maintained in a holding chamber with artificial cerebrospinal fluid (ACSF, bubbled with 5% $CO_2$ and 95% $O_2$) containing (mM): NaCl 124, KCl 3, $CaCl_2$ 2, $MgCl_2$ 2, $NaH_2PO_4$ 1.23, $NaHCO_3$ 26, and glucose 10, with pH adjusted to 7.4 with NaOH. After a 1-h recovery period, the slices were transferred to a recording chamber and constantly perfused with aCSF (33 °C) at a rate of 2 mL/min. Whole-cell patch-clamp recordings were performed on NPY-hrGFP neurons using a current clamp as previously reported (Suyama et al, 2017). The micropipettes were made of borosilicate glass (Narishige, Tokyo, Japan) with a Sutter micropipette puller (P-1000) and backfilled with a pipette solution containing (mM) potassium gluconate 135, $MgCl_2$ 2, HEPES 10, EGTA 1.1, Mg-ATP 2.5, $Na_2$-GTP 0.3, $Na_2$-phosphocreatine 10, with a pH set to 7.3 with KOH. The input and series resistances were monitored throughout the experiments, and the input resistance was partially compensated. Only recordings with stable

series resistance and input resistance were accepted. All data were sampled at 3–10 kHz, filtered at 1–3 kHz, and analyzed using a PClamp 10 (Axon Instruments, Union City, CA, USA).

## Calcium imaging of isolated NPY-hrGFP neurons

Calcium imaging of isolated NPY-hrGFP neurons was performed as previously described (Kohno et al, 2003) with slight modifications. A brain slice containing the ARC was prepared on ice using a surgical blade, and the ARC tissue was excised. The dissected ARC tissue was then incubated in 10 mM HEPES-buffered Krebs-Ringer bicarbonate buffer (HKRB) containing 10 mM glucose, 20 U/ml papain (Sigma-Aldrich, P4762), 0.015 mg/ml DNase II (Sigma-Aldrich, D-4138), and 0.75 mg/ml BSA (Sigma-Aldrich, A2153) for 16 min at 36 °C. Subsequently, gentle mechanical trituration was performed, followed by the centrifugation of the cell suspension at $100 \times g$ for 5 min. The pellet containing the isolated single ARC cells was resuspended in HKRB and distributed onto coverslips. The cells were maintained in moisture-saturated dishes for up to 4 h at 33 °C. The cytosolic $Ca^{2+}$ concentration ($[Ca^{2+}]i$) was measured using ratiometric fura-2 microfluorometry combined with digital imaging. Briefly, after incubation with 2 μM fura-2/AM (Dojindo, Kumamoto, Japan, F016) for 45 min, the cells were mounted in a chamber at 33 °C and superfused with HKRB containing 10 mM glucose at 1 ml/min at 33 °C. Fluorescence images due to excitation at 340 and 380 nm were detected every 10 s using a cooled charge-coupled device camera (ORCA-R2 C10600, Hamamatsu Photonics, Hamamatsu, Japan), and ratio images were generated using Aquacosmos (Hamamatsu Photonics). Data were obtained from single cells that exhibited NPY-hrGFP fluorescence.

## KIF1A ATPase activity test

6xHis and stop codons were added at the 3′ end of the cDNA *Kif1a-201*/ENSMUST00000086819.12 (1–1446 nt) and the cDNA *Kif1a-203*/ENSMUST00000171556.8 (1–1419 nt). Then the NdeI and PstI sequences were added at 5′ and 3′ ends, respectively. Purified PCR products and pCold III vector (Takara Bio, Shiga, Japan) were treated with NdeI and PstI. Then, *Kif1a-201*(1–1446 nt)-His or *Kif1a-203*(1–1419 nt)-His was inserted into the pCold III vector using a DNA ligation kit (Takara Bio, 6023) to produce pCold III-*Kif1a-201*(1-482aa)-His and pCold III-*Kif1a-203* (1-473aa)-His. These vectors were then transfected into ZIP-competent BL21 (DE3) cells (BioDynamic, Tokyo, Japan). Transfected *E. coli* cells were cultured in LB at 37 °C until the optical density at 600 nm ($OD_{600}$) reached approximately 0.5. IPTG (0.5 mM) was added, and *E. coli* was cultured at 15 °C for 24 h. His-tagged proteins were purified from the *E. coli* pellet using the QIAexpress Ni-NTA Fast Start kit (QIAGEN). The KIF1A protein was further purified via anion exchange chromatography on a Source15Q column (Cytiva, Tokyo, Japan) attached to an AKTA pure25 system (Cytiva), with a gradient of 0–500 mM NaCl in 20 mM Tris-HCl buffer (pH 8.5) containing 0.15 mM ATP and 0.035% 2-mercaptoethanol to elute the protein. The fractions containing KIF1A were pooled, and the solvent was transferred to kinesin reaction buffer from the KINESIN ATPase END-POINT BIOCHEM KIT (Cytoskeleton, Denver). ATPase activity was measured using the KINESIN ATPase END-POINT BIOCHEM KIT (cytoskeleton) according to the manufacturer's instructions. Briefly, 150 ng of KIF1A protein

and a specific number of microtubules were added to a kinesin reaction buffer containing taxol, and 30 μL of the reaction solution was prepared. The reaction was initiated by adding 10 μl of 2 mM ATP and stopped after 5 min by adding 70 μl of CytoPhos. The absorbance was measured at 650 nm using a SpectraMax ABS microplate reader (Molecular Devices, San Jose, CA, USA). The absorbance of the reaction in the absence of microtubules was subtracted from that in the presence of the microtubules. The inorganic phosphorus (Pi) concentration was determined using a standard Pi curve. ATPase activity was measured in triplicate. The $K_m$ value was calculated using the Lineweaver–Burk equation, and the $k_{cat}$ value was calculated by dividing $V_{max}$ by KIF1A concentration (μM).

## Size-exclusion chromatography

The KIF1A-201(1-482)-His and KIF1A-203(1-473)-His proteins were purified as described above and concentrated at approximately 1.2 μg/μl using the Amicon Ultra-4 centrifugal filter (Sigma-Aldrich). A total of 250 μg of each protein was applied to size-exclusion chromatography (Superdex 200 Increase 10/300 GL, Cytiva, 28-9909-44) connected to an AKTA pure25 system (Cytiva) with a buffer consisting of 20 mM PIPES of-KOH (pH 7.0), 150 mM KCl of, 1 mM of $MgCl_2$, and 0.1% Tween 20. Standard globular proteins, including thyroglobulin, ferritin, aldolase, albumin, and ovalbumin (Gel Filtration Calibration Kit, Cytiva, 403842), were used to prepare the calibration curve. Kav was calculated using the formula Kav = (Ve − Vo)/(Vc − Vo), where Vo is the column void volume, as indicated by the blue dextran 2000 (Cytiva) peak, Ve is the elution volume, and Vc is the geometric column volume. The calibration curve was prepared by plotting Kav against the corresponding $log_{10}$ molecular weight value (Fig. EV11C,D), which was used to determine the molecular weights of the KIF1A fragments.

## Tubulin preparation and gliding assay

The preparation and gliding assay were performed according to previously described methods (Morikawa et al, 2022). Tubulin was purified from porcine brains using six cycles of polymerization and depolymerization in high-molarity PIPES buffer (1 M PIPES-KOH, 1 mM EGTA, 1 mM $MgCl_2$, pH 6.8) to remove contaminating microtubule-associated proteins. The purified tubulin was then polymerized in PEM buffer (100 mM PIPES-KOH, 1 mM EGTA, 1 mM $MgCl_2$, pH 6.8) containing 1 mM GTP and 7% DMSO at 37 °C for 30 min. For the gliding assay, tubulin labeled with Alexa Fluor 647 (AF647) succinimidyl ester (Thermo Fisher Scientific) was prepared by incubating polymerized microtubules with dye for 30 min at 37 °C. Functionally labeled tubulin was purified through two additional cycles of polymerization and depolymerization. Absorption spectra were used to determine labeling efficiency.

For the gliding assay, a mixture of 80% tubulin and 20% AF647 tubulin was polymerized at 37 °C for 20 min in BRB80 buffer (80 mM PIPES-KOH, 1 mM $MgCl_2$, 1 mM EGTA, pH 6.8) containing 1 mM GTP and 10% DMSO, followed by stabilization with the addition of 10 μM taxol at 37 °C for another 20 min. Microtubules were collected from the pellet by ultracentrifugation at 35,000 rpm for 15 min using an S55A2 rotor (Eppendorf Himac). The assay was performed in a chamber made by sandwiching two No.1S coverslips (Matsunami

Glass, Osaka, Japan). An anti-His tag antibody (MBL International Cat# D291-3, RRID: AB_10597733) diluted 1:8 into BRB80 was injected in the chamber and incubated at room temperature for 5 min. Blocking was performed sequentially with 1% Pluronic F127 in BRB80 and 0.5 mg/mL κ-casein in BRB80 for 5 min each. The purified motor protein at 40 μg/mL in BRB80 was then added, dissolved, and incubated for 5 min. After several washes with BRB80, a small number of microtubules flowed in.

Observations were conducted in BRB80 buffer supplemented with an oxygen scavenging system as follows: 80 mM PIPES-KOH, 1 mM $MgCl_2$, 1 mM EGTA, 10 μM taxol, 5 mM ATP, 0.5 mg/mL κ-casein, 560 μg/mL glucose oxidase, 136 μg/mL catalase, and 2% glucose. Imaging was performed using a Nikon Eclipse Ti inverted microscope with a 647 nm fiber laser (MPB Communications), a Plan Apo TIRF ×100 Oil Immersion objective, and maintained at 37 °C in a homemade temperature-controlled chamber with a Perfect Focus System (PFS). Time-lapse images were recorded using an EMCCD iXon3 DU897 (Oxford Instruments) at 512 × 512 resolution, with an exposure time of 0.5 s and gain of 300, acquiring 100 images over 49.5 s.

Kymographs were created using the KymoResliceWide plugin (Eugene Katrukha) of FIJI, and velocities were calculated by computing tangents from the average slope angles of detectable lines by pixel size and pixel time.

## Hippocampal primary culture and live imaging

For KIF1A-EGFP, the full-length Kif1a cDNA was amplified by PCR using the forward primer (5′-GTAGAATTCGCCAC-CATGGCCGGGGCCTC-3′) and the reverse primer (5′-GTAG-GATCCCGGACCCGCATCTGCGCAGATC-3′), then inserted into the vector pEGFP-N1 at EcoRI and BamHI sites after proper digestion.

Primary hippocampal cell culture and transfection were performed as previously described (Ichinose et al, 2023). Hippocampi were dissected from ICR mice (Charles River; IMSR Cat# CRL: 022, RRID: IMSR_CRL: 022) on embryonic day 16 (E16). Sex was not determined, and three or more embryos were used. The hippocampi were digested with 0.25% trypsin (Thermo Fisher Scientific) in HBSS (FUJIFILM Wako) for 15 min at 37 °C. Dissociated hippocampal cells were seeded at a density of $3 \times 10^4$ cells per well on an eight-well chamber cover (Matsunami Glass) coated with 0.04% polyethyleneimine (Merck) and BioCoat poly-D-lysine (Corning, Corning, NY, USA). All primary cells were cultured in MEM (Thermo Fisher Scientific) supplemented with 1 mM pyruvate (Thermo Fisher Scientific), 0.6% glucose, 2 mM GlutaMAX (Thermo Fisher Scientific), 2% B27 Plus (Thermo Fisher Scientific), and 100 U/mL Penicillin-Streptomycin (Thermo Fisher Scientific). Cells were maintained at 37 °C in a humidified atmosphere containing 95% air and 5% $CO_2$. Cultured neurons were transfected using the High-Efficiency $Ca^{2+}$ Phosphate Transfection Kit (Takara Bio). The culture medium was replaced with fresh MEM containing pyruvate, glucose, and GlutaMAX. Next, a mixture of 2 μg of plasmid, 3.1 μl of the 2 M $CaCl_2$, and 50 μl of Hank's equilibrium salt solution was prepared and incubated at room temperature for 15 min. The DNA/$Ca^{2+}$ phosphate suspension was then added to the culture medium and incubated in an incubator with 5% $CO_2$ at 37 °C for 40 min. Subsequently, the DNA/$Ca^{2+}$ phosphate precipitates were dissolved

for 15 min in a pre-equilibrated medium in a 10% $CO_2$ incubator before being replaced with the original medium.

Live imaging was recorded using a confocal laser scanning microscope (LSM 880; ZEISS) equipped with a ×63/1.4 Plan Apochromat oil immersion objective with a resolution of 512 × 75 at a pixel time of 16.48 μs and 1.65 s frame intervals. Neurons were carefully selected to avoid elevated levels. During observation, the medium was replaced with Leibovitz L-15 medium (FUJIFILM Wako) and kept at 37 °C in a homemade temperature-controlled chamber. Kymographs were created using the KymoResliceWide FIJI plugin, and the velocities were calculated by computing tangents from the average slope angles of detectable lines by pixel size and pixel time.

## Analysis of Kif1a-expressing PC12 cells

Mouse Kif1a-201/ENSMUST00000086819.12 or Kif1a-203/ENSMUST00000171556.8 cDNA and 3xHA sequences were inserted into the pEF-BOS vector (Mizushima and Nagata, 1990) to generate the plasmid vectors pEF-BOS-Kif1a-201-3xHA and pEF-BOS-Kif1a-203-3xHA. The plasmid and the pEGFP-N1 vector were transfected into PC12 cells using Lipofectamine 3000 (Thermo Fisher Scientific) after linearization of the pEGFP-N1 vector with Af1III. Transfected cells were selected using a medium containing 700 μg/mL G418 (FUJIFILM Wako Pure Chemical Industries, 074-06801). Single colonies were isolated to obtain a homogeneous cell population. For reverse transcription PCR, RNA was extracted from cells, and cDNA was synthesized using the ImProm-II Reverse Transcription System (Promega, A3800). PCR was carried out using the primers Kif1a-exon12/13/14F (5′-GAGA-GACCTGCTGTATGCCC-3′) and Kif1a-exon12/13/14 R (5′-GAA-GATGAGGGGCTCATGCC-3′). The PCR products were analyzed on 2% agarose gels, and images were captured using a WSE-5400 Printgraph Classic (ATTO, Tokyo, Japan). Band densities were measured using ImageJ software. To measure NPY release, the cells were seeded in 24-well plates. Neural differentiation was induced by treating cells with 50 ng/mL NGF (Sigma, SRP3015) for 1 week. Two different DMEMs (11965, Thermo Fisher Scientific and high-KCl DMEM (50 mM KCl and 65 mM NaCl based on 11965; custom-ordered; Invitrogen) were used. Both DMEMs were fully equilibrated in a 5% $CO_2$ atmosphere at 37 °C before use. Culture medium was collected 10 min after each treatment, and the NPY concentration was measured using mouse neuropeptide Y EIA (RayBiotech Life, Peachtree Corners, GA, USA, EIAM-NPY-1) following the manufacturer's instructions. For immunocytochemistry, 1000 cells/well were seeded in plates with poly-L-lysine-coated cover glasses in culture medium (DMEM + 5% FBS and 5% horse serum). From the following day, the cells were treated with 50 ng/mL NGF for 5–7 days to induce differentiation. Cells were then fixed with 4% PFA, blocked with 5% FBS in PBS for 30 min, and immunostained with a neuronal marker mouse anti-β-tubulin III antibody (Sigma, T8578, 1:200) and rabbit anti-NPY antibody (Immunostar, 22940, 1:1000). The secondary antibodies used were the donkey anti-mouse Alexa Fluor 405 (Thermo Fisher, A31553, 1:200) and donkey anti-rabbit Alexa Fluor 594 (Thermo Fisher, A-21207, 1:200). PC12 cells were observed using a laser scanning confocal microscope (Zeiss LSM 880, Carl Zeiss Microscopy GmbH).

## Mouse *Kif1a* mRNA exon 13 splicing analysis

Seven-week-old C57BL/6J mice were sacrificed after being fed ad libitum, fasted overnight, or fed for 3 h, and ARC tissues were dissected at ZT 3. RNA was extracted, cDNA was synthesized, and PCR was performed using the primers Kif1a-95F (5′-AGGCTGA-GAGACCTGCTGT-3′) and Kif1a-96R (5′-GGACAGGGCTGA-GAGTGAAG-3′). The PCR products were run on 2% agarose gels, images of the DNA bands were obtained, and their band densities were measured using ImageJ software. For the quantitative analysis of exon 13-including *Kif1a* mRNA, total *Kif1a* and exon 13-including *Kif1a* copy numbers were quantified by QIAcuity digital PCR (QIAGEN, Hiden, Germany) with the QIAcuity Nanoplate 8.5k 24-well (QIAGEN, 250011), QIAcuity probe PCR kit (QIAGEN, 250101), and the following TaqMan probes: for the detection of total *Kif1a* mRNA, Mm00492863_m1 (Thermo Fisher Scientific) and for specific detection of *Kif1a* mRNA containing exon 13, a custom-designed TaqMan probe set—Kif1a-Exon12/13/14 (probe: 5′-ACCAACACTGTGCCCGGAGGACCCAAATTG-3′, forward primer: 5′-GAGAGACCTGCTGTATGCCC-3′, and reverse primer 5′-GAAGATGAGGGGCTCATGCC-3′).

## In situ hybridization

In situ hybridization was performed using two methods: (1) the shHCR method with cDNA probes and fluorescent short hairpin DNA and (2) digoxigenin (DIG)-labeled cRNA probes and NBT/BCIP.

shHCR in situ hybridization was performed as previously described (Tsuneoka and Funato, 2020). To detect *Fto* and *Agrp* mRNA, the cDNA probes contain split initiator sequences for the amplification of hairpin DNAs S45 and S41 (Table EV1). Brain slices were mounted on glass slides, washed with PBS, and immersed in methanol for 10 min. After washing with PBS containing 0.1% Tween 20 (PBST) twice for 5 min each, the slices were prehybridized before hybridization for 5 min in a hybridization buffer containing 10% dextran sulfate, 0.5× saline sodium citrate (SSC), 0.1% Tween 20, 50 μg/mL heparin, and 1× Denhardt's solution. Slices were then treated overnight with a hybridization buffer containing a mixture of 25 nM probes at 37 °C. Subsequently, they were washed three times for 10 min each in 0.5× SSC containing 0.1% Tween 20 at 37 °C. The ISHpalette short hairpin amplifiers SaraFluor488-S45 (IPL-G-S45, NEPA GENE, Chiba, Japan) and ATTO550-S41 (IPL-R-S41, NEPA GENE) were heated to 95 °C for 1 min, then gradually cooled to 65 °C for 15 min and 25 °C for 40 min. Slices were incubated in amplification buffer (10% dextran sulfate in 8× SSC, 0.2% Triton X-100, 100 mM MgCl₂) for 5 min, immersed in amplification buffer containing diluted hairpin DNA (1:50) for 2 h at 25 °C and then washed with PBST and PBS. Fluorescent images were obtained using an FV3000 microscope (Olympus, Tokyo, Japan). Images of *Fto* and *Agrp* shHCR in situ hybridization were overlaid, and *Agrp*-expressing cells were selected. Finally, the mean *Fto* shHCR in situ hybridization signal intensity was measured using ImageJ software.

In situ hybridization using DIG-labeled cRNA probes was performed as previously described (Liang et al, 2000), with minor modifications. *FTO* cRNA probes were generated by PCR amplification of mouse brain cDNA using the primers 5′-CTACCTCCAGGTGGAGACCAT-3′ and 5′-GCAGTCTCCCTGGTG

AA-3′. The amplified PCR product was gel-purified and subcloned into the pGEM-T Easy Vector (Promega) according to the manufacturer's protocol. The sequence and orientation of the insert were confirmed by DNA sequencing. The amplicon inserted in the plasmid was then amplified using the primers M13F (5′-GTAAAACGACGGCCAGT-3′) and M13R (5′-GGAAACAGC-TATGACCATG-3′). To generate antisense DIG-labeled cRNA probes, in vitro transcription using T7 or SP6 RNA polymerases was performed according to the manufacturer's instructions (Roche, 1277073). Sections were washed in diethyl pyrocarbonate (DEPC)-treated 0.1 M phosphate buffer at pH 7.4 followed by 0.1 M phosphate buffer containing Triton X-100. The sections were then incubated in acetylation buffer containing 0.25% acetic anhydride and 0.1 M triethanolamine (pH 8.0) for 10 min. After washing in DEPC-treated PBS, sections were incubated for 1 h at 60 °C in hybridization buffer containing 5× SSC (NIPPON GENE, 318-90225), 2% blocking reagent (Roche 1096 176), 50% formamide (Nacalai Tesque, Kyoto, Japan), 0.1% N-lauroylsarcosine (NLS) (Sigma, St. Louis, MO, L7414), and 0.1% sodium dodecyl sulfate (NIPPON GENE, 311-90271). The sections were incubated overnight with 4 μg/mL DIG-labeled RNA probes diluted in hybridization buffer at 60 °C and then sequentially washed in 2× SSC containing 50% formamide and 0.1% NLS at 60 °C, 2× SSC containing 0.1% NLS at 37 °C, and 0.2× SSC containing 0.1% NLS at 37 °C. After incubation in 0.1 M Tris-HCl buffer (pH 7.5) containing 0.15 M NaCl, sections were incubated with anti-DIG-alkaline phosphatase antibody (Roche, 1093274, 1:1,000) diluted in 1% blocking reagent for 5 h. Sections were then washed in 0.1 M Tris-HCl (pH 7.5) buffer containing 0.15 M NaCl and 0.05% Tween 20, followed by 0.1 M Tris-HCl (pH 9.5) buffer containing 0.1 M NaCl, 10 mM MgCl2, and 0.1% Tween 20. Color development was conducted in NBT/BCIP solution (Roche, 11681451001) for 1 h at 37 °C, with the reaction stopped using PBS containing ethylene-diaminetetraacetic acid. Sections were washed for 48 h at 37 °C in 0.1 M Tris-HCl (pH 7.5) buffer containing 0.15 M NaCl and 0.5% Tween 20, mounted on glass slides, and covered using mounting medium (Dako, S3023) and a coverslip. Control experiments to confirm the specificity of this protocol involved hybridization with sense probes.

## X-gal staining for detection of β-galactosidase activity

After washing in PBS, sections of hemizygous Fto-*lacZ* mice tissue were incubated for 24 h at 37 °C in PBS solution containing 1 mg/mL of X-gal (FUJIFILM Wako Pure Chemical Industries, 029-07853), 5 mM $K_3Fe(CN)_6$ (FUJIFILM Wako Pure Chemical Industries, 167-03722), 5 mM $K_4Fe(CN)_6$ (FUJIFILM Wako Pure Chemical Industries, 161-03742), 2 mM MgCl₂ (Nacalai Tesque, 20909-55), 0.02% sodium deoxycholate (FUJIFILM Wako Pure Chemical Industries, 192-08312), and 0.02% Nonidet P-40 (Sigma, 21-3277-2-25G-J). After incubation, sections were washed with PBS and then mounted on glass slides or subjected to immunofluorescence staining. X-gal staining and fluorescence images were acquired using a BZ-9000 microscope (Keyence, Tokyo, Japan) and a FV10i-DOC confocal microscope (Olympus). The blue color in the bright-field photographs of X-gal staining was inverted for visualization, and the photographs were then combined using the screen-blending mode in Photoshop (Adobe; San Jose, CA). The number of cells was counted on the right and left sides of the ARC in each

section, and confocal microscopy images of X-gal staining were obtained at 647 nm excitation and 683 nm emission, as previously reported (Levitsky et al, 2013).

## Quantitative PCR (qPCR) analysis of RNA

ARC tissue samples were dissected, and total RNA was extracted using QIAzol lysis reagent (Qiagen, Hilden, Germany) and chloroform. RNA samples were treated with DNase I and reverse-transcribed using ReverTra Ace qPCR RT Master Mix with gDNA Remover (FSQ-301; TOYOBO, Osaka, Japan). TaqMan assays were performed using primers and probes against *Fto* (Mm00488755_m1) and 18S rRNA (Hs99999901_s1).

## Statistical analysis

Data are presented as mean ± SEM values unless otherwise specified. Statistical analyses were performed using IBM SPSS Statistics 23 software (IBM, Armonk, NY, USA) and GraphPad Prism v.10.4.1 (GraphPad Software, San Diego, CA, USA). After confirming the normal distribution of the data, comparisons between two genotypes were performed using unpaired Student's *t* tests unless otherwise specified. For statistical analyses involving more than two groups, one-way analysis of variance (ANOVA) was performed, followed by Tukey's multiple comparison test. $P < 0.05$ was considered statistically significant. Investigators were blinded to the group allocation during data collection and analysis.

# Data availability

m⁶A-seq data are accessible via the NCBI Sequence Read Archive (https://www.ncbi.nlm.nih.gov/bioproject/PRJNA430767).

The source data of this paper are collected in the following database record: biostudies:S-SCDT-10_1038-S44318-025-00503-3.

# Peer review information

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

## Acknowledgements

This study was supported by research grants from JSPS KAKENHI (25870666, 17K08566, 21H03349, 23K21617, 25K03043), the MSD Life Science Research Foundation, the Ono Medical Research Foundation, and the Koyanagi Foundation to DK and from the Takeda Science Foundation to TK and a research fellowship from the Takeda Science Foundation to VYS. This work was also supported in part by the Fostering Health Professionals for Changing Needs of Cancer; the Promotion Plan for the Platform of Human Resource Development for Cancer and New Paradigms—Establishing Centres for Fostering Medical Researchers of the Future programs of the Ministry of Education, Culture, Sports, Science and Technology (MEXT) of Japan; and the Gunma University Initiative for Advanced Research (GIAR). This work was the result of using the research equipment shared through the MEXT Project for promoting public utilization of advanced research infrastructure (Program for supporting the introduction of the new sharing system) Grant Number JPMXS0420600120. This research was supported by the Platform Project for Supporting Drug Discovery and Life Science Research (Basis for Supporting Innovative Drug Discovery and Life Science Research (BINDS)) from AMED under grant number JP21am0101120 to IH. This research was partially supported by the Multidisciplinary Frontier Brain and Neuroscience Discoveries program (Brain/MINDS 2.0; JP24wm0625103 to HH) from the Japan Agency for Medical Research and Development (AMED). This work was the result of using research equipment shared in the Institute for Molecular and Cellular Regulation Joint Usage/Research Support Center (IMCR-JURSC), Gunma University. We thank the Wellcome Trust Sanger Institute Mouse Genetics Project and the EMMA partner CNB–CSIC (Spain) for providing the *Fto*$^{tm1a(EUCOMM)Wtsi}$ mice. The authors thank Dr. Gregory S Barsh for providing the *Agrp*-Cre mice, Dr. Joel K Elmquist and Dr. Bradford B Lowell for providing the *Agrp*-Ires-Cre mice, NPY-hrGFP mice, the *Sf1*-Cre mice, and the *Sim1*-Cre mice. The authors thank Drs Takeshi Inagaki, and Kohichi Matsunaga for their helpful advice. The authors thank Mr. Y Morishita, Ms. S Umizawa, and Ms. S Fujimoto, and Ms. Y Watanuki for their helpful technical assistance.

## Author contributions

**Daisuke Kohno**: Conceptualization; Funding acquisition; Investigation; Writing—original draft; Project administration; Writing—review and editing. **Reika Kawabata-Iwakawa**: Investigation. **Sotaro Ichinose**: Investigation. **Shigetomo Suyama**: Investigation. **Kazuto Ohashi**: Supervision; Investigation. **Winda Ariyani**: Investigation. **Tetsushi Sadakata**: Resources; Supervision. **Hiromi Yokota-Hashimoto**: Investigation. **Ryosuke Kobayashi**: Investigation. **Takuro Horii**: Investigation. **Vina Yanti Susanti**: Investigation. **Ayumu Konno**: Investigation. **Haruka Tsuneoka**: Investigation. **Chiharu Yoshikawa**: Investigation. **Sho Matsui**: Supervision. **Akihiro Harada**: Resources. **Toshihiko Yada**: Supervision. **Izuho Hatada**: Supervision; Funding acquisition. **Hirokazu Hirai**: Supervision; Funding acquisition. **Masahiko Nishiyama**: Resources; Supervision. **Tsutomu Sasaki**: Supervision; Writing—review and editing. **Tadahiro Kitamura**: Supervision; Funding acquisition; Writing—review and editing.

Source data underlying figure panels in this paper may have individual authorship assigned. Where available, figure panel/source data authorship is listed in the following database record: biostudies:S-SCDT-10_1038-S44318-025-00503-3.

## Disclosure and competing interests statement

The authors declare no competing interests.

# Expanded View Figures

**Figure EV1.   Distribution of FTO in the hypothalamic feeding center.**

(A, B) In situ hybridization of *Fto*. *Fto* mRNA is abundantly localized in the arcuate nucleus (ARC), ventromedial hypothalamus (VMH) and paraventricular hypothalamus (PVH). Scale bar: (A) 500 μm, (B) 100 μm. (C) Transmitted light microscopy image of X-gal staining (light blue) in *Fto-LacZ* mice. *LacZ* expression patterns in *Fto-LacZ* mice recapitulated *Fto* expression patterns observed by *Fto* in situ hybridization. Scale bar: 500 μm. (D–G) Color-reversed transmitted light microscopy image of X-gal staining (red) and NPY-hrGFP fluorescence (green) (D) and immunofluorescence (green) of POMC (E), NUCB2 (F), or tyrosine hydroxylase (TH) (G) in the ARC of *Fto-lacZ* mice. Scale bar: 30 μm. (H) The percentage of X-gal-expressing neurons among neurons expressing NPY-hrGFP or immunoreactive for POMC, NUCB2, or TH in the ARC. $n = 3$ for each group; error bars represent SEM. (I) Confocal microscopy image of X-gal staining (blue) and NPY-hrGFP fluorescence (green) of *Fto-LacZ*/NPY-hrGFP mouse. Scale bar: 30 μm. 3V: third ventricle. (J–O) Color-reversed transmitted light microscopy image of X-gal staining (red) and the immunofluorescence (green) of oxytocin (J), vasopressin (K), NUCB2 (L), TH (M), corticotropin-releasing hormone (CRH) (N), or thyrotropin-releasing hormone (TRH) (O) in the PVH of *Fto-lacZ* mice. Scale bar: 30 μm. (P) The percentage of X-gal-expressing neurons among neurons immunoreactive for TRH ($n = 3$), TH ($n = 3$), oxytocin ($n = 3$), NUCB2 ($n = 4$), AVP ($n = 3$), or CRH ($n = 3$) in the PVH. Error bars represent SEM.

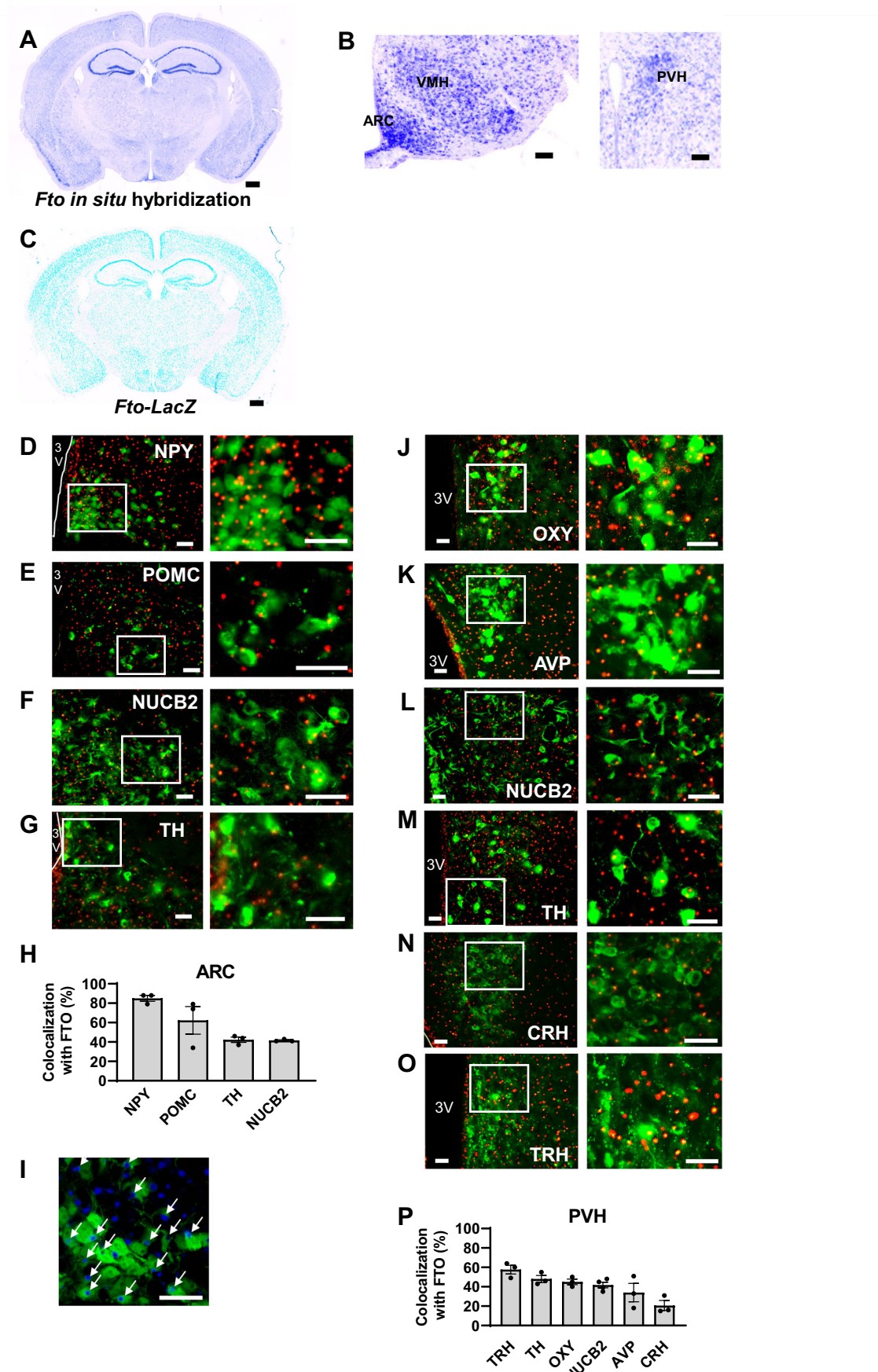

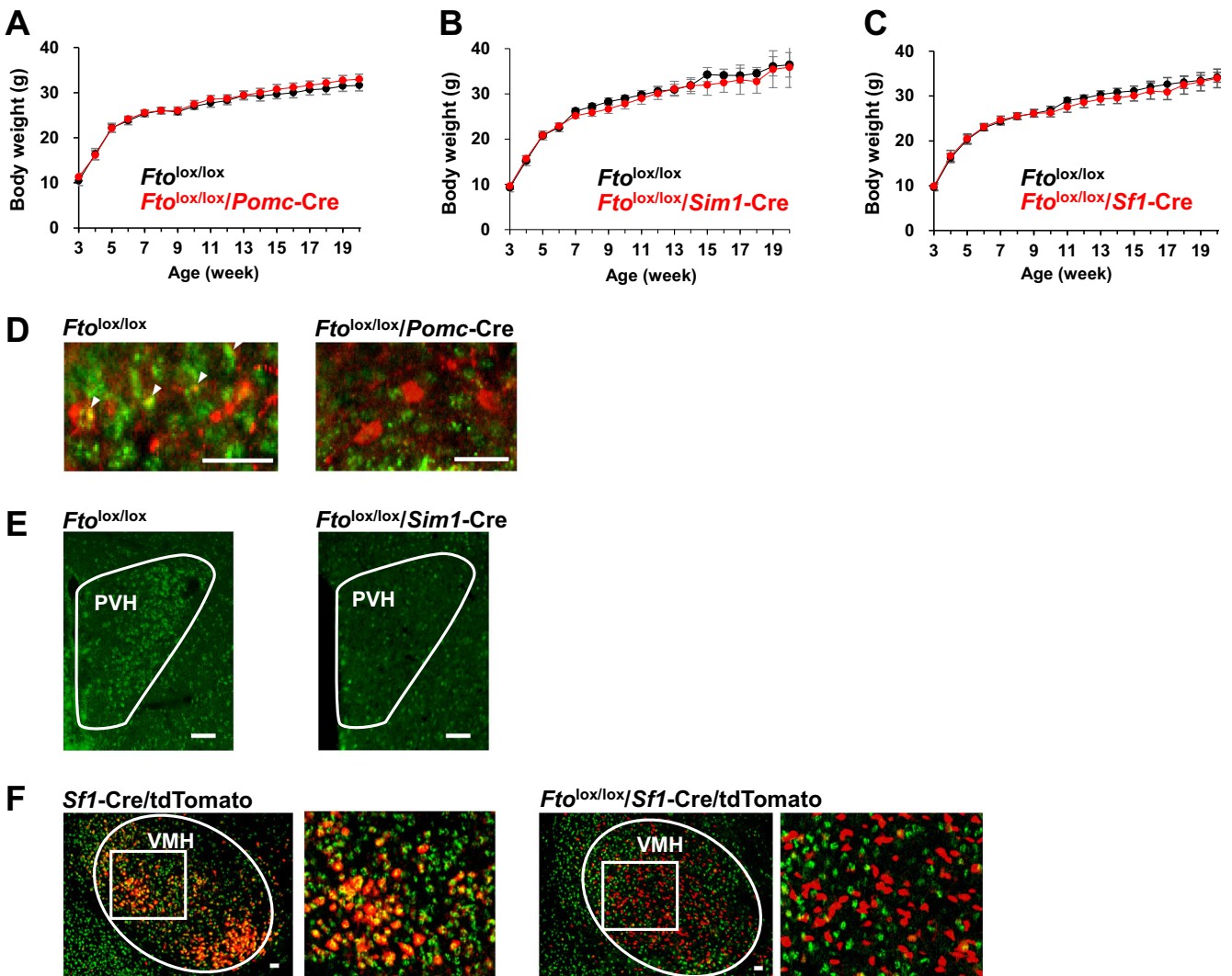

**Figure EV2. *Fto* conditional knockout mice specific for hypothalamic neurons.**

(A–C) The body weights of the male mice lacking *Fto* specifically in *Pomc*-Cre (*n* = 10–12) (*Fto*lox/lox/*Pomc*-Cre) (A), *Sim1*-Cre (*n* = 3–13) (*Fto*lox/lox/*Sim1*-Cre) (B), and *Sf1*-Cre (*Fto*lox/lox/*Sf1*-Cre) (*n* = 6–13) (C) were comparable to those of control (*Fto*lox/lox) mice. Error bars represent SEM. Data were analyzed using unpaired Welch's *t* test. (D–F) FTO immunofluorescence (green) was colocalized with POMC-immunofluorescence (red) (arrow) in *Fto*lox/lox mice but not in *Fto*lox/lox/*Pomc*-Cre mice (D). FTO immunofluorescence in the PVH was abundant and sparse in *Fto*lox/lox mouse and *Fto*lox/lox /*Sim1*-Cre mouse, respectively (E). FTO immunofluorescence colocalized with tdTomato fluorescence in the VMH of *Sf1*-Cre/tdTomato mice but not in VMH of *Fto*lox/lox/*Sf1*-Cre/tdTomato mice (F). Scale bar: 30 μm.

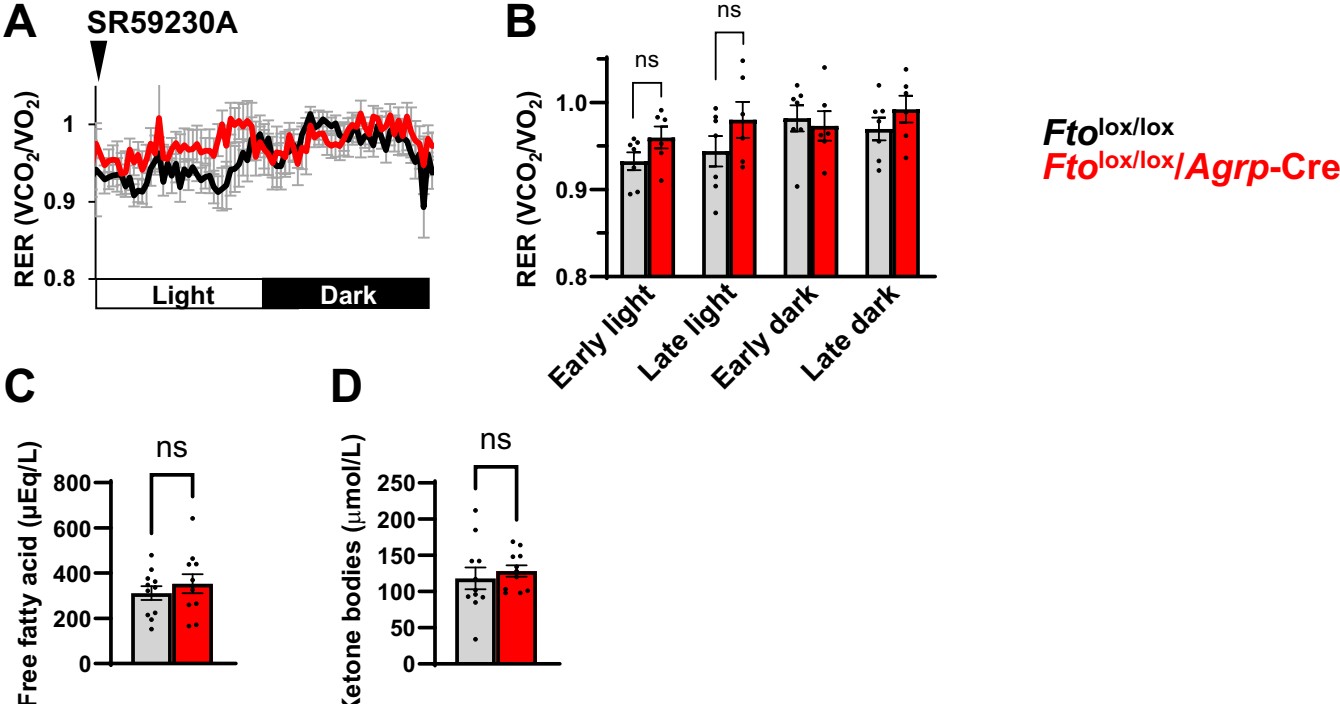

**Figure EV3. Administration of a β3 adrenergic receptor antagonist affected the levels of respiratory exchange ratio (RER), free fatty acid, and ketone bodies in *Fto*<sup>lox/lox</sup>/*Agrp*-Cre mice.**

(A) The RERs of $Fto^{lox/lox}$ ($n = 7$, black) and $Fto^{lox/lox}$/*Agrp*-Cre ($n = 6$, red) male mice injected intraperitoneally with SR59230A, a β3 adrenergic receptor antagonist (10 mg/kg body weight), at ZT0 (arrowhead). Error bars represent SEM. Data were analyzed using unpaired Student's $t$ test. (B) Average RER of $Fto^{lox/lox}$ ($n = 7$, gray) and $Fto^{lox/lox}$/*Agrp*-Cre ($n = 6$, red) male mice injected with SR59230A at ZT0. Error bars represent SEM. Data were analyzed using unpaired Student's $t$ test. (C, D) Levels of free fatty acids (C) and ketone bodies (D) in the serum of male $Fto^{lox/lox}$ (gray, $n = 11$) and $Fto^{lox/lox}$/*Agrp*-Cre (red, $n = 11$) mice injected with SR59230A at ZT0 and with blood collected at ZT2. Error bars represent SEM. Data were analyzed using unpaired Student's $t$ test.

**Figure EV4. Generation of AgRP neuron-specific *Fto* overexpression mice.**

Diagram of the Cre-inducible *Fto* overexpression AAV vector, AAV-hSyn-Flex-*Fto*-mCherry (**A**). Validation of the specific overexpression of FTO was conducted by western blotting using ARC samples from *Agrp*-Ires-Cre mice injected with either Flex-mCherry (control) or Flex-*Fto*-mCherry 4 weeks earlier. FTO protein levels were significantly higher in *Agrp*-Ires-Cre mice injected with AAV-hSyn-Flex-*Fto*-mCherry (green, $n = 3$) compared to those injected with AAV-hSyn-Flex-mCherry (gray, $n = 3$) (**B**). Error bars represent SEM. Data were analyzed using unpaired Student's *t* test; ****$P < 0.001$ (exact *P* value: 0.0002). Immunohistochemistry of mCherry using *Agrp*-Ires-Cre mice injected with AAV-hSyn-Flex-*Fto*-mCherry (**C**). Scale bar: 30 μm.

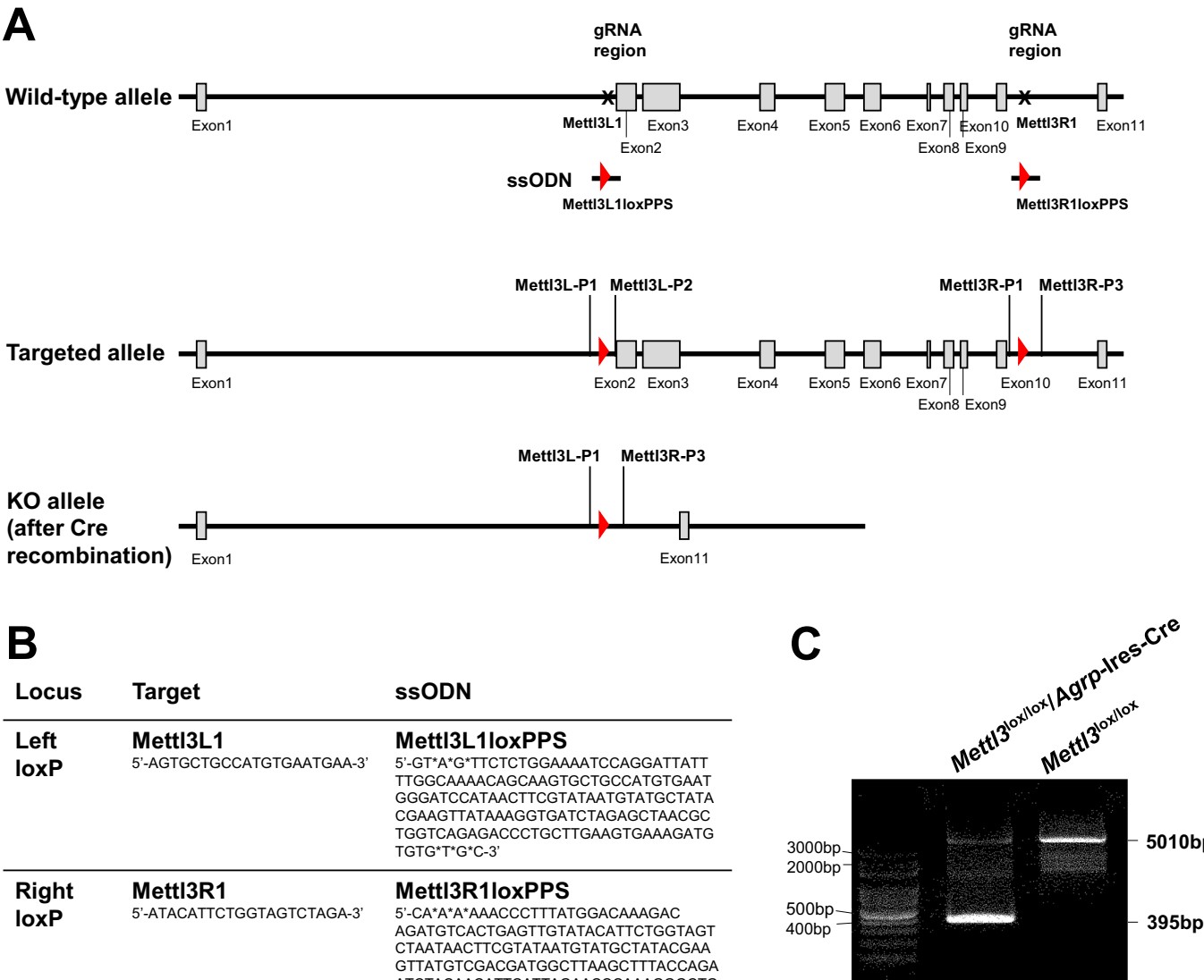

**Figure EV5. AgRP neuron-specific *Mettl3* knockout mice did not show body weight phenotype.**

(A) Schematic illustration of generation of a conditional allele at the *Mettl3* locus. Two loxP sites were inserted into *Mettl3* intron 1 and intron 10. (B) Sequences of ssODNs with 5′- and 3′-homology arms flanking loxP and a restriction site. Asterisks indicate phosphorothioate bonds. (C) Genomic DNA was extracted from the ARC, and PCR was performed using the Mettl3L-P1 and Mettl3R-P3 primers. A deletion-specific DNA fragment (395 bp) was detected in *Mettl3*[lox/lox]/*Agrp*-Ires-Cre mouse samples, whereas a non-deletion-specific DNA fragment (5010 bp) was detected in *Mettl3*[lox/lox] mouse samples.

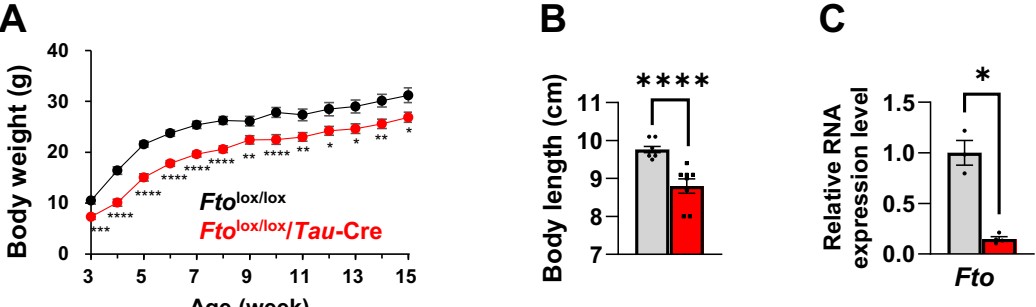

**Figure EV6. *Tau*-Cre specific *Fto*-knockout mice had reduced body weight and body length.**

Body weight ($n = 9$– 19) (**A**) and body length ($n = 8$) (**B**) of 10-week-old of *Fto*^lox/lox^/*Tau*-Cre mice were significantly lower than those of control (*Fto*^lox/lox^) mice. *Fto* mRNA expression levels in the ARC of *Fto*^lox/lox^/*Tau*-Cre mice ($n = 4$) were significantly lower than in the ARC of *Fto*^lox/lox^ mice ($n = 3$) (**C**). Error bars represent SEM. Data were analyzed using unpaired Student's *t* test (**A**) and unpaired Welch's *t* test (**B, C**); *$P < 0.05$, **$P < 0.01$, ***$P < 0.005$, ****$P < 0.001$ (exact *P* values: 0.003 [3-week-old], $6.18 \times 10^{-7}$ [4-week-old], $1.77 \times 10^{-8}$ [5-week-old], $3.37 \times 10^{-8}$ [6-week-old], $6.97 \times 10^{-7}$ [7-week-old], $8.37 \times 10^{-6}$ [8-week-old], 0.008 [9-week-old], 0.0007 [10-week-old], 0.006 [11-week-old], 0.013 [12-week-old], 0.011 [13-week-old], 0.008 [14-week-old], 0.022 [15-week-old]) (**A**); 0.0009 (**B**); 0.017 (**C**)).

**A**

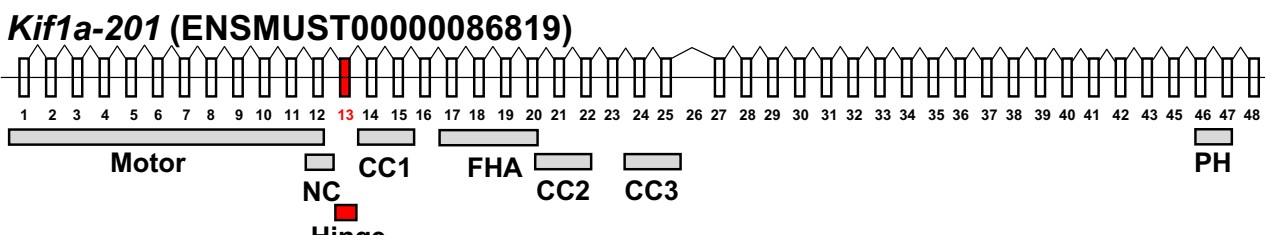

**B**

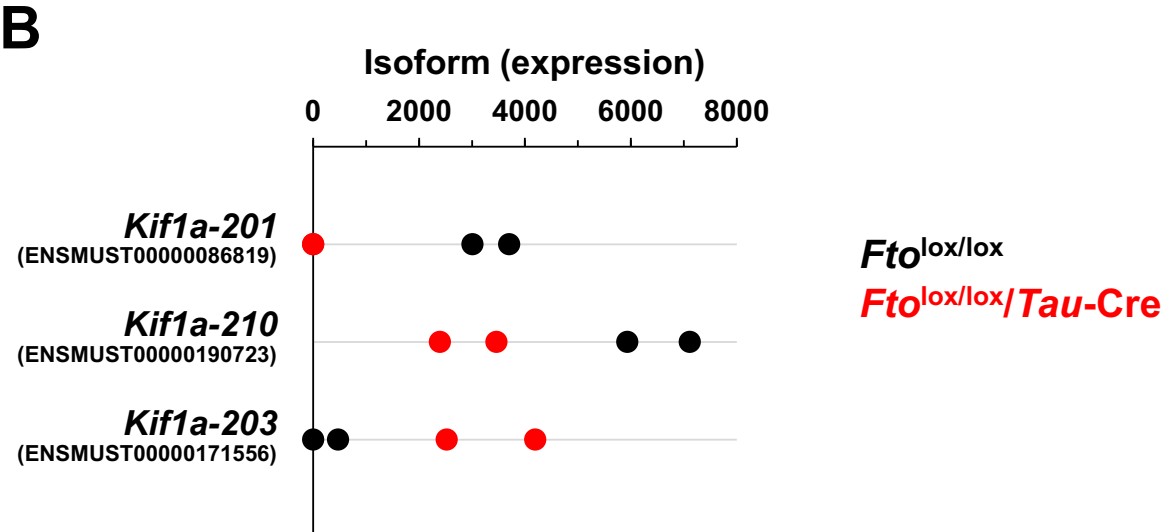

**C**

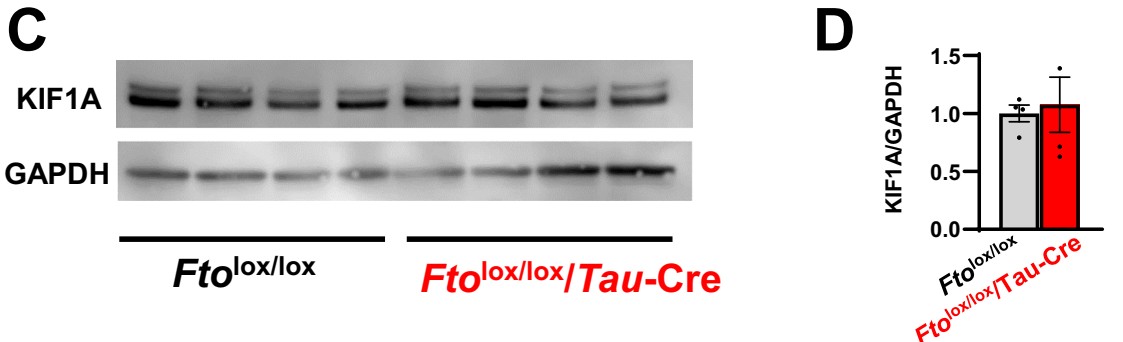

**D**

◄ **Figure EV7. Lack of FTO altered alternative splicing but not protein expression of *Kif1a*.**

(A) Alternative splicing generates cDNAs of *Kif1a* splice variants, *Kif1a-201*, *Kif1a-210*, and *Kif1a-203*. (B) Expression levels of each splice variant in the mediobasal hypothalamus of *Fto*^lox/lox^ (black) and *Fto*^lox/lox^/*Tau*-Cre (red) mice. Two pooled samples were analyzed for each genotype. (C, D) Western blotting of KIF1A in the mediobasal hypothalamus of *Fto*^lox/lox^ and *Fto*^lox/lox^/*Tau*-Cre mice (C). There was no significant difference in the KIF1A protein levels between *Fto*^lox/lox^ ($n = 4$) and *Fto*^lox/lox^/ *Tau*-Cre ($n = 4$) mice (D). Error bars represent SEM. Data were analyzed using unpaired Student's *t* test.

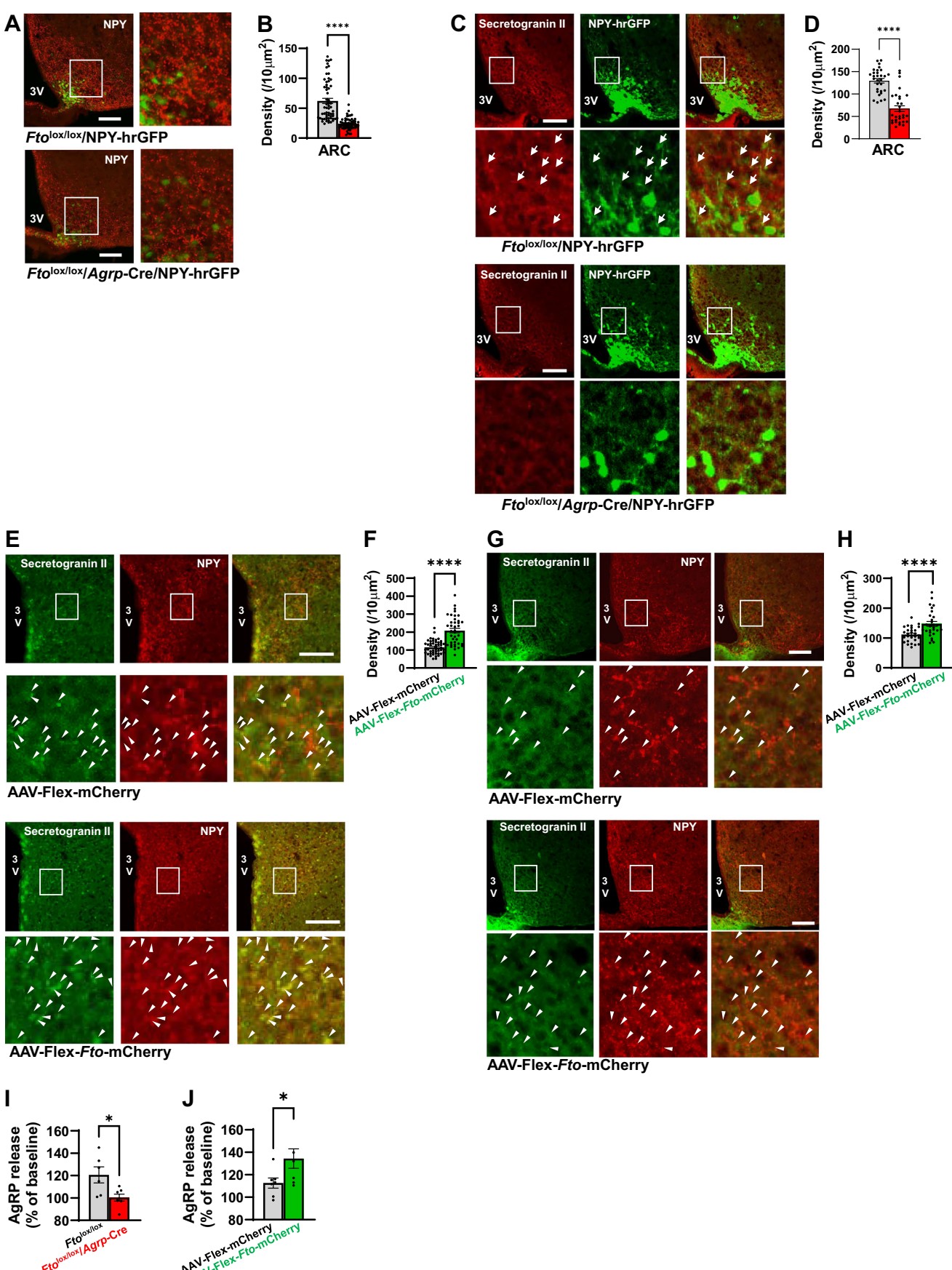

◀ **Figure EV8.   FTO is indispensable for the axonal transport of dense-core vesicles (DCV) containing NPY and AgRP.**

(**A**) NPY immunofluorescence (red) and the cell bodies of NPY/AgRP neurons visualized using NPY-hrGFP (green). (**B**) Density of NPY immunofluorescence in the ARC of $Fto^{lox/lox}$/NPY-hrGFP (gray, $n = 60$) and $Fto^{lox/lox}$/$Agrp$-Cre/NPY-hrGFP mice (red, $n = 60$). Error bars represent SEM. Data were analyzed using unpaired Welch's $t$ test; ****$P < 0.001$ (exact $P$ value: $2.72 \times 10^{-12}$). (**C, D**) Immunofluorescence of secretogranin II (red) and NPY-hrGFP fluorescence (green) in the ARC (**C**). Density of secretogranin II immunofluorescence in the ARC of $Fto^{lox/lox}$/NPY-hrGFP (gray, $n = 30$) and $Fto^{lox/lox}$/$Agrp$-Cre/NPY-hrGFP (red, $n = 30$) mice (**D**). Error bars represent SEM. Data were analyzed using unpaired Student's $t$ test; ****$P < 0.001$ (exact $P$ value: $4.83 \times 10^{-10}$). (**E–H**) Immunofluorescence of secretogranin II (green) and NPY (red) in the PVH (**E**) and ARC (**G**) of $Agrp$-Ires-Cre mice injected with AAV-hSyn-Flex-mCherry or AAV-hSyn-Flex-$Fto$-mCherry. Density of secretogranin II immunofluorescence in the PVH (**F**, $n = 48$ sites from three mice (AAV-hSyn-flex-mCherry) and 39 sites from three mice (AAV-hSyn-Flex-$Fto$-mCherry)) and ARC (**G**, $n = 28$ sites from three mice per AAV) of $Agrp$-Ires-Cre mice injected with AAV-hSyn-Flex-mCherry or AAV-Flex-$Fto$-mCherry. Error bars represent SEM. Data were analyzed using unpaired Welch's $t$ test; ****$P < 0.001$ (exact $P$ values: $1.56 \times 10^{-8}$ (**F**), 0.0006 (**H**)). (**I, J**) AgRP release from brain slices of $Fto^{lox/lox}$ ($n = 6$) and $Fto^{lox/lox}$/$Agrp$-Cre/NPY-hrGFP ($n = 7$) mice (**I**) and of $Agrp$-Ires-Cre mice injected with AAV-hSyn-Flex-mCherry (gray, $n = 7$) or AAV-hSyn-Flex-$Fto$-mCherry (green, $n = 7$) (**J**) in response to a solution containing a low concentration (2.5 mM) of glucose and 100 μM glutamate. Scale bar: 100 μm. Error bars represent SEM. Data were analyzed using unpaired Student's $t$ test (**J**); *$P < 0.05$ (exact $P$ values: 0.019 (**I**), 0.044 (**J**)).

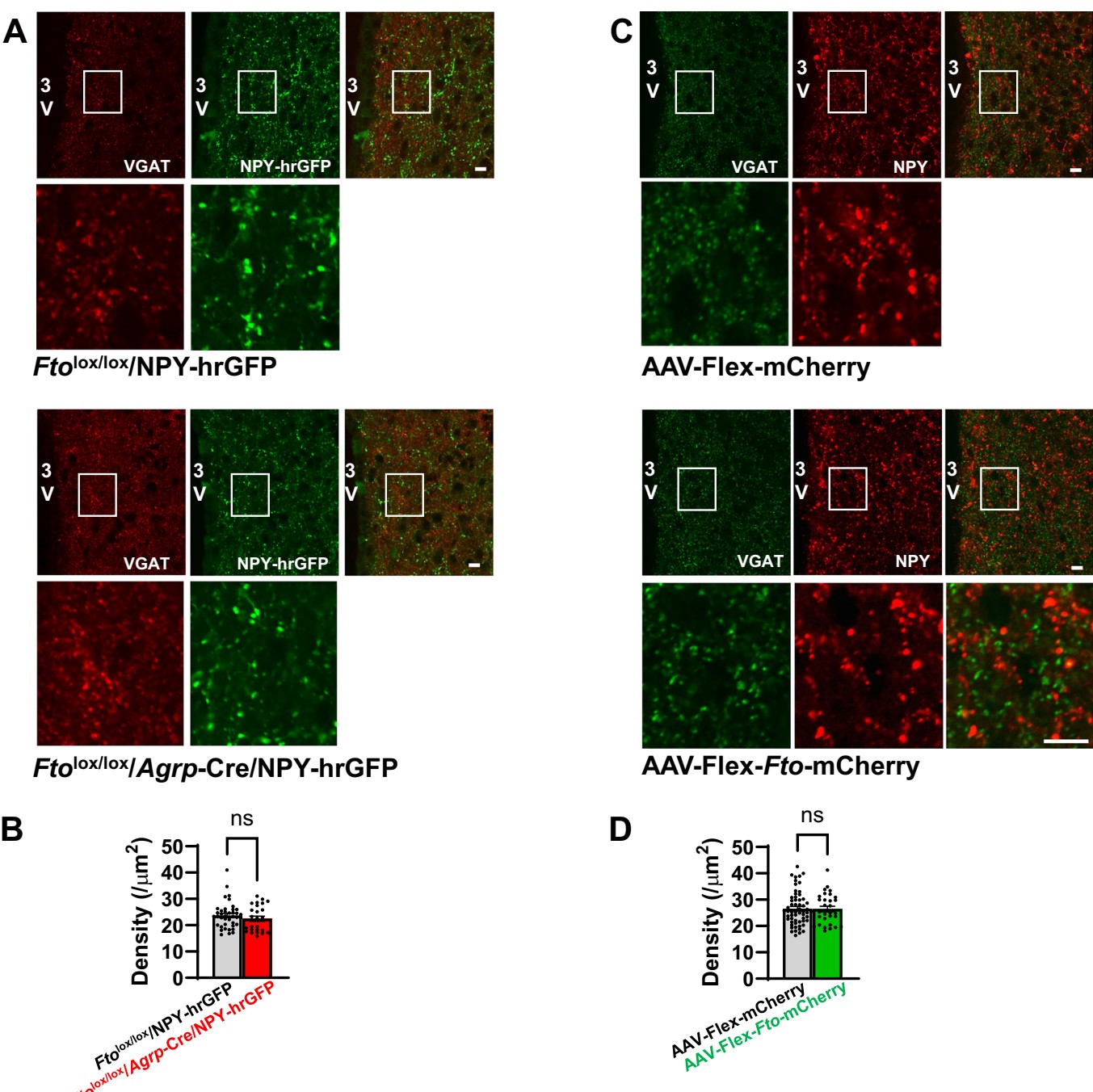

**Figure EV9. FTO does not alter vesicular GABA transporter (VGAT) density in NPY/AgRP fibers in the PVH.**

(A) VGAT immunofluorescence (red) and NPY-hrGFP (green) in the PVH of $Fto^{lox/lox}$/NPY-hrGFP and $Fto^{lox/lox}$/$Agrp$-Cre/NPY-hrGFP mice. Scale bar: 10 µm. (B) Density of VGAT immunofluorescence in the PVH adjacent to the NPY fibers of $Fto^{lox/lox}$/NPY-hrGFP (gray, $n = 3$ mice, 40 areas) and $Fto^{lox/lox}$/$Agrp$-Cre/NPY-hrGFP (red, $n = 3$ mice, 26 areas) mice. Error bars represent SEM. Data were analyzed using unpaired Student's $t$ test. (C) VGAT immunofluorescence (green) and NPY immunofluorescence (red) in the PVH of $Agrp$-Ires-Cre mice injected with AAV-hSyn-Flex-mCherry and AAV-hSyn-Flex-$Fto$-mCherry. Scale bar: 10 µm. (D) Density of VGAT immunofluorescence in the PVH adjacent to NPY fibers in the PVH of $Agrp$-Ires-Cre mice injected with AAV-hSyn-Flex-mCherry (gray, $n = 3$ mice, 59 sites) or AAV-hSyn-Flex-$Fto$-mCherry (green, $n = 3$ mice, 30 sites). Error bars represent SEM. Data were analyzed using unpaired Student's $t$ test.

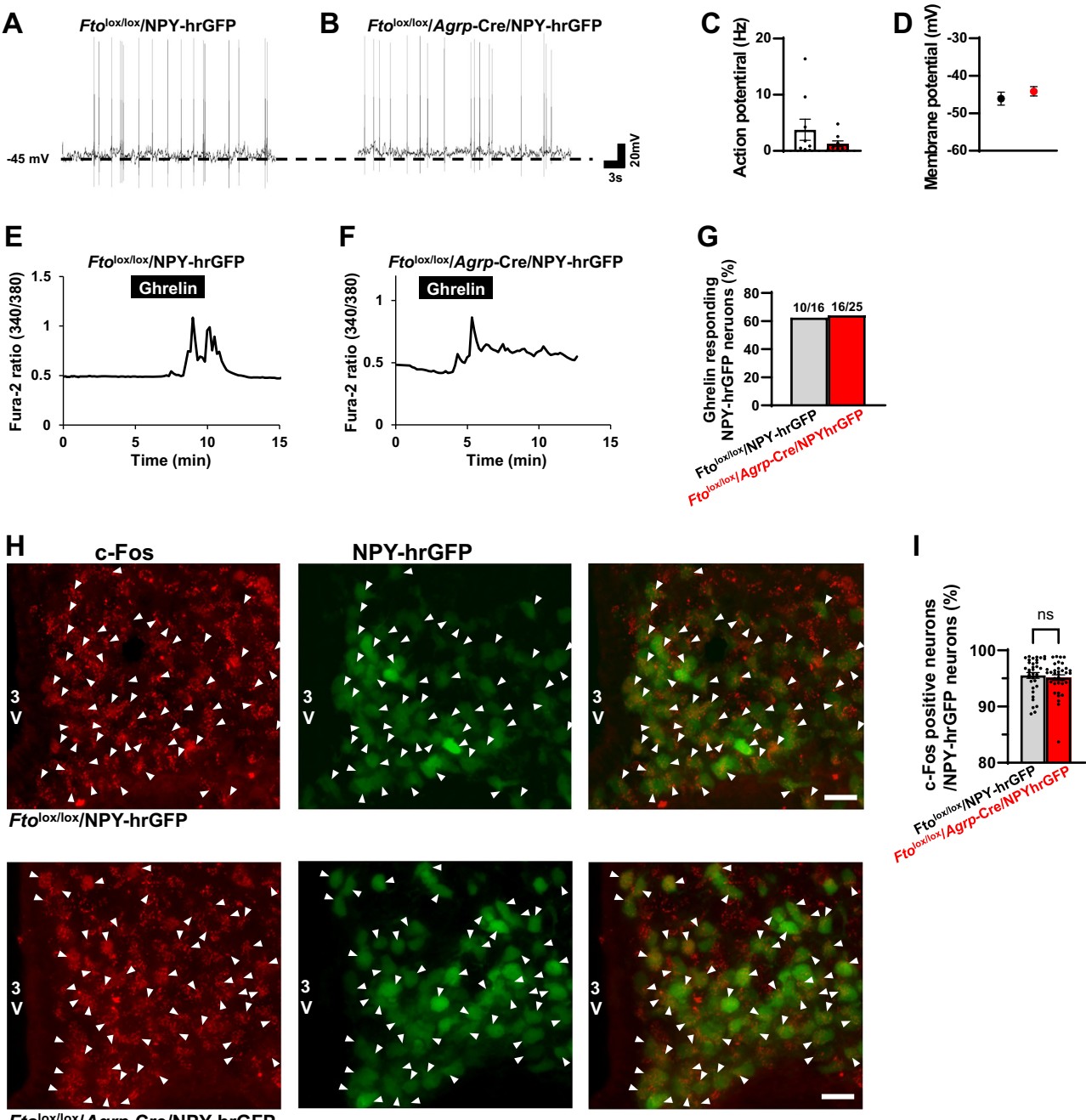

**Figure EV10. Cellular activity of NPY/AgRP neurons lacking FTO is normal.**

(A–D) Whole-cell patch-clamp recordings of AgRP neurons identified by NPY-hrGFP fluorescence from *Fto*lox/lox/NPY-hrGFP (A) and *Fto*lox/lox/*AgRP*-Cre/NPY-hrGFP mice (B). There were no significant differences in the action potential (C) or membrane potential (D) between *Fto*lox/lox/NPY-hrGFP (*n* = 9) and *Fto*lox/lox/*AgRP*-Cre/NPY-hrGFP (red) mice (*n* = 9). Error bars represent SEM. (E–G) Fura-2 calcium imaging of isolated NPY-hrGFP neurons was performed. Representative Fura-2 ratio traces of NPY-hrGFP neurons from *Fto*lox/lox/NPY-hrGFP (E) and *Fto*lox/lox/*AgRP*-Cre/NPY-hrGFP mice subjected to ghrelin at $10^{-10}$ M. (G) The percentage of NPY-hrGFP neurons responded to ghrelin. The number above each bar indicates the number of NPY-hrGFP neurons responded to ghrelin over the number of NPY-hrGFP neurons analyzed. (H, I) c-Fos-immunofluorescence (red) in NPY-hrGFP (green) neurons of overnight-fasted *Fto*lox/lox/NPY-hrGFP and *Fto*lox/lox/*AgRP*-Cre-NPY-hrGFP mice (H). Arrowheads indicate neurons exhibiting both c-Fos and NPY-hrGFP. Scale bar. 20 μm. The percentage of c-Fos-positive neurons among NPY-hrGFP neurons in *Fto*lox/lox/NPY-hrGFP (*n* = 36 unilateral sections from three mice) and *Fto*lox/lox/*AgRP*-Cre-NPY-hrGFP (*n* = 35 unilateral sections from three mice) mice (I). Error bars represent SEM. Data were analyzed using unpaired Student's *t* test; exact *P* value: 0.566.

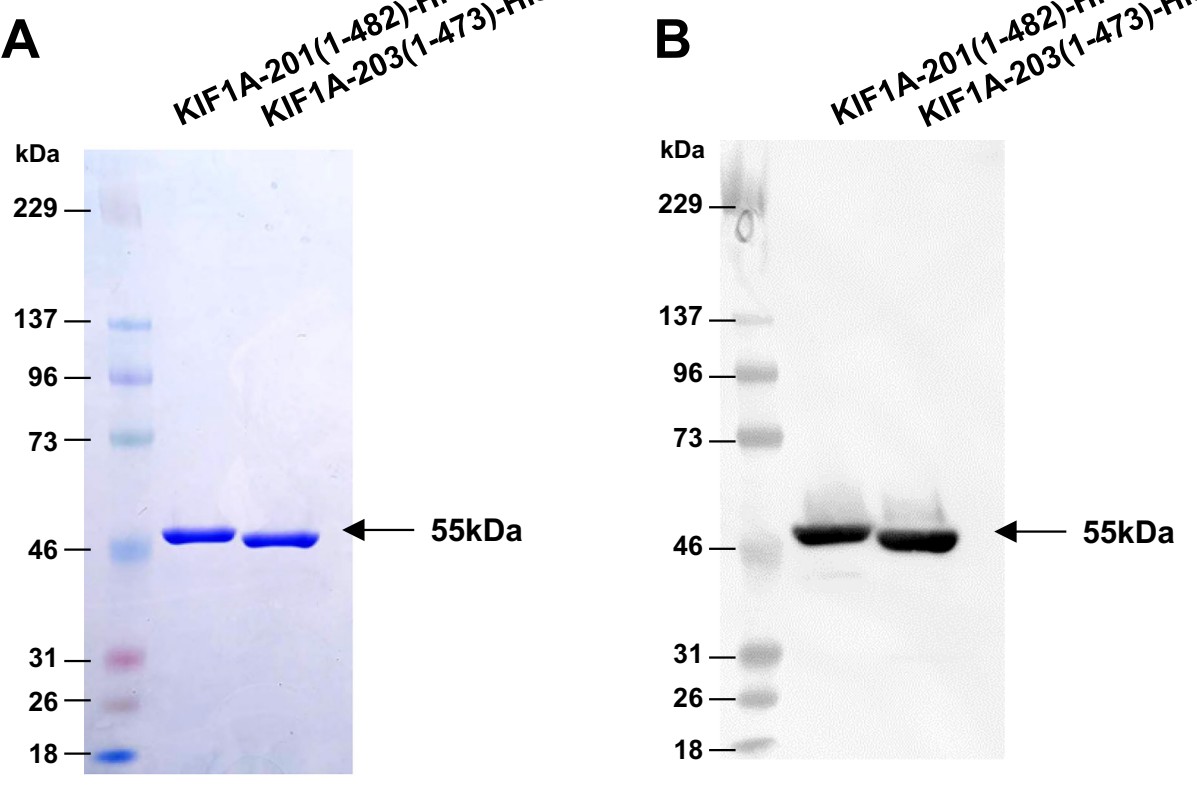

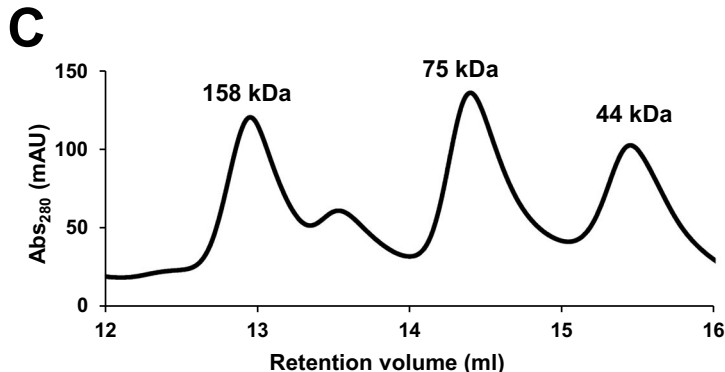

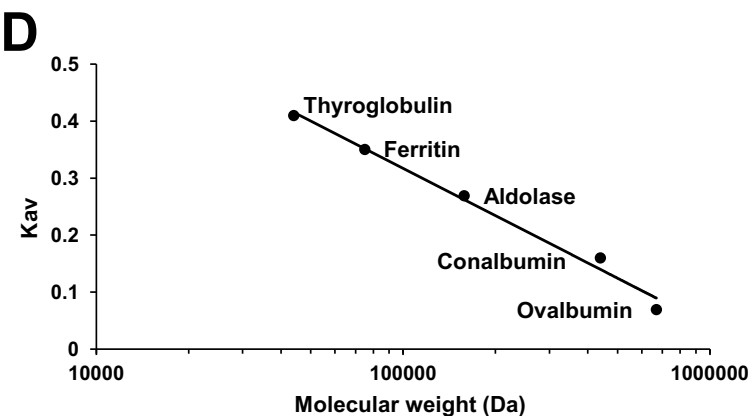

**Figure EV11.   Validation of recombinant KIF1A proteins, and molecular weight determination by size-exclusion chromatography.**

(A, B) Validation of recombinant KIF1A proteins. Recombinant KIF1A-201(1-482)-His and KIF1A-203(1-473)-His proteins (2.7 μg each) were analyzed by SDS-PAGE. The gel was stained with Coomassie Brilliant Blue R-250 (A). Following SDS-PAGE, proteins were transferred onto a membrane for western blotting using anti-KIF1A antibody (B). A protein molecular weight marker image, captured under white light exposure, was overlaid on the western blotting image. These analyses confirmed that the KIF1A proteins were highly purified. (C, D) Molecular weight determination by size-exclusion chromatography. Retention volume of standard globular proteins in size-exclusion chromatography (C). The calibration curve was generated by plotting the Kav value for each standard against its corresponding molecular weight, and used to determine the molecular weight of the KIF1A fragments (D).

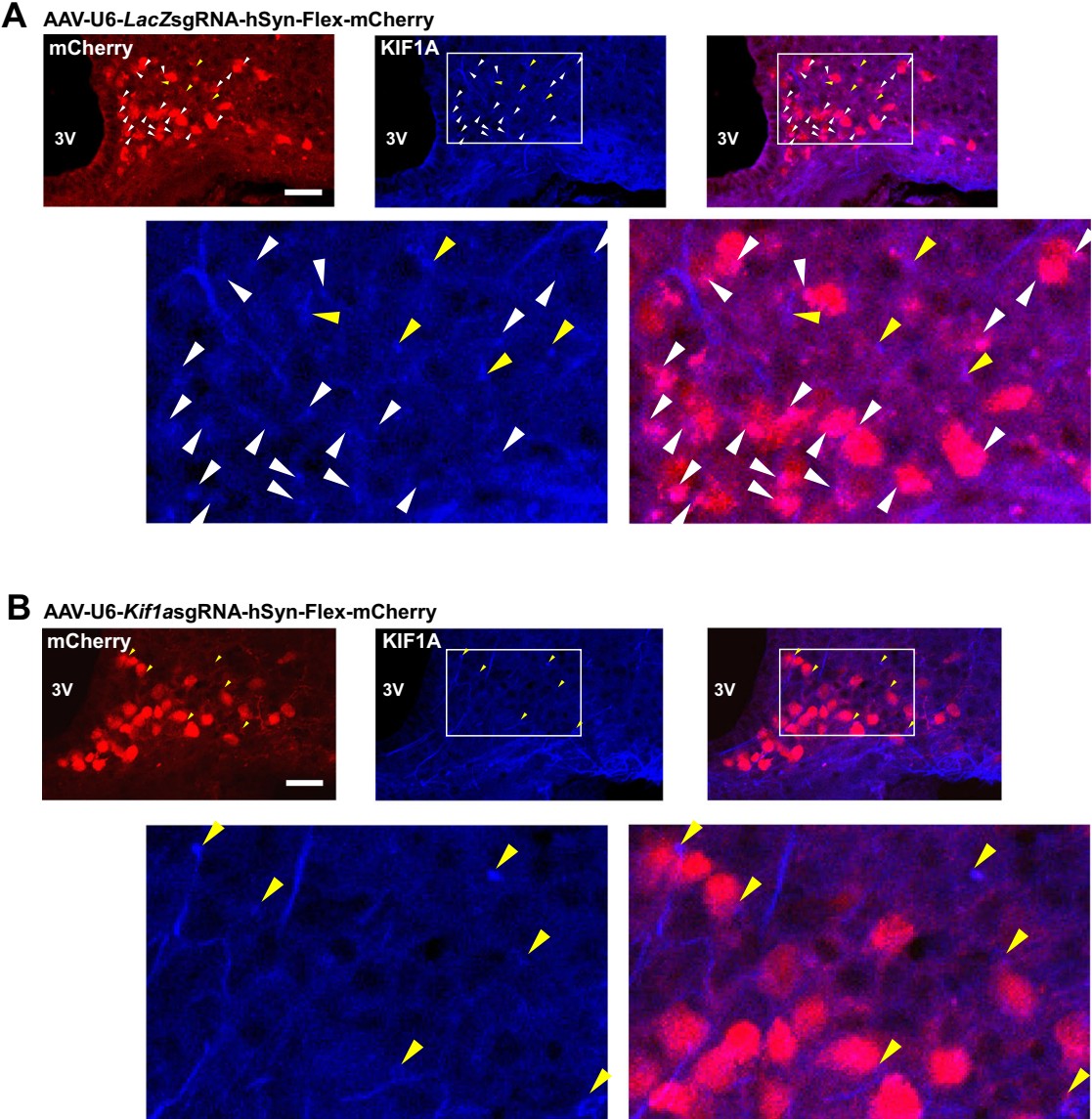

**Figure EV12. Validation of KIF1A knockdown in AgRP neuron-specific *Kif1a* knockdown mouse.**

AAV-U6-*Kif1a*sgRNA-hSyn-Flex-mCherry (A) or AAV-U6-LacZsgRNA-hSyn-Flex-mCherry (B) was injected into the ARC of Rosa26-LSL-Cas9 knock-in/*Agrp*-Ires-Cre mice. The mCherry expression (red), KIF1A immunofluorescence (blue), and merged images are shown. White arrowheads indicate neurons exhibiting both types of fluorescence, while yellow arrowheads indicate neurons exhibiting KIF1A fluorescence only. Scale bar: 30 μm.

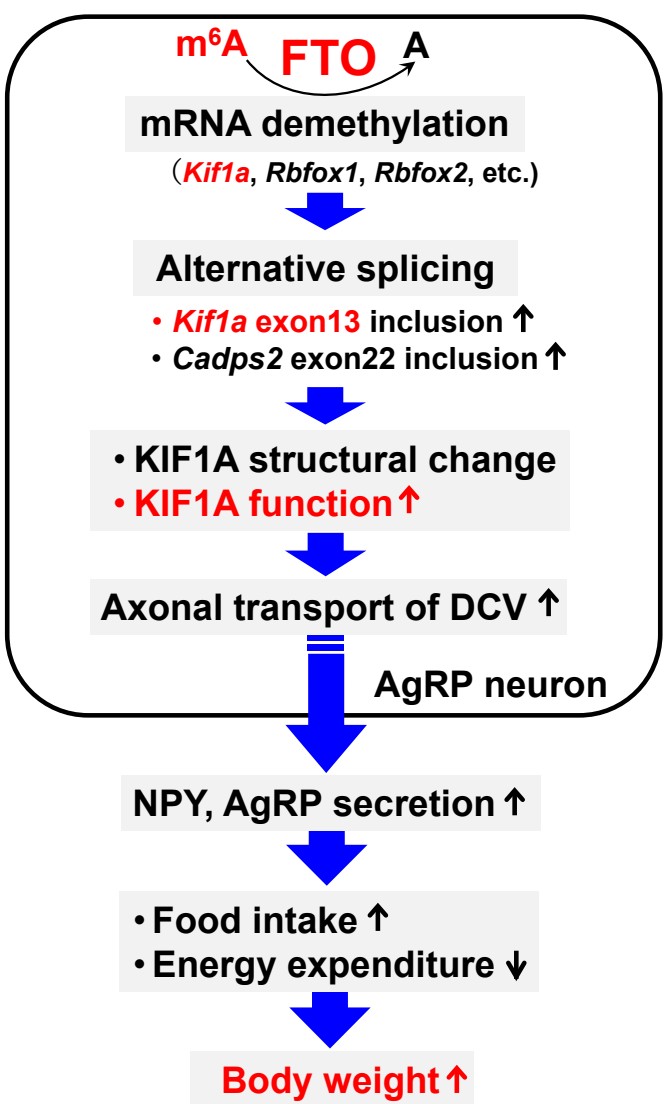

**Figure EV13. Schematic representation of the role of FTO in AgRP neurons.**

FTO in AgRP neurons controls body weight. In AgRP neurons, FTO demethylates mRNAs associated with membrane trafficking and alternative splicing, including *Kif1a*, *Rbfox1*, and *Rbfox2*. Then, alternative splicing of *Kif1a* exon 13 inclusion and *Cadps2* exon 22 inclusion is upregulated. *Kif1a* exon 13 inclusion alters the protein structure of KIF1A and reinforces KIF1A function. As a result, FTO enhances the axonal transport of DCVs and the secretion of NPY and AgRP, thereby increasing food intake, decreasing energy expenditure, and increasing body weight.

