## [Peer Review File · The EMBO Journal]

FTO promotes weight gain via altering Kif1a splicing and axonal vesicle trafficking in AgRP neurons

Daisuke Kohno, Reika Kawabata-Iwakawa, Sotaro Ichinose, Shigetomo Suyama, Kazuto Ohashi, Winda Ariyani, Tetsushi Sadakata, Hiromi Yokota-Hashimoto, Ryosuke Kobayashi, Takuro Horii, Vina Susanti, Ayumu Konno, Haruka Tsuneoka, Chiharu Yoshikawa, Sho Matsui, Akihiro Harada, Toshihiko Yada, Izuho Hatada, Hirokazu Hirai, Masahiko Nishiyama, Tsutomu Sasaki, and Tadahiro Kitamura

Corresponding author: Daisuke Kohno (daisuke.kohno@gunma-u.ac.jp)

Review Timeline:

Submission Date:	5th Nov 24
Editorial Decision:	5th Dec 24
Revision Received:	30th Apr 25
Editorial Decision:	21st May 25
Revision Received:	26th May 25
Accepted:	26th Jun 25

Editor: Daniel Klimmeck

Transaction Report:

Dear Dr Kohno,

Thank you again for the submission of your manuscript (EMBOJ-2024-119533) to The EMBO Journal. As mentioned earlier, your study was assessed by three reviewers with expertise in central control of body metabolism and neuroscience, whose comments are enclosed below.

As you will see from the experts' reports, the referees acknowledge the analysis and potential interest and value of your findings. However, they also express important issues regarding the completeness of your study, which need to be addressed thoroughly to make them supportive of publication in the EMBO Journal. Further, the reviewers raise a number of issues related to the presentation of the findings, additional controls and improved methods annotation required, statistics applied and overall discussion of related literature, that would need to be conclusively addressed to achieve the level of robustness and clarity needed for The EMBO Journal.

Given the overall interest stated and broader angle of your findings, we are able to invite you to revise your manuscript experimentally to address the referees' comments. I need to stress though that we do require strong support from the referees on a revised version of the study in order to move on to publication of the work.

I would appreciate if you could contact me during the next weeks for exchange e.g. a video call to discuss your perspective on the comments and potential plan for revisions.

Please feel free to contact me if you have any questions or need further input on the referee comments.

When submitting your revised manuscript, please carefully review the instructions below.

Please feel free to approach me any time should you have additional questions related to this.

Thank you for the opportunity to consider your work for publication.

I look forward to your revision.

Kind regards,

Daniel Klimmeck

Daniel Klimmeck, PhD
Senior Editor
The EMBO Journal

Instruction for the preparation of your revised manuscript:

- 1) a .docx formatted version of the manuscript text (including legends for main figures, EV figures and tables). Please make sure that the changes are highlighted to be clearly visible.
- 2) individual production quality figure files as .eps, .tif, .jpg (one file per figure).
- 3) a .docx formatted letter INCLUDING the reviewers' reports and your detailed point-by-point response to their comments. As part of the EMBO Press transparent editorial process, the point-by-point response is part of the Review Process File (RPF), which will be published alongside your paper.
- 4) a complete author checklist, which you can download from our author guidelines ([https://wol-prod-cdn.literatumonline.com/pb-assets/embo-site/Author Checklist%20-%20EMBO%20J-1561436015657.xlsx](https://wol-prod-cdn.literatumonline.com/pb-assets/embo-site/Author%20Checklist%20-%20EMBO%20J-1561436015657.xlsx)). Please insert information in the checklist that is also reflected in the manuscript. The completed author checklist will also be part of the RPF.

6) It is mandatory to include a 'Data Availability' section after the Materials and Methods. Before submitting your revision, primary datasets produced in this study need to be deposited in an appropriate public database, and the accession numbers and database listed under 'Data Availability'. Please remember to provide a reviewer password if the datasets are not yet public (see <https://www.embopress.org/page/journal/14602075/authorguide#datadeposition>).

7) Our journal encourages inclusion of *data citations in the reference list* to directly cite datasets that were re-used and obtained from public databases. Data citations in the article text are distinct from normal bibliographical citations and should directly link to the database records from which the data can be accessed. In the main text, data citations are formatted as follows: "Data ref: Smith et al, 2001" or "Data ref: NCBI Sequence Read Archive PRJNA342805, 2017". In the Reference list, data citations must be labeled with "[DATASET]". A data reference must provide the database name, accession number/identifiers and a resolvable link to the landing page from which the data can be accessed at the end of the reference. Further instructions are available at .

8) At EMBO Press we ask authors to provide source data for the main and EV figures. Our source data coordinator will contact you to discuss which figure panels we would need source data for and will also provide you with helpful tips on how to upload and organize the files.

Numerical data can be provided as individual .xls or .csv files (including a tab describing the data). For 'blots' or microscopy, uncropped images should be submitted (using a zip archive or a single pdf per main figure if multiple images need to be supplied for one panel). Additional information on source data and instruction on how to label the files are available at .

9) We replaced Supplementary Information with Expanded View (EV) Figures and Tables that are collapsible/expandable online (see examples in <https://www.embopress.org/doi/10.15252/embj.201695874>). A maximum of 5 EV Figures can be typeset. EV Figures should be cited as 'Figure EV1, Figure EV2" etc. in the text and their respective legends should be included in the main text after the legends of regular figures.

11) For data quantification: please specify the name of the statistical test used to generate error bars and P values, the number (n) of independent experiments (specify technical or biological replicates) underlying each data point and the test used to calculate p-values in each figure legend. The figure legends should contain a basic description of n, P and the test applied. Graphs must include a description of the bars and the error bars (s.d., s.e.m.).

We realize that it is difficult to revise to a specific deadline. In the interest of protecting the conceptual advance provided by the work, we recommend a revision within 3 months (5th Mar 2025). Please discuss the revision progress ahead of this time with the editor if you require more time to complete the revisions.

Referee #1:

In this study, the authors investigated the role of FTO-regulated pathways in the AgRP neurons and identified a novel function of FTO. AgRP neuron-specific deletion of FTO resulted in lean phenotype, while AgRP neuron-specific overexpression of FTO led to obesity. Changes of food intake were largely responsible for the body weight phenotypes. The authors further explored the molecular mechanisms and identified alternative splicing of Kif1A as the key to the observed phenotype. The authors also presented evidence that transport of NPY/AgRP vesicle was altered in the deletion and overexpression models. Finally, the authors demonstrated that the FTO-Kif1A pathway works in the AgRP neurons in response to changes in feeding status. Overall, this is a very interesting study addressing a very important question in this field. Only minor issues need to be addressed as specified below.

1. Page 3, line 125, the authors mention nesfatin, but this is not shown in Figure 1. On the other hand, NUCB2 is shown in Figure 1, but is not mentioned in the text. Please address this issue so that the results describe what is presented in Figures.
2. Extended Data Fig. 1, the authors need to present scale bars in panels A, C, D-G, J-O.
3. Extended data Fig. 1E, FTO and POMC co-localizes in only a few cells, which does not match what is described in line 123 (60 %) and panel H in the same figure. Please show an image that better represents the statistics.
4. Page 4, lines 151-153, the authors mentioned that AgRP neurons control the sympathetic nervous system. Did the authors measure in the *Ftlox/lox::Agrp-Cre* mice and compare that to measurements from the *Ftlox/lox* mice? This data would further support the observed phenotype.
5. Extended Data Fig. 7E-H, the authors reported comparable NPY/AgRP neuron activity between genotypes. Is different axonal transport sufficient to result in differences in release when the neuronal activity is not different? What is the authors' explanation regarding this issue?

Referee #2:

Kohno et al. performed an extensive evaluation, to determine the cell type-specific role of FTO in hypothalamic neurons in the control of energy balance. The author found that FTO is highly expressed in several hypothalamic nuclei, and by screening several mouse lines, they reveal that only knockout of FTO in AgRP hunger neurons resembles the previously described phenotype observed in full body or brain deletion of this gene linked to human obesity. Specifically, they found that FTO knockout from AgRP neurons diminished body weight gain, which was primarily caused by reduced food intake. Conversely, AAV-based overexpression of FTO in AgRP neurons increased food intake, diminished energy expenditure, and caused obesity. Remarkably, through genetic screens, the authors uncovered that FTO expression is linked to expression of membrane trafficking proteins, including those involved in axonal transport. Based on this, they demonstrated that FTO deletion decreases projection density, DVC number, and neuropeptide release from AgRP neurons. In addition, they showed that FTO expression in AgRP is altered by the energy state.

Overall, this is a very intriguing study, which provides in-depth insights into the mechanisms of FTO in regulating feeding neurocircuits. I find the aspect of FTO-mediated alterations in neuropeptide release particularly novel and exciting. The paper is suitable for publication, pending the authors resolve the following points.

Major concerns:

1. In addition to the neuropeptides AgRP and NPY, AgRP neurons also communicate with downstream neurons through the synaptic release of GABA. It would be important to show whether and how FTO affects this fast-acting neurotransmitter, which is released from synaptic vesicles of AgRP neuron terminals, and has been shown to play a major role in energy balance (e.g., PMID: 19160495)
2. To complement the finding that FTO knockout decreases DVC number and neuropeptide release from AgRP neurons, it would be greatly appreciated if the authors could determine whether FTO overexpression increases these parameters. This

would further substantiate the bidirectional function of FTO in AgRP neurons.

3. Based on the findings presented in Fig. 8, which nicely demonstrate the physiological regulation of FTO expression, I would encourage the authors to test how FTO deletion affects neuropeptide/DVC number in AgRP neuron terminals upon fasting and refeeding. This would allow to causally link FTO expression to energy state-dependent control of feeding neurocircuits.

Minor concerns:

1. Analysis of FTO co-expression in AgRP neurons in control and knockout mice (Extended Data Fig. 2D) should be included.

2. In Fig. 5 L and M, it appears that the controls have a strikingly different release probability of AgRP. How do the authors explain this?

3. Action potential firing frequency of AgRP neurons in FTO knockout mice appears to be dramatically lower. Although the authors describe that this is not significantly different, it could be that this is due to the higher variability and/or low number of recorded neurons. I would recommend increasing the number of recorded neurons. Alternatively, the authors could consider determining Fos expression in AgRP neurons (e.g., upon fasting), in control and FTO knockout mice, as this has been widely used as an indicator of their activity regulation.

4. It is not entirely clear how the authors come up with the idea that axonal transport could be altered. A more detailed explanation of the idea and hypothesis would make it easier to follow this important point.

Editorial comments:

1. The paraventricular hypothalamus is generally not accepted as a nucleus of the mediobasal hypothalamus. Only the ARC, and sometimes the VMH, are part of this area. Please adjust.

2. Given that only some parameters of systemic metabolism were assessed, I would suggest changing "Metabolism" to "Energy expenditure" in the Extended Data Fig. 10.

3. The abstract is difficult to read as it contains many abbreviations. I would suggest reducing them to 3, maximum 4.

4. There are several grammar, spelling, and typo errors throughout the text. I would highly recommend proof reading this again.

Referee #3:

In this study, Kohno et al. investigate the role of FTO in hypothalamic AgRP neurons. Using a comprehensive methodological approach, the authors demonstrate that deletion or overexpression of FTO in AgRP neurons decreases or increases body weight, respectively, primarily due to changes in food intake. FTO was shown to demethylate m6A modifications on exons and introns of pre-mRNA encoding proteins involved in membrane trafficking and alternative splicing. Notably, FTO modulated the splicing of the Kif1a gene by enhancing the inclusion of exon 13, thereby affecting the dimerization and function of this protein. In this context, FTO increased dense-core vesicle trafficking and the release of AgRP. Finally, knockdown of Kif1a in AgRP neurons overexpressing FTO led to reduced body weight.

This is an interesting study that nicely combines physiological insights with mechanistic exploration of an under-investigated topic (epitranscriptomic modifications), which could have implications for the field of central metabolic control. Overall, the experiments are conducted to a high standard, and the manuscript is well-written. However, I have several concerns and suggestions that the authors should address:

1. While the authors use methodologies that allow specific targeting of AgRP neurons in many parts of the study, in some key sections (e.g., analysis of splice variants), they employ more general approaches (e.g., Tau-Cre mice), which target neurons broadly rather than specifically targeting AgRP neurons. The authors should explicitly state in the text why they transitioned from AgRP-specific to broader neuron-targeting strategies. If this shift is due to limitations in sensitivity or other constraints, this should be clarified. Furthermore, this reasoning should be extended to other parts of the manuscript that do not implement neuron-specific approaches. The authors should also temper the interpretation of data obtained from general neuronal or cell culture experiments, avoiding direct extrapolations to AgRP neurons.

2. The immunofluorescence images throughout the manuscript are of poor quality. In many cases, the images appear overexposed, making it difficult (or even impossible) to discern the staining. Higher-quality, better-balanced images should be provided.

3. Did the authors include an IgG control for the m6A immunoprecipitation experiments?

4. Are m6A sites enriched around exon 13 of Kif1a? Is the regulation of exon 13 inclusion a direct effect of FTO? The authors should consider commenting on this aspect in the manuscript.

5. In Figure 3E, the authors integrate RNA-seq with m6A-seq data. This integration should be more clearly explained in the text

to aid reader understanding.

6. The electrophysiology experiments (Extended Data Fig. 7E-H) were conducted under basal conditions, which may explain why no changes were observed. Did the authors attempt to stimulate the neurons with a physiological activator (e.g., ghrelin)?

7. Similarly, in the hypothalamic explant experiments, the authors use high potassium as a proxy for total AgRP content. This allows to assess the total releasable pool of neuropeptide (a proxy of production). Have they tested AgRP secretion using physiological inducers instead?

8. Figure 8C relies on conventional PCR, which is not a quantitative technique. Using qPCR to specifically target exon 13 would provide more robust and quantitative data.

Response to the referees

We appreciate the valuable feedback of the referees on the manuscript, which has been instrumental in enhancing its overall quality. We have addressed the referees' suggestions and corrections. Detailed responses to each comment are listed below, and the authors' responses are shown in blue. The revised content of the manuscript is marked in blue.

Referee comments

Referee #1:

In this study, the authors investigated the role of FTO-regulated pathways in the AgRP neurons and identified a novel function of FTO. AgRP neuron-specific deletion of FTO resulted in lean phenotype, while AgRP neuron-specific overexpression of FTO led to obesity. Changes of food intake were largely responsible for the body weight phenotypes. The authors further explored the molecular mechanisms and identified alternative splicing of Kif1A as the key to the observed phenotype. The authors also presented evidence that transport of NPY/AgRP vesicle was altered in the deletion and overexpression models. Finally, the authors demonstrated that the FTO-Kif1A pathway works in the AgRP neurons in response to changes in feeding status. Overall, this is a very interesting study addressing a very important question in this field. Only minor issues need to be addressed as specified below.

1. Page 3, line 125, the authors mention nesfatin, but this is not shown in Figure 1. On the other hand, NUCB2 is shown in Figure 1, but is not mentioned in the text. Please address this issue so that the results describe what is presented in Figures.

Authors' response:

Thank you for your appreciation of our research and pointing out our errors. In response to the referee's comment, the term "Nesfatin" was replaced by "NUCB2" in line 134.

2. Extended Data Fig. 1, the authors need to present scale bars in panels A, C, D-G, J-O.

Authors' response:

In response to the referee's comment, scale bars have been added to Extended Data Fig. 1 A, C-G, J-O.

3. Extended data Fig. 1E, FTO and POMC co-localizes in only a few cells, which does not

match what is described in line 123 (60 %) and panel H in the same figure. Please show an image that better represents the statistics.

Authors' response:

In response to your comment, the image shown in Fig. EV1E was replaced with another, showing that approximately 60% of the POMC neurons express *Fto-lacZ*.

4. Page 4, lines 151-153, the authors mentioned that AgRP neurons control the sympathetic nervous system. Did the authors measure in the *Ftlox/lox::Agrp-Cre* mice and compare that to measurements from the *Ftlox/lox* mice? This data would further support the observed phenotype.

Authors' response:

In response to the referee's comment, we analyzed the involvement of the sympathetic nervous system by administering a β 3-adrenergic blocker. The β 3 blocker cancelled the reduction of RER, free fatty acids, and ketone bodies observed in *Fto^{lox/lox}/Agrp-Cre* mice, which is shown in lines 161–165 and Fig. EV3.

5. Extended Data Fig. 7E-H, the authors reported comparable NPY/AgRP neuron activity between genotypes. Is different axonal transport sufficient to result in differences in release when the neuronal activity is not different? What is the authors' explanation regarding this issue?

Authors' response:

In response to the referee's comment, we have added a discussion of the relationship between axonal transport and neurotransmitter release in lines 502–508.

Referee #2:

Kohno et al. performed an extensive evaluation, to determine the cell type-specific role of FTO in hypothalamic neurons in the control of energy balance. The author found that FTO is highly expressed in several hypothalamic nuclei, and by screening several mouse lines, they reveal that only knockout of FTO in AgRP hunger neurons resembles the previously described phenotype observed in full body or brain deletion of this gene linked to human obesity. Specifically, they found that FTO knockout from AgRP neurons diminished body weight gain, which was primarily caused by reduced food intake. Conversely, AAV-based overexpression of FTO in AgRP neurons increased food intake, diminished energy expenditure, and caused obesity. Remarkably, through genetic screens, the authors

uncovered that FTO expression is linked to expression of membrane trafficking proteins, including those involved in axonal transport. Based on this, they demonstrated that FTO deletion decreases projection density, DVC number, and neuropeptide release from AgRP neurons. In addition, they showed that FTO expression in AgRP is altered by the energy state.

Overall, this is a very intriguing study, which provides in-depth insights into the mechanisms of FTO in regulating feeding neurocircuits. I find the aspect of FTO-mediated alterations in neuropeptide release particularly novel and exciting. The paper is suitable for publication, pending the authors resolve the following points.

Major concerns:

1. In addition to the neuropeptides AgRP and NPY, AgRP neurons also communicate with downstream neurons through the synaptic release of GABA. It would be important to show whether and how FTO affects this fast-acting neurotransmitter, which is released from synaptic vesicles of AgRP neuron terminals, and has been shown to play a major role in energy balance (e.g., PMID: 19160495)

Authors' response:

Thank you for your comments. In response to the referee's comment, we have analyzed the density of VGAT, a marker for synaptic vesicles containing GABA, in NPY fibers within the PVH. The results are shown in lines 306–312 and Fig EV 9 and are discussed in lines 498–501.

2. To complement the finding that FTO knockout decreases DVC number and neuropeptide release from AgRP neurons, it would be greatly appreciated if the authors could determine whether FTO overexpression increases these parameters. This would further substantiate the bidirectional function of FTO in AgRP neurons.

Authors' response:

Thank you for your important comments. In response to the referee's comment, we performed Secretogranin II immunohistochemistry on *Fto*-overexpressing mice, and the results are presented in Fig. EV8E–H and described in lines 295–297.

3. Based on the findings presented in Fig. 8, which nicely demonstrate the physiological regulation of FTO expression, I would encourage the authors to test how FTO deletion affects neuropeptide/DVC number in AgRP neuron terminals upon fasting and refeeding. This would allow to causally link FTO expression to energy state-dependent control of

feeding neurocircuits.

Authors' response:

Thank you for your valuable comments. In response to the referee's suggestion, Secretogranin II immunohistochemistry was performed on fasting and refeeding mice; the results are shown in Fig. 8F–G and described in lines 398–402.

Minor concerns:

1. Analysis of FTO co-expression in AgRP neurons in control and knockout mice (Extended Data Fig. 2D) should be included.

Authors' response:

In response to the referee's comment, the analysis of FTO co-expression in AgRP neurons in control and knockout mice is included in the main figure (Fig.1A).

2. In Fig. 5 L and M, it appears that the controls have a strikingly different release probability of AgRP. How do the authors explain this?

Authors' response:

Thank you for pointing out this important issue. We speculate that this discrepancy could be due to differences in mouse age. To address this, we re-measured the AgRP release from 10 week-old of *Fto*^{lox/lox} and *Fto*^{lox/lox}/*Agrp-Cre* mice (Fig.5L), matching the ages of *Fto*-overexpressing and control mice in Fig.5M. The new data are now presented in Fig.5L.

3. Action potential firing frequency of AgRP neurons in FTO knockout mice appears to be dramatically lower. Although the authors describe that this is not significantly different, it could be that this is due to the higher variability and/or low number of recorded neurons. I would recommend increasing the number of recorded neurons. Alternatively, the authors could consider determining Fos expression in AgRP neurons (e.g., upon fasting), in control and FTO knockout mice, as this has been widely used as an indicator of their activity regulation.

Authors' response:

In response to the referee's comment, we compared c-Fos expression in AgRP neurons under overnight fasting conditions between *Fto* knockout and control mice. In addition, we assessed the percentage of NPY-hrGFP neurons that showed an increase in intracellular calcium concentration in response to ghrelin in both groups. These results are presented in Fig. EV 10 E–I and are described in lines 317–324.

4. It is not entirely clear how the authors come up with the idea that axonal transport could be altered. A more detailed explanation of the idea and hypothesis would make it easier to follow this important point.

Authors' response:

In response to the referee's comment, we have added an explanation for shifting the focus of this study to axonal transport, as described in lines 280–284. Additionally, we have provided the names of the membrane trafficking genes enriched in Fig.4B in Lines 271–273.

Editorial comments:

1. The paraventricular hypothalamus is generally not accepted as a nucleus of the mediobasal hypothalamus. Only the ARC, and sometimes the VMH, are part of this area. Please adjust.

Authors' response:

In response to the referee's comment, "Mediobasal hypothalamus" was replaced by "hypothalamic nuclei of the feeding center" in lines 128, by "hypothalamic" in line 237, 1044 and 1128, and by "hypothalamus" in line 1518,

2. Given that only some parameters of systemic metabolism were assessed, I would suggest changing "Metabolism" to "Energy expenditure" in the Extended Data Fig. 10.

Authors' response:

In response to the referee's comment, "Metabolism" has been replaced by "Energy expenditure" in Fig. EV13.

3. The abstract is difficult to read as it contains many abbreviations. I would suggest reducing them to 3, maximum 4.

Authors' response:

In response to the referee's comment, abbreviations in the abstract were reduced.

4. There are several grammar, spelling, and typo errors throughout the text. I would highly recommend proof reading this again.

Authors' response:

In response to the referee's comments, the manuscript has been thoroughly edited for

English language.

Referee #3:

In this study, Kohno et al. investigate the role of FTO in hypothalamic AgRP neurons. Using a comprehensive methodological approach, the authors demonstrate that deletion or overexpression of FTO in AgRP neurons decreases or increases body weight, respectively, primarily due to changes in food intake. FTO was shown to demethylate m6A modifications on exons and introns of pre-mRNA encoding proteins involved in membrane trafficking and alternative splicing. Notably, FTO modulated the splicing of the *Kif1a* gene by enhancing the inclusion of exon 13, thereby affecting the dimerization and function of this protein. In this context, FTO increased dense-core vesicle trafficking and the release of AgRP. Finally, knockdown of *Kif1a* in AgRP neurons overexpressing FTO led to reduced body weight.

This is an interesting study that nicely combines physiological insights with mechanistic exploration of an under-investigated topic (epitranscriptomic modifications), which could have implications for the field of central metabolic control. Overall, the experiments are conducted to a high standard, and the manuscript is well-written. However, I have several concerns and suggestions that the authors should address:

1. While the authors use methodologies that allow specific targeting of AgRP neurons in many parts of the study, in some key sections (e.g., analysis of splice variants), they employ more general approaches (e.g., Tau-Cre mice), which target neurons broadly rather than specifically targeting AgRP neurons. The authors should explicitly state in the text why they transitioned from AgRP-specific to broader neuron-targeting strategies. If this shift is due to limitations in sensitivity or other constraints, this should be clarified. Furthermore, this reasoning should be extended to other parts of the manuscript that do not implement neuron-specific approaches. The authors should also temper the interpretation of data obtained from general neuronal or cell culture experiments, avoiding direct extrapolations to AgRP neurons.

Authors' response:

Thank you for your comments. In response to the referee's comment, we have added an explanation for the shift to neuron-specific *Fto*-KO mice in lines 236-238. We also tempered the interpretation of data obtained from neuron-specific *Fto*-KO mice or cell culture experiments as described in lines 382 and 497-498

2. The immunofluorescence images throughout the manuscript are of poor quality. In many

cases, the images appear overexposed, making it difficult (or even impossible) to discern the staining. Higher-quality, better-balanced images should be provided.

Authors' response:

In response to the referees' comment, we improved most of the fluorescence image quality by adjusting the brightness and contrast levels. In the fluorescent images of NPY-hrGFP in the ARC (Fig. 5A, EV8C), saturation of signal in the cell bodies was unavoidable when visualizing neural fibers.

3. Did the authors include an IgG control for the m6A immunoprecipitation experiments?

Authors' response:

In response to the referee's comment, we added a description of the control IgG experiment in lines 1022–1024.

4. Are m6A sites enriched around exon 13 of Kif1a? Is the regulation of exon 13 inclusion a direct effect of FTO? The authors should consider commenting on this aspect in the manuscript.

Authors' response:

In response to the referee's comment, we have added a discussion regarding the m6A sites around the Kif1a exon13 in lines 467–476.

5. In Figure 3E, the authors integrate RNA-seq with m6A-seq data. This integration should be more clearly explained in the text to aid reader understanding.

Authors' response:

In response to the referee's comment, we have added an explanation regarding the consolidation of RNA-seq data and m6A-seq data in lines 219–220 and improved the description of Fig.3E legend in lines 1511-1512.

6. The electrophysiology experiments (Extended Data Fig. 7E-H) were conducted under basal conditions, which may explain why no changes were observed. Did the authors attempt to stimulate the neurons with a physiological activator (e.g., ghrelin)?

Authors' response:

The activity of NPY/AgRP neurons in response to ghrelin was examined using single-cell Fura-2 Calcium imaging, and we confirmed that the responsiveness of NPY neurons to ghrelin was comparable between control and AgRP neuron-specific *Fto* knockout mice. The results are shown in Fig EV 10E–G and lines 317–320.

7. Similarly, in the hypothalamic explant experiments, the authors use high potassium as a proxy for total AgRP content. This allows to assess the total releasable pool of neuropeptide (a proxy of production). Have they tested AgRP secretion using physiological inducers instead?

Authors' response:

In response to the referee's comment, AgRP secretion from hypothalamic slices of AgRP neuron-specific *Fto*-knockout and *Fto*-overexpression mice was measured under stimulation with low glucose and glutamate, and a significant difference in AgRP release was detected between the genotypes after the stimulation with low glucose and glutamate, as shown in Fig. EV8 I and J and lines 302–305.

8. Figure 8C relies on conventional PCR, which is not a quantitative technique. Using qPCR to specifically target exon 13 would provide more robust and quantitative data.

Authors' response:

In response to the referee's comment, we analyzed the exon 13 inclusion rate using digital PCR under ad libitum, fasting, and refeeding conditions, which is shown in Fig 8E and in lines 396–398, 1354–1363, and 1609–1612.

Dear Dr Kohno,

Thank you for submitting your revised manuscript (EMBOJ-2024-119533R) to The EMBO Journal. Your amended study was sent back to the referees for their scientific reassessment, and we have received re-reports from all of them, which I enclose below. As you will see, the experts state that the work has been substantially enhanced by the revisions and they are now broadly in favour of publication, pending minor amendments.

Thus, we are pleased to inform you that your manuscript has been accepted in principle for publication in The EMBO Journal.

Please carefully consider the remaining minor points raised by referee #3 by adjusting the discussion of the findings and introducing textual changes in the manuscript where appropriate.

Also, we now need you to take care of a number of issues related to formatting and data presentation as detailed below, which should be addressed at re-submission.

Please contact me at any time if you have additional questions related to below points.

As you might have noted on our webpage, every paper at the EMBO Journal now includes a 'Synopsis', displayed on the html and freely accessible to all readers. The synopsis includes a 'model' figure as well as 2-5 one-short-sentence bullet points that summarize the article. I would appreciate if you could provide this figure and the bullet points.

Thank you for giving us the chance to consider your manuscript for The EMBO Journal. I look forward to your final revision.

Again, please contact me at any time if you need any help or have further questions.

Best regards,

Daniel Klimmeck

>> Authors: please define the corresponding author on the manuscript title page.

>> Section order should be corrected as follows: Title page - Abstract & Keywords - Introduction - Results - Discussion - Methods - Data Availability - Acknowledgements - Disclosure and Competing Interests Statement - References - Figure Legends - Table(s) - Expanded View Figure Legends.

>> 'Materials and Methods' should be 'Methods', please move it after the Discussion

>> Figure callouts: Ensure that Figure 6E and Expanded Figure EV12 are called out.

>> Figures in separate files: main figures and EV figures should be uploaded as individual, high resolution figure files. The EV figure legends should be added to the manuscript text, after the main figure legends and under the heading "Expanded View Figure Files".

>> Funding: Please enter the following funding information into our online system: 'Fostering Health Professionals for Changing Needs of Cancer; the Promotion Plan for the Platform of Human Resource Development for Cancer and New Paradigms - Establishing Centres for Fostering Medical Researchers of the Future programs of the Ministry of Education, Culture, Sports, Science and Technology of Japan'.

>> Source data: remove the folder for Fig 6 source data.

>> Consider additional changes and comments from our production team as indicated below:

- Figure legends:

1. Please define the annotated p values ****/****/**/* as well as provide the exact p-values for the same in the legend of figure 7L as appropriate.
2. Please note that the exact p values are not provided in the legends of figures 1B, C, E, F, K, L, M, N; 2A, C, D, E, G, 5E, G, I, K, L, M; 7C, G, J, N; 8B, D, E, G, H; EV4 B, EV6 A-C, EV8 B, D, F, H, I, J
3. Please indicate the statistical test used for data analysis in the legends of figures 1B, C, E, F, K-N; 2A, C, D, E, G, 4A, 5E, G, I, K, L, M; 7L, N; 8B, D, E, G, H; EV3 B-D; EV4 B; EV6A-C; EV8 B, D, F, H, I, J; EV9B, D
4. Please note that information related to n is missing in the legends of figures 2C, D; 4A, 5B, C, E, G, I, K, L, M; 7G, 8D, EV1 H, P
5. Please note that the error bars are not defined in the legends of figures 1B-N, 2A, C-J; 5B, C, E, G, I, K, L, M; 7L, N; 8B, D, E, G, H; EV1 H, P; EV2 A-C; EV4 B, EV6 A-C; EV7 D, EV8B, D, F, H; EV9 B, D; EV10C, D, I
6. Please note that information related to n is missing in the legends of figures 2C, D; 4A, 5B, C, E, G, I, K, L, M; 7G, 8D, EV1 H, P
7. Please note that the error bars are not defined in the legends of figures 1B-N, 2A, C-J; 5B, C, E, G, I, K, L, M; 7L, N; 8B, D, E, G, H; EV1 H, P; EV2 A-C; EV4 B, EV6 A-C; EV7 D, EV8B, D, F, H; EV9 B, D; EV10C, D, I
8. Please note that the white arrow heads are not defined in the legend of figure EV10 H. This needs to be rectified.

Referee #1:

The authors have satisfactorily addressed all of my previous concerns. I do not have any further comments.

Referee #2:

The authors did an excellent job in successfully addressing all of my previous concerns. I wish to congratulate them on this extensive and exciting study!

Referee #3:

The authors have addressed most of my concerns; however, a couple of minor issues remain:

The authors justify the use of the Tau-Cre mouse line "to obtain low background and sufficient amounts of RNA." While the latter point is reasonable, the rationale regarding "low background" is unclear. Please clarify what is meant by "low background" in this

context or consider removing this justification.

I previously requested the inclusion of an IgG control for the m6A immunoprecipitation experiments. The authors now mention in the Methods section that the IP did not yield sufficient RNA and, therefore, the IgG control was omitted. Given the importance of this control, this limitation should be acknowledged more prominently in the manuscript-ideally in the Results section or, alternatively, in the Discussion.

Response to the referee

We thank Referee #3 for the additional constructive comments. We have carefully revised the manuscript to address the remaining concerns. Detailed responses are provided below. Our replies are shown in blue, and the corresponding revisions in the manuscript are highlighted in blue.

Referee comment**Referee #3:**

The authors have addressed most of my concerns; however, a couple of minor issues remain:

The authors justify the use of the Tau-Cre mouse line "to obtain low background and sufficient amounts of RNA." While the latter point is reasonable, the rationale regarding "low background" is unclear. Please clarify what is meant by "low background" in this context or consider removing this justification.

Authors' response:

We appreciate the referee's comment. To clarify the rationale, we have revised the sentence from "To obtain a low background and sufficient amounts of RNA" to "To obtain sufficient amounts of RNA from tissue lacking the effect of FTO-induced demethylation" as shown in line 244 of the revised manuscript.

I previously requested the inclusion of an IgG control for the m6A immunoprecipitation experiments. The authors now mention in the Methods section that the IP did not yield sufficient RNA and, therefore, the IgG control was omitted. Given the importance of this control, this limitation should be acknowledged more prominently in the manuscript-ideally in the Results section or, alternatively, in the Discussion.

Authors' response:

We thank the referee for pointing this out. In response to the comment, we have expanded the description of the m6A-seq analysis in the Results section (lines 205-209), explicitly noting the omission of the IgG control due to insufficient RNA yield and clearly acknowledging this as a technical limitation of the study.

Dear Dr Kohno,

Thank you for submitting the revised version of your manuscript. I have now evaluated your amended manuscript and concluded that the remaining minor concerns have been sufficiently addressed.

I am thus pleased to inform you that your manuscript has been accepted for publication in the EMBO Journal.

On a different note, I would like to alert you that EMBO Press offers a format for a video-synopsis of work published with us, which essentially is a short, author-generated film explaining the core findings in hand drawings, and, as we believe, can be very useful to increase visibility of the work. Please see the following link for representative examples and their integration into the article web page:

<https://www.embopress.org/doi/full/10.15252/emj.2019103932>

Best regards,

Daniel Klimmeck

Daniel Klimmeck, PhD
Senior Editor
The EMBO Journal
EMBO
Postfach 1022-40
Meyerohofstrasse 1
D-69117 Heidelberg
contact@embojournal.org